# COMBINATORIAL-PROBABILISTIC TRADE-OFF: P-VALUES OF COMMUNITY PROPERTIES TEST IN THE STOCHASTIC BLOCK MODELS

**Shuting Shen & Junwei Lu**
Department of Biostatistics
Harvard T.H. Chan School of Public Health
`{shs145@g,junweilu@hsph}.harvard.edu`

## ABSTRACT

We propose an inferential framework testing the general community combinatorial properties of the stochastic block model. We aim to test the hypothesis on whether a certain community property is satisfied, e.g., whether a given set of nodes belong to the same community, and provide p-values for uncertainty quantification. Our framework is applicable to all symmetric community properties. To ease the challenges caused by the combinatorial nature of community properties, we develop a novel shadowing bootstrap method. Utilizing the symmetry, our method can find a shadowing representative of the true assignment and the number of tested assignments in the alternative is largely reduced. In theory, we introduce a combinatorial distance between two community classes and show a combinatorial-probabilistic trade-off phenomenon. Our test is honest as long as the product of the combinatorial distance between two community property classes and the probabilistic distance between two connection probabilities is sufficiently large. Besides, we show that such trade-off also exists in the information-theoretic lower bound. We also implement numerical experiments to show the validity of our method.

## 1 INTRODUCTION

Clustering is an important feature for network studies, which refers to the presence of node communities in the underlying graph. Community partitions the nodes into subgroups, within which a higher level of connectivity is perceived. Its broad spectrum of applications includes the fields of sociology (Wasserman & Faust, 1994), biology (Barabási & Oltvai, 2004), physics (Newman, 2003) and internet (Albert et al., 1999). Stochastic block model (SBM) (Holland et al., 1983) is one of the most widely studied statistical models depicting the network community structures. It is a random graph model which divides the nodes into disjoint communities and assigns the probability of connection between two nodes according to their community memberships. One of its central problems in previous studies is community detection. However, most of the existing research focused on estimating the community labeling without uncertainty quantification (Choi et al., 2012; Mossel et al., 2012; Airoldi et al., 2013; Massoulié, 2014; Abbe et al., 2016; Mossel et al., 2016). Some fundamental limits of community recovery have also been established in previous studies. For example, Abbe et al. (2016) showed the optimal phase transition for the exact recovery of the community assignments using the maximum likelihood. The semi-definite relaxation methods (Abbe et al., 2016; Hajek et al., 2016; Agarwal et al., 2017; Bandeira, 2018) and the spectral methods (Yun & Proutiere, 2014; Abbe & Sandon, 2015; Gao et al., 2017; Abbe et al., 2020) are also shown to be optimal in exact recovery. Besides the exact recovery, Zhang & Zhou (2016) quantified the statistical rate of the community estimation via the mismatch ratio and showed the minimax rate of the mismatch ratio for community detection. In summary, the community estimation studies have two major limits: 1) it does not provide the p-values to evaluate the uncertainty of the estimation, which are essential in many scientific applications, and 2) it requires the recovery of community assignments for all nodes, while in many scientific applications, we are interested in the community properties of a specific subset of nodes, e.g., whether two sets of nodes belong to the same community. We formulate the following examples of statistical hypotheses for illustration.

**Example 1.1** (Same community test for $m$ nodes). We want to test whether $m$ given nodes are in the same cluster or not. Without loss of generality, we have the hypothesis:

$\text{H}_0$ : Nodes $1, \ldots, m$ belong to the same community,

$\text{H}_1$ : There exists two nodes $1 \leq j \neq k \leq m$ belonging to two different communities.

**Example 1.2** (Group community test). For two groups of nodes, we know in prior that nodes within each group belong to the same community. We aim to further test whether these two groups belong to the same community. We denote one node set as $S_m = \{1, \ldots, m\}$ and the other as $S_{m'} = \{m+1, \ldots, m+m'\}$. The group community hypothesis is

$\text{H}_0$ : Nodes in $S_m \cup S_{m'}$ belong to the same community,

$\text{H}_1$ : Nodes in $S_m$ belong to community $a$, but nodes in $S_{m'}$ belong to community $b \neq a$.

**Example 1.3** (Equal-sized communities test). Given an SBM of $n$ nodes and $K$ communities, we aim to test whether each community has the same size. Namely, we aim to test the hypothesis:

$\text{H}_0$ : Each community has the size $n/K$,

$\text{H}_1$ : At least one of the communities has size not equal to $n/K$.

In order to conduct hypothesis tests including the above examples, we develop a general community property test. We consider the SBM with $n$ nodes and $K$ communities. Denote the community assignment of the nodes by $z = (z(1), ..., z(n)) \in \{1, \ldots, K\}^n$. The homogeneous SBM assumes that the edges of the random graph are independent Bernoulli random variables with connection probability $p$ if $z(i) = z(j)$ and $q$ if $z(i) \neq z(j)$. We reparameterize $p, q$ as $p = \rho_n \lambda_1$ and $q = \rho_n \lambda_2$, where $\lambda_1$ and $\lambda_2$ are constants independent of $n$, and $\rho_n$ is the signal strength. Let $\mathcal{C}_0, \mathcal{C}_1 \subset \{1, \ldots, K\}^n$ be two disjoint community assignment families. We are interested in the *general community property test*:

$$\text{H}_0 : z \in \mathcal{C}_0 \text{ versus } \text{H}_1 : z \in \mathcal{C}_1. \tag{1.1}$$

We characterize the hardness of the test by two kinds of "distances": the probabilistic distance between $p$ and $q$, and the combinatorial distance between $\mathcal{C}_0$ and $\mathcal{C}_1$. The existing literature on community detection only focused on the probability distance, e.g., the Rényi divergence $I(p, q) = -2 \log \left( \sqrt{pq} + \sqrt{(1-p)(1-q)} \right)$ (Zhang & Zhou, 2016). In comparison, our paper introduces a novel combinatorial distance between $\mathcal{C}_0$ and $\mathcal{C}_1$ denoted as $d(\mathcal{C}_0, \mathcal{C}_1)$ (see Definition 2.4) and proposes a general testing method that is honest and powerful when

$$\text{Combinatorial-Probabilistic Trade-Off: } I(p, q) d(\mathcal{C}_0, \mathcal{C}_1) = \Omega(n^\epsilon) \tag{1.2}$$

for some arbitrarily small $\epsilon > 0$. On the other hand, we show the minimax lower bound of the test in the sense that $\text{H}_0$ and $\text{H}_1$ in (1.1) cannot be differentiated when $I(p, q) d(\mathcal{C}_0, \mathcal{C}_1) \leq c \log n$ for some constant $c > 0$.[1] The multiplication between $I(p, q)$ and $d(\mathcal{C}_0, \mathcal{C}_1)$ reveals the trade-off between the probabilistic distance and the combinatorial distance in the general community property test.

## 2 COMMUNITY PROPERTIES OF THE STOCHASTIC BLOCK MODEL

In our paper, we consider the fixed assignment stochastic block model, denoted by $\mathcal{M}(n, K, p, q, z)$. Denote by $[n] = \{1, \ldots, n\}$ for any integer $n$. For simplicity, we start with the scenario where the community sizes are even, and will generalize to the uneven case in Appendix B. We denote the even assignment class by $\mathcal{K}^n := \{z \in [K]^n : |\{i : z(i) = k\}| = n/K, \forall k \in [K]\}$. In our paper, we assume $K$ to be bounded. Let $\mathbf{A} \in \{0, 1\}^{n \times n}$ be the symmetric adjacency matrix of the random graph generated from the SBM. In the following part of the paper, we will study the community property test with an observation of the adjacency matrix $\mathbf{A} \sim \mathcal{M}(n, K, p, q, z)$.

### 2.1 SYMMETRIC COMMUNITY PROPERTIES

In this section, we aim to define the community property and the distance between two community families. In general, we say a community property is a subset of $[K]^n$. However, such a definition is too general and may include some ill-posed examples. For instance, if we can transfer one

---

[1]We refer to Theorem 3.2 and Theorem 4.1 for the rigorous arguments about the upper and lower bounds.

assignment to another under certain permutation of the community labels, they are essentially the same assignment and should belong to the same community property. This motivates us to give the following definition of equivalent assignments.

**Definition 2.1** (Equivalent community assignments). Let $S_K$ be the symmetric group containing all bijections from $[K]$ to itself. We say two assignments $z$ and $z' \in [K]^n$ are equivalent, denoted as $z \simeq z'$, if there exists a permutation $\sigma \in S_K$ such that $\sigma(z) = z'$. Here $\sigma(z)$ means implementing the permutation $\sigma$ to each entry of the vector $z$. More generally, given a node set $\mathcal{N} \subseteq [n]$, we denote by $z_\mathcal{N}$ the sub-vector of $z$ with entries in $\mathcal{N}$. We say $z_\mathcal{N} \simeq z'_\mathcal{N}$ if there exists a permutation $\sigma \in S_K$ such that $\sigma(z_\mathcal{N}) = z'_\mathcal{N}$.

With Definition 2.1, we give the definition of symmetric community properties as follows.

**Definition 2.2** (Symmetric community properties). We say a community property $\mathcal{C}_0$ is symmetric, if there exist a node set $\mathcal{N} \subseteq [n]$ and an assignment $\widetilde{z} \in \mathcal{K}^n$, such that $\mathcal{C}_0 = \{z \in \mathcal{K}^n : z_\mathcal{N} \simeq \widetilde{z}_\mathcal{N}\}$. We say some $\mathcal{C}_1 \subseteq \mathcal{K}^n \backslash \mathcal{C}_0$ is an alternative property of $\mathcal{C}_0$ if $\mathcal{C}_1$ is closed under permutations on the support $\mathcal{N}$, i.e., for any $z \in \mathcal{C}_1 \subseteq \mathcal{K}^n \backslash \mathcal{C}_0$, if some $z'$ satisfies $z'_\mathcal{N} \simeq z_\mathcal{N}$, then $z' \in \mathcal{C}_1$ as well.

Intuitively, the node set $\mathcal{N}$ and the assignment $\widetilde{z}$ in Definition 2.2 are the representative node set and assignment generating all possible assignments in the community property via permutation. The community property $\mathcal{C}_0$ is "symmetric" in the sense that all its assignments are equivalent on the support of node set $\mathcal{N}$. We impose the following assumption on testing symmetric properties.

**Assumption 2.1** (Symmetric community property test). In the hypothesis test $H_0 : z \in \mathcal{C}_0$ v.s. $H_1 : z \in \mathcal{C}_1$, we assume $\mathcal{C}_0, \mathcal{C}_1 \subseteq \mathcal{K}^n$ and $\mathcal{C}_0$ is symmetric and $\mathcal{C}_1$ is an alternative property of $\mathcal{C}_0$.

By Definition 2.2, $\mathcal{C}_1 = \mathcal{C}_0^c$ is an alternative property of $\mathcal{C}_0$. Meanwhile, $\mathcal{C}_1$ satisfying the assumption above could be a strict subset of $\mathcal{C}_0^c$, allowing more examples in practice. In fact, we can show that Examples 1.1 and 1.2 satisfies Assumption 2.1, whose concrete forms of $\mathcal{N}$ and $\widetilde{z}$ are given below.

• Example 1.1: Same community test for $m$ nodes. We can define

$$\mathcal{C}_0 = \{z \in \mathcal{K}^n : z(1) = \cdots = z(m)\} \text{ and } \mathcal{C}_1 = \mathcal{K}^n \backslash \mathcal{C}_0, \tag{2.1}$$

while $\mathcal{N} = [m]$ and $\widetilde{z}$ is any assignment satisfying $\widetilde{z}_\mathcal{N} = (1, \ldots, 1) \in [K]^m$.

• Example 1.2: Same community test for groups. We have

$$\begin{aligned}
\mathcal{C}_0 &= \{z \in \mathcal{K}^n : z(1) = \cdots = z(m) = z(m+1) = \cdots = z(m+m')\} \\
\mathcal{C}_1 &= \{z \in \mathcal{K}^n : z(1) = \cdots = z(m) \neq z(m+1) = \cdots = z(m+m')\},
\end{aligned} \tag{2.2}$$

where $\mathcal{C}_0$ is symmetric by choosing $\mathcal{N} = [m + m']$ and $\widetilde{z}_\mathcal{N} = (1, \ldots, 1) \in [K]^{m+m'}$.

## 2.2 COMBINATORIAL DISTANCE BETWEEN COMMUNITY PROPERTIES

In order to depict the relationship between $\mathcal{C}_0$ and $\mathcal{C}_1$, we propose a metric of distance in terms of the number of misaligned edges. We first define the set of misaligned edges between two assignments.

**Definition 2.3.** For any two assignments $z_0 \in \mathcal{C}_0$ and $z_1 \in \mathcal{C}_1$, we define the two sets of misaligned edges as

$$\begin{aligned}
\mathcal{E}_1(z_0, z_1) &= \{(i, j) : i < j, i, j \in [n], z_0(i) = z_0(j), z_1(i) \neq z_1(j)\} \text{ and} \\
\mathcal{E}_2(z_0, z_1) &= \{(i, j) : i < j, i, j \in [n], z_0(i) \neq z_0(j), z_1(i) = z_1(j)\},
\end{aligned}$$

where $\mathcal{E}_1(z_0, z_1)$ contains the edges whose corresponding nodes are assigned to the same community in $z_0$ but to two different communities by $z_1$, and $\mathcal{E}_2(z_0, z_1)$ is the opposite. See Figure 1 for illustration. We define $n_i(z_0, z_1) = |\mathcal{E}_i(z_0, z_1)|$, for $i = 1, 2$ as the cardinality of the two edge sets.

Now we are ready to propose the metric of assignment distance defined as follows.

**Definition 2.4** (Community property distance). We define the distance between two assignments $z_0$ and $z_1$ as $d(z_0, z_1) = n_1(z_0, z_1) \vee n_2(z_0, z_1)$. Correspondingly, we also define $d(z_0, \mathcal{C}_1) = \inf_{z_1 \in \mathcal{C}_1} d(z_0, z_1)$ and the distance between two community properties

$$d(\mathcal{C}_0, \mathcal{C}_1) = \inf_{z_0 \in \mathcal{C}_0, z_1 \in \mathcal{C}_1} d(z_0, z_1).$$

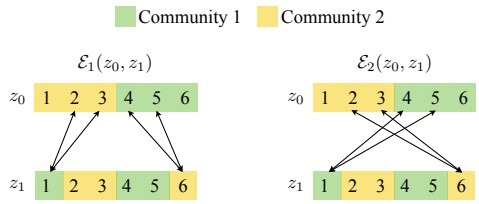

Figure 1: Example of misaligned edges in Definition 2.3.

By definition, the distance $d(\mathcal{C}_0, \mathcal{C}_1)$ is the minimal number of misaligned edges between $\mathcal{C}_0$ and $\mathcal{C}_1$. We refer the computation of $d(\mathcal{C}_0, \mathcal{C}_1)$ for more general examples to Section 2.4.

### 2.3 LIKELIHOOD-RATIO TEST FOR COMMUNITY PROPERTIES

Our method starts with defining a likelihood-ratio test statistic. We denote the observed adjacency matrix from the true model as $\mathbf{A} \sim \mathcal{M}(n, K, p, q, z^*)$, where $z^*$ is the true assignment. The likelihood function of the stochastic block model is

$$f(\mathbf{A}; z, p, q) = \Pi_{i<j} p^{\mathbb{1}(z(i)=z(j))\mathbf{A}_{ij}} (1-p)^{\mathbb{1}(z(i)=z(j))(1-\mathbf{A}_{ij})} q^{\mathbb{1}(z(i)\neq z(j))\mathbf{A}_{ij}} (1-q)^{\mathbb{1}(z(i)\neq z(j))(1-\mathbf{A}_{ij})}.$$

We then denote the log-likelihood ratio statistic as

$$\text{LRT} = \log \frac{\sup_{z\in\mathcal{C}_1} f(\mathbf{A}; z, p, q)}{\sup_{z\in\mathcal{C}_0\cup\mathcal{C}_1} f(\mathbf{A}; z, p, q)}. \tag{2.3}$$

In order to conduct the property test, we study the limiting distribution of the likelihood ratio statistic. In specific, when $1/\rho_n = o(n^{1-c_2})$ for some constant $c_2 > 0$, we are able to decompose the LRT as follows

$$\text{LRT} = \sup_{z\in\mathcal{C}_1} \log f(\mathbf{A}; z, p, q) - \log f(\mathbf{A}; z^*, p, q) + o(1)$$

$$= \sup_{z\in\mathcal{C}_1} g(p, q)\left( \sum_{(i,j)\in\mathcal{E}_2(z^*,z)} \mathbf{A}_{ij} - \sum_{(i,j)\in\mathcal{E}_1(z^*,z)} \mathbf{A}_{ij} \right) + o(1), \tag{2.4}$$

where $g(p, q) = \log p(1-q)/(q(1-p))$. We observe that the leading term in (2.4) is the difference of edges in two edge sets: $\mathcal{E}_2(z^*, z)$ and $\mathcal{E}_1(z^*, z)$. By Definition 2.4, the property distance $d(\mathcal{C}_0, \mathcal{C}_1)$ is larger when the two edge sets are larger, which makes the leading term larger as well. This implies why $d(\mathcal{C}_0, \mathcal{C}_1)$ characterizes the difficulty of the test.

**Remark 2.1.** The likelihood ratio statistic in (2.3) is similar to the one in Wang & Bickel (2017). They considered the hypothesis on a specific community property: the number of communities.

$$\mathcal{C}_0 = \{z|z \in [K-1]^n\} \text{ and } \mathcal{C}_1 = \{z|z \in [K]^n\}\backslash\mathcal{C}_0. \tag{2.5}$$

They showed that the suprema $\sup_{z\in\mathcal{C}_1} f(\mathbf{A}; z, p, q)$ in (2.3) is unique, as illustrated in Figure 2(a), which makes their LRT asymptotically normal for $\mathcal{C}_0, \mathcal{C}_1$ in (2.5). However, this is not always true for the general community properties. For some properties, there will be an exponential number of candidate assignments maximizing the likelihood in (2.3), as illustrated in Figure 2(b). Thus the LRT is no longer asymptotically normal for the general case. Therefore, despite the similar formality of the LRT, our testing procedure will be different from theirs.

To characterize the suprema in the LRT, we define the boundary of the alternative properties.

**Definition 2.5** (Boundary of community class). For a given null assignment $z_0 \in \mathcal{C}_0$, we define the boundary of $\mathcal{C}_1$ as:

$$B_{z_0} = \big\{ z \in \mathcal{C}_1 : d(z_0, z) = d(z_0, \mathcal{C}_1) \big\}.$$

Namely, $B_{z_0}$ is the projection of $z_0$ onto $\mathcal{C}_1$. Our analysis shows that under the null, the likelihood maximizer is asymptotically equivalent to the boundary $B_{z*}$.[2] In the next section, we will study the asymptotic property of the LRT under such case by studying the structure of $B_{z_0}$.

---

[2]See Lemma D.2 in the Appendix for the rigorous argument.

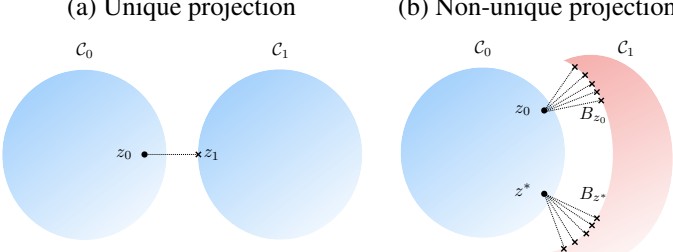

Figure 2: The boundary of $\mathcal{C}_1$ given $z_0$. On the left panel, the projection is unique and $B_{z_0} = z_1$. On the right panel, the projection is not unique, where $z_0$ is the shadowing assignment of $z^*$, and $B_{z^*}$ and $B_{z_0}$ have similar structures.

## 2.4 SHADOWING BOOTSTRAP FOR THE PROPERTY TEST

There are two major challenges to estimate the p-value of LRT. First, the suprema of the likelihood are not unique and therefore the limiting distribution of LRT is not necessarily normal. By (2.4), we can in turn study the limiting distribution of the leading term

$$L := \sup_{z \in \mathcal{C}_1} g(p, q) \left( \sum_{(i,j) \in \mathcal{E}_2(z^*, z)} \mathbf{A}_{ij} - \sum_{(i,j) \in \mathcal{E}_1(z^*, z)} \mathbf{A}_{ij} \right). \tag{2.6}$$

Chernozhukov et al. (2013) studied the limiting distribution of the maximal of high dimensional empirical process and proposed to estimate its quantile by multiplier bootstrap. However, their method imposes that the dimension $d$ of the empirical process and the sample size $n$ satisfies $\log d/n^{1/5} = o(1)$, whereas in (2.6), the dimension $d = |\mathcal{C}_1|$ could be of the order $K^n$ and violates the scaling condition. To handle this problem, our key observation is that the supremum over the alternative $\mathcal{C}_1$ can be represented by the supremum over its boundary $B_{z*}$. In particular, we show that the leading term $L$ is asymptotically the same as the following statistic:

$$L_0 := \sup_{z \in B_{z^*}} g(p, q) \left( \sum_{(i,j) \in \mathcal{E}_2(z^*, z)} \mathbf{A}_{ij} - \sum_{(i,j) \in \mathcal{E}_1(z^*, z)} \mathbf{A}_{ij} \right). \tag{2.7}$$

The cardinality of $B_{z^*}$ is much smaller than that of $\mathcal{C}_1$ (usually polynomial to $n$), and therefore satisfies the scaling condition of high dimensional multiplier bootstrap. However, we cannot construct $B_{z^*}$ in practice as $z^*$ is unknown. This leads to the second challenge: how to find $B_{z^*}$ in practice? Our key insight to solve the second challenge is to utilize the symmetry property in Definition 2.2. This insight relies on the following lemma characterizing the covariance of two processes.

**Lemma 2.2** (Shadowing symmetry). For a given $z \in \mathcal{C}_0$, we list the assignments in the boundary $B_z$ as $z_1, z_2, \ldots, z_{|B_z|}$. Define a $|B_z|$-dimensional vector $\boldsymbol{L}_z$ as

$$(\boldsymbol{L}_z)_k = g(p, q) \left( \sum_{(i,j) \in \mathcal{E}_2(z, z_k)} \mathbf{A}_{ij} - \sum_{(i,j) \in \mathcal{E}_1(z, z_k)} \mathbf{A}_{ij} \right), \text{ for } k = 1, 2, \ldots, |B_z|.$$

Suppose Assumption 2.1 holds. For any $z_0, z_0' \in \mathcal{C}_0$, we have $|B_{z_0}| = |B_{z_0'}|$ and $\mathrm{Cov}(\boldsymbol{L}_{z_0})$ equals to $\mathrm{Cov}(\boldsymbol{L}_{z_0'})$ up to permutation, i.e., there exists a permutation $\tau \in S_{|B_{z_0}|}$ such that $\mathrm{Cov}(\boldsymbol{L}_{z_0})_{kl} = \mathrm{Cov}(\boldsymbol{L}_{z_0'})_{\tau(k)\tau(l)}$ for all $k, l = 1, \ldots, |B_{z_0}|$.

We refer to Appendix F.1 for the proof of Lemma 2.2. Therefore, we can avoid directly constructing $B_{z^*}$ and choose instead any $z \in \mathcal{C}_0$ as a "shadowing assignment" to construct the shadowing statistic

$$L_0(z_0) := \sup_{z \in B_{z_0}} g(p, q) \left( \sum_{(i,j) \in \mathcal{E}_2(z_0, z)} \mathbf{A}_{ij} - \sum_{(i,j) \in \mathcal{E}_1(z_0, z)} \mathbf{A}_{ij} \right). \tag{2.8}$$

The following proposition shows that the quantiles of $L_0(z_0)$ and $L_0$ are asymptotically the same.

**Proposition 2.3.** Suppose Assumption 2.1 holds, $\log |B_{z^*}| = O(\log n)$ and $1/\rho_n = o(n^{1-c_2})$ for some constant $c_2 > 0$. For any $z_0 \in \mathcal{C}_0$, we have

$$\lim_{n \to \infty} \sup_{t \in \mathbb{R}} |\mathbb{P}(L_0 < t) - \mathbb{P}(L_0(z_0) < t)| = 0.$$

We defer the proof to Appendix D.1. Now we are ready to present the shadowing bootstrap procedure. Based on the previous discussion, we aim to estimate the p-value based on $L_0(z_0)$. To achieve this, we take an arbitrary $z_0 \in \mathcal{C}_0$, and generate one realization of the adjacency matrix $\widehat{\mathbf{A}} \sim \mathcal{M}(n, K, \widehat{p}, \widehat{q}, z_0)$. Here $\widehat{p}$ and $\widehat{q}$ are the maximum likelihood estimator

$$(\widehat{p}, \widehat{q}) = \mathrm{argmax}_{(p,q)} \sup_{z \in \mathcal{C}_0 \cup \mathcal{C}_1} f(\mathbf{A}; z, p, q) \tag{2.9}$$

The likelihood ratio statistic is

$$\widehat{\mathrm{LRT}} = \log \sup_{z \in \mathcal{C}_1} f(\mathbf{A}; z, \widehat{p}, \widehat{q}) - \log \sup_{z \in \mathcal{C}_0 \cup \mathcal{C}_1} f(\mathbf{A}; z, \widehat{p}, \widehat{q}). \tag{2.10}$$

The next step is to find the assignments in $B_{z_0}$. We can construct $B_{z_0}$ by Definition 2.5 in general. We will show later how to construct $B_{z_0}$ for the concrete examples. To estimate the p-value, we apply the Gaussian multiplier bootstrap. Let $\{e_{ij}\}_{1 \leq i < j \leq n}$ be independent standard Gaussian random variables and define

$$W_n = \sup_{z \in B_{z_0}} \sum_{1 \leq i < j \leq n} \left(\widehat{\mathbf{A}}_{ij} - \mathbb{E}_{\widehat{p}, \widehat{q}}(\widehat{\mathbf{A}}_{ij})\right)\left(\mathbb{1}[(i,j) \in \mathcal{E}_2(z_0, z)] - \mathbb{1}[(i,j) \in \mathcal{E}_1(z_0, z)]\right)e_{ij}, \tag{2.11}$$

where $\mathbb{E}_{\widehat{p}, \widehat{q}}(\widehat{\mathbf{A}}_{ij}) = \widehat{p}$ if $z_0(i) = z_0(j)$ and $\mathbb{E}_{\widehat{p}, \widehat{q}}(\widehat{\mathbf{A}}_{ij}) = \widehat{q}$ otherwise. Then we estimate the p-value via the multiplier bootstrap

$$p_W = \mathbb{P}\left(g(\widehat{p}, \widehat{q})W_n + g(\widehat{p}, \widehat{q})\widehat{\mu}_0 \geq \widehat{\mathrm{LRT}} | \widehat{\mathbf{A}}, \mathbf{A}\right), \tag{2.12}$$

where $\widehat{\mu}_0 = d(\mathcal{C}_0, \mathcal{C}_1)(\widehat{q} - \widehat{p})$ is the estimator of the mean of the process in (2.8). Finally, we reject the null $\mathrm{H}_0 : z^* \in \mathcal{C}_0$ if $p_W \leq \alpha$ and do not reject $\mathrm{H}_0$ otherwise.

Now we provide concrete steps to construct $B_{z_0}$ and calculate $d(\mathcal{C}_0, \mathcal{C}_1)$ for Examples 1.1 and 1.2.

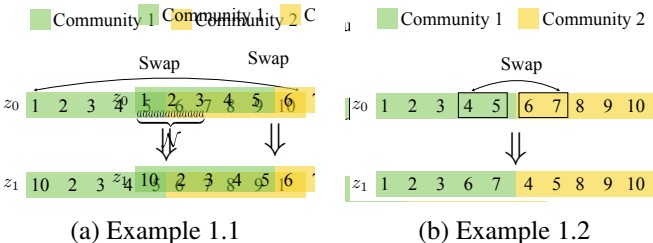

(a) Example 1.1                    (b) Example 1.2

Figure 3: Procedure to construct the assignment in the boundary $B_{z_0}$. Panel (a) is to test whether the first 3 nodes belong to the same community, and Panel (b) is to test whether node sets $\{1, 2, 3\}$ and $\{4, 5\}$ belong to the same community.

● Example 1.1: For any $z_0 \in \mathcal{C}_0$, to construct $B_{z_0}$, we aim to find assignments whose distance to $z_0$ is $d(z_0, \mathcal{C}_1)$. The simplest way is to exchange the community assignment of one node $s \in [m]$ with another node $s'$ from a different community (see $z_1$ in Figure 3(a) for an example when $n = 10$, $m = 3$, and $K = 2$). It is easy to check all such assignments belong to $B_{z_0}$. On the other hand, any other operation will incur more node-wise misclassification and the edge-wise misalignment will be much larger. As for $d(\mathcal{C}_0, \mathcal{C}_1)$, we start with evaluating $n_1(z_0, z_1)$ for some $z_1 \in B_{z_0}$. The edges whose connection probability is changed from $p$ to $q$ will be the edges between the swapped nodes and the rest of the nodes in their original communities. Therefore, we have $n_1(z_0, z_1) = 2(n/K-1)$. Similarly, $n_2(z_0, z_1) = 2(n/K - 1)$ and thus $d(\mathcal{C}_0, \mathcal{C}_1) = 2(n/K - 1)$. In summary, $B_{z_0}$ is composed of all the assignments obtained from swapping two nodes from different communities in $z_0$. The distance between two classes is $d(\mathcal{C}_0, \mathcal{C}_1) = 2(n/K - 1)$.

• Example 1.2: In this example, without loss of generality, we can assume that $m' \leq m$. Then to project an arbitrary $z_0 \in \mathcal{C}_0$ onto $\mathcal{C}_1$, we will exchange the cluster assignment of the set $\mathcal{S}_{m'} = \{m+1, \ldots, m+m'\}$ with another set $\mathcal{S}'_{m'}$ from a different cluster of cardinality $m'$ to obtain the smallest number of misaligned edges. See Figure 3(b). Correspondingly, for $z_1 \in B_{z_0}$, we have $d(z_0, z_1) = n_1(z_0, z_1) = n_2(z_0, z_1) = 2m'(n/K - m')$, and thus $d(\mathcal{C}_0, \mathcal{C}_1) = 2m'(n/K - m')$. In summary, $B_{z_0}$ is composed of all the assignments which can be obtained from reassigning the label of nodes $m+1, \ldots, m+m'$ in $z_0$. The distance between two classes is $d(\mathcal{C}_0, \mathcal{C}_1) = 2m'(n/K - m')$.

## 3 VALIDITY OF COMMUNITY PROPERTY TEST

In this section, we show that our testing method is honest and powerful. Before presenting our theorems, we first give the following assumption for the alternative class $\mathcal{C}_1$.

**Assumption 3.1** (Scattering of $\mathcal{C}_1$). For any $z_0 \in \mathcal{C}_0$, we have $|B_{z_0}| = O(n^{c_0})$ for some constant $c_0 > 0$.

We call this assumption the *scattering assumption* as it ensures that the assignments in $\mathcal{C}_1$ are scattered and not too concentrated on the boundary. In general, if relabeling a constant number of nodes can change an assignment from $\mathcal{C}_0$ to $\mathcal{C}_1$, then $|B_{z_0}|$ will be upper bounded by $\binom{n}{c_0} = O(n^{c_0})$ for some integer $c_0 > 0$. The following main theorem shows that our test is honest and powerful for general symmetric community properties.

**Theorem 3.2.** Suppose Assumptions 2.1 and 3.1 hold, $d(\mathcal{C}_0, \mathcal{C}_1) = o(n^{c_1})$ for some constant $c_1 < 2$, and $1/\rho_n = o(n^{1-c_2})$ for some constant $c_2 > 0$. We have

$$\lim_{n \to \infty} \sup_{z^* \in \mathcal{C}_0} \mathbb{P}(p_W \leq \alpha) = \alpha \text{ and } \lim_{n \to \infty} \sup_{z^* \in \mathcal{C}_0} \mathbb{P}(\text{reject H}_0) = \alpha.$$

Moreover, if $d(\mathcal{C}_0, \mathcal{C}_1)I(p, q) = \Omega(n^{\varepsilon})$ for some arbitrarily small constant $\varepsilon > 0$, we have

$$\lim_{n \to \infty} \inf_{z^* \in \mathcal{C}_1} \mathbb{P}(\text{reject H}_0) = 1.$$

We defer the proof to Appendix D.2. One may note that under the regime of Theorem 3.2, exact recovery can be achieved. However, previous exact recovery methods cannot provide uncertainty quantification. On the other hand, our approach can provide the p-values for the general community properties which is more challenging compared to estimating the community labeling. We now apply Theorem 3.2 to Examples 1.1 and 1.2 by checking Assumption 3.1. The following proposition shows that Assumption 3.1 is satisfied for Examples 1.1 and 1.2.

**Proposition 3.3.** For Example 1.1, we have $|B_{z_0}| = O(mn)$ and $d(\mathcal{C}_0, \mathcal{C}_1) = 2(n/K - 1) = O(n/K)$. For Example 1.2, $|B_{z_0}| = O(K(n/K)^{m \wedge m'})$ and $d(\mathcal{C}_0, \mathcal{C}_1) = 2(m \wedge m')(n/K - m \wedge m')$.

Plugging these results to the general Theorem 3.2, we have the following corollary.

**Corollary 3.4** (Examples 1.1 and 1.2). Suppose $1/\rho_n = o(n^{1-c_2})$ for some constant $c_2 > 0$. If $m \leq n/K$ for Example 1.1 or $m \wedge m' = O(1)$ for Example 1.2, we have $\lim_{n \to \infty} \sup_{z^* \in \mathcal{C}_0} \mathbb{P}(\text{reject H}_0) = \alpha$. Moreover, if $I(p, q)n/K = \Omega(n^{\varepsilon})$ for some small positive constant $\varepsilon$, we have $\lim_{n \to \infty} \sup_{z^* \in \mathcal{C}_1} \mathbb{P}(\text{reject H}_0) = 1$.

## 4 INFORMATION-THEORETIC LOWER BOUND

In this section, we discuss the information-theoretic lower bound of community property test. We will give the lower bound of the minimax risk of all possible test $\psi$ defined as

$$r(\mathcal{C}_0, \mathcal{C}_1) = \inf_{\psi} \left\{ \sup_{z \in \mathcal{C}_0} \mathbb{P}_z(\psi = 1) + \sup_{z \in \mathcal{C}_1} \mathbb{P}_z(\psi = 0) \right\}.$$

We will show that the combinatorial-probabilistic trade-off phenomenon appears in the lower bound as well, thus it essentially characterizes the hardness of the community property test. In order to establish the lower bound, we first introduce the concept of packing number for $\mathcal{C}_1$. A key element in defining the packing number is the metric assigned to $\mathcal{C}_1$. Our first insight is that the more misaligned edges there are between $\mathcal{C}_0$ and $\mathcal{C}_1$, the easier it is to differentiate $\mathcal{C}_1$ from $\mathcal{C}_0$. This motivates us to consider the misaligned edge set $\mathcal{E}_{1,2}(z_0, z_1) = \mathcal{E}_1(z_0, z_1) \cup \mathcal{E}_2(z_0, z_1)$ and use its

cardinality as a "metric". Our second insight is that how hard it is to differentiate $\mathcal{C}_1$ from $\mathcal{C}_0$ does not depend on the complexity of the entire $\mathcal{C}_1$ but the boundary set $B_{z_0}$, which is also implied by our shadowing bootstrap statistic in (2.11). Therefore, we give the following definition of packing number of $B_{z_0}$ to characterize the hardness of test.

**Definition 4.1** ($\varepsilon$-packing of $B_{z_0}$)**.** For any $z_0 \in \mathcal{C}_0$, we say $\{z_1, z_2, ..., z_N\} \subseteq B_{z_0}$ is an $\varepsilon$-packing of $B_{z_0}$, if for any $z_j \neq z_k$ we have $|\mathcal{E}_{1,2}(z_0, z_j) \cap \mathcal{E}_{1,2}(z_0, z_k)| \leq \varepsilon$. The $\varepsilon$-packing number of $B_{z_0}$, denoted as $N(B_{z_0}, \varepsilon)$, is the maximum cardinality of any $\varepsilon$-packing of $B_{z_0}$.

By Definition 2.5, $B_{z_0}$ collects the alignments in $\mathcal{C}_1$ which are closest to $z_0 \in \mathcal{C}_0$, which are the hardest cases to test. The following theorem shows the lower bound of the community property test can be characterized by the packing number of these hardest cases.

**Theorem 4.1.** Suppose $\mathcal{C}_0, \mathcal{C}_1 \subseteq \mathcal{K}^n$, $1/\rho_n = o(n^{1-c_2})$ for some constant $c_2 > 0$ and $p \leq 1 - \delta$ for some constant $\delta > 0$. If there exists a $z_0 \in \mathcal{C}_0$ such that $\log N(B_{z_0}, \sqrt{d(z_0, \mathcal{C}_1)}) = O(\log n)$ and

$$\limsup_{n \to \infty} \frac{d(z_0, \mathcal{C}_1) I(p, q)}{\log N(B_{z_0}, \sqrt{d(z_0, \mathcal{C}_1)})} < 1, \tag{4.1}$$

then $\liminf_{n \to \infty} r(\mathcal{C}_0, \mathcal{C}_1) \geq 1/2$.

We defer the proof to Appendix E.1. From (4.1), we see that the packing entropy $\log N(B_{z_0}, \sqrt{d(z_0, \mathcal{C}_1)})$ is the lower bound of the signal strength. In general, the packing entropy is $O(\log n)$. Comparing to the upper bound $d(\mathcal{C}_0, \mathcal{C}_1) I(p, q) = \Omega(n^\varepsilon)$ for some arbitrarily small constant $\varepsilon > 0$ in Theorem 3.2, there is a gap to $O(\log n)$ in the lower bound. We conjecture that this gap exists as both our upper and lower bounds are for general community property test. We will find a finer analysis in future research. The following theorem gives an alternative lower bound result relaxing the scaling conditions in Theorem 4.1.

**Theorem 4.2.** Suppose $\mathcal{C}_0, \mathcal{C}_1 \subseteq \mathcal{K}^n$, $0 < q < p \leq 1 - \delta$ for some constant $\delta > 0$ and $\lim_{n \to \infty} d(\mathcal{C}_0, \mathcal{C}_1) p = \infty$. If one of the following conditions:

(1) $d(\mathcal{C}_0, \mathcal{C}_1) I(p, q) \leq c$ for some sufficiently small constant $c$;

(2) $\lim_{n \to \infty} d(\mathcal{C}_0, \mathcal{C}_1) I(p, q) = \infty$, but there exists a $z_0 \in \mathcal{C}_0$ such that

$$\limsup_{n \to \infty} \frac{d(z_0, \mathcal{C}_1) I(p, q)}{\log N(B_{z_0}, 0)} < 1,$$

is satisfied, then $\liminf_{n \to \infty} r(\mathcal{C}_0, \mathcal{C}_1) \geq 1/2$.

We defer the proof to Appendix E.1. Theorems 4.1 and 4.2 both show the lower bound with the combinatorial-probabilistic trade-off. Theorem 4.1 has a sharper lower bound on $d(z_0, \mathcal{C}_1) I(p, q)$ under a stronger scaling condition. In comparison, Theorem 4.2 has a less sharp lower bound with weaker scaling conditions. When $d(\mathcal{C}_0, \mathcal{C}_1) I(p, q)$ is bounded, we cannot differentiate two hypotheses. When $d(\mathcal{C}_0, \mathcal{C}_1)$ goes to infinity, Theorem 4.2 condition (2) gives the lower bound $d(\mathcal{C}_0, \mathcal{C}_1) I(p, q) < \log N(B_{z_0}, 0)$. If we have stronger scaling conditions in Theorem 4.1, we get a sharper lower bound $d(\mathcal{C}_0, \mathcal{C}_1) I(p, q) < \log N(B_{z_0}, \sqrt{d(z_0, \mathcal{C}_1)})$. Now we apply the general theorems for the lower bound to Examples 1.1 and 1.2. By (4.1) in Theorem 4.1, a key quantity for the lower bound is the packing number $N(B_{z_0}, \sqrt{d(z_0, \mathcal{C}_1)})$. The following proposition gives concrete results for the two examples.

**Proposition 4.3.** The packing number for Example 1.1 is $N(B_{z_0}, \sqrt{d(z_0, \mathcal{C}_1)}) = m$, and the packing number for Example 1.2 is $N(B_{z_0}, \sqrt{d(z_0, \mathcal{C}_1)}) = N(B_{z_0}, 0) = 1$.

Applying Proposition 4.3 to Theorem 4.1 and Theorem 4.2, we provide the lower bound for Examples 1.1 and 1.2 in Corollary 4.4 and Corollary 4.5 respectively.

**Corollary 4.4.** For $\mathcal{C}_0$ and $\mathcal{C}_1$ in (2.1), if $1/\rho_n = o(n^{1-c_2})$ for some constant $c_2 > 0$, $p < 1 - \delta$ for some constant $\delta > 0$ and $\limsup_{n \to \infty} 2nI(p, q)/(K \log m) < 1$, then $\liminf_{n \to \infty} r(\mathcal{C}_0, \mathcal{C}_1) \geq 1/2$.

**Corollary 4.5.** For $\mathcal{C}_0$ and $\mathcal{C}_1$ in (2.2), if $np \to \infty$, $0 < q < p \leq 1 - \delta$ for some constant $\delta > 0$ and $\limsup_{n \to \infty} nI(p, q) < c$ for some sufficiently small constant $c > 0$, then $\liminf_{n \to \infty} r(\mathcal{C}_0, \mathcal{C}_1) \geq 1/2$.

## 5 Numerical Results on Synthetic Data

We conduct the shadowing bootstrap on Examples 1.1 and 1.3. We test both hypotheses at the significance level $\alpha = 0.05$, and consider $K = 2$ and $n = 200, 600, 1000$. The connection probabilities are set to be $p = (1+\Delta)\rho_n$ and $q = (1-\Delta)\rho_n$, where $\rho_n = (n/K)^{-0.3}$ and $\Delta$ is the parameter controlling the difference between $p$ and $q$. We set $\Delta$ from 0 to 0.8. The maximum likelihood estimator is initialized by the singular value decomposition estimator to boost computation. For Example 1.1, we set $m = \lceil (n/K)^{\delta}/2 \rceil$ for $\delta = 0.3, 0.5, 0.7$ to explore the influence of $m$ on type-I and II errors.

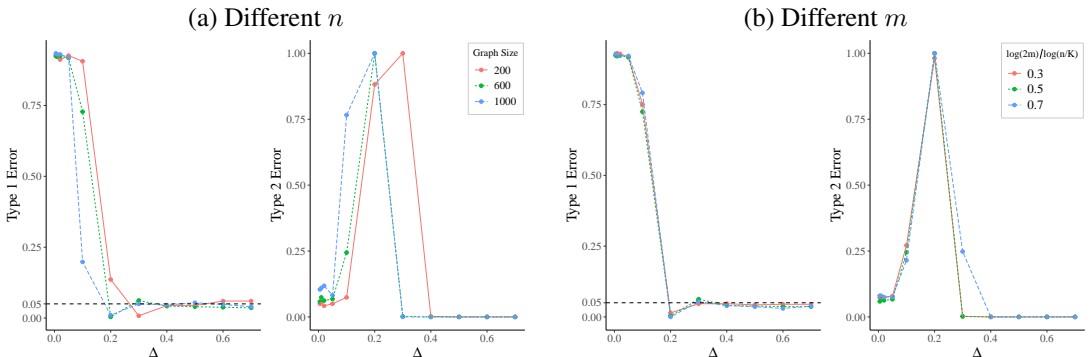

Figure 4: Type-I and type-II errors for Example 1.1. Panel (a) shows the results for different graph sizes $n = 200, 600, 1000$ with $m$ set as $\lceil (n/K)^{0.5}/2 \rceil$. Panel (b) shows the results $m = \lceil (n/K)^{\delta}/2 \rceil$ with $\delta = 0.3, 0.5, 0.7$ and the graph size $n = 600$.

In Figure 4, we show how the type-I and type-II errors vary with $\Delta$, $n$ and $m$ for Example 1.1 (via 500 Monte Carlos). As $\Delta$ increases, the type-I error converges to the nominal level 0.05, which shows that our method is honest. Type-II error is small when $\Delta$ is around zero as the test will always reject the null when the signal strength is too small, while it increases drastically as the type-I error drops to 0. When $\Delta$ is large enough, the type-II error converges to 0, showing that our test is powerful. In Figure 4(b), we can see that the type-I and type-II errors for different $m$'s converge similarly as $\Delta$ increases. This is consistent with Corollary 3.4 as the scaling condition is irrelevant to $m$. Simulation results for Example 1.3 are shown in Figure 5. The type 1 and type 2 error rates vary similarly as in Example 1.1. In summary, the type-I and type-II errors will not converge to the nominal levels until the metric product $d(\mathcal{C}_0, \mathcal{C}_1)I(p,q)$ is approximately larger than 10, which matches the theoretical lower bound (please refer to Tables 2 and 3 in the appendix).

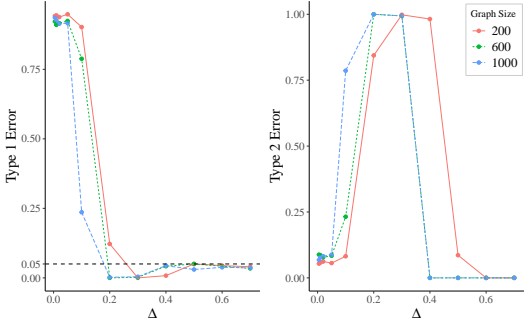

Figure 5: Type-I and type-II errors for Example 1.3 as $\Delta$ varies in $[0, 0.8]$ and the graph size $n = 200, 600, 1000$.

## 6 Conclusion

We propose a likelihood ratio test for the community property test with uncertainty quantification, implemented via a "shadowing bootstrap" method by utilizing the symmetry of $\mathcal{C}_0$ and $\mathcal{C}_1$. We show that our test is generally honest and powerful as long as $\mathcal{C}_0$ and $\mathcal{C}_1$ are symmetric. We also prove the minimax lower bound of the general community property test.

ACKNOWLEDGMENTS

The authors are grateful for the support of NSF DMS1916211, NIH/NCI R35 CA220523, NIH R01 ES32418 and U01CA209414.

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

# A    DISCUSSION AND REMARKS ON MAIN RESULTS

In this section, we provide some remarks on the results in the main text to offer insight into our settings that differ from classical hypothesis testing.

## A.1    FORM OF LIKELIHOOD RATIO STATISTIC

In this paper, our proposed likelihood ratio statistic (LRT) in (2.3) differs from that in the classical hypothesis testing:

$$\log \frac{\sup_{z \in \mathcal{C}_0} f(\mathbf{A}; z, p, q)}{\sup_{z \in \mathcal{C}_0 \cup \mathcal{C}_1} f(\mathbf{A}; z, p, q)}, \tag{A.1}$$

where on the numerator we take the supremum over $\mathcal{C}_1$ instead of $\mathcal{C}_0$. Here we adopt a different form of LRT to reduce the number of potential candidates in the alternative. More specifically, the traditional LRT in classical hypothesis testing is for the continuous parameter space, whereas when the null and the alternative are both combinatorial, the LRT will have a fundamentally different behavior due to the discrete optimization over an exponential number of candidates. In the numerator, we replace $\sup_{z \in \mathcal{C}_0}$ by $\sup_{z \in \mathcal{C}_1}$ such that under the null, the number of potential candidates on the numerator can be significantly reduced to the supremum over the boundary $B_{z_0}$, and we can take advantage of the symmetric community property to characterize the null distribution of the LRT, which is not feasible when the numerator is over $\mathcal{C}_0$.

## A.2    PLUG-IN ESTIMATORS FOR $p$ AND $q$

In (2.10), we estimate the connection probabilities $p$ and $q$ from (2.9) by taking the supremum over $z \in \mathcal{C}_0 \cup \mathcal{C}_1$. Here we apply $\widehat{p}$ and $\widehat{q}$ obtained from $\sup_{z \in \mathcal{C}_0 \cup \mathcal{C}_1}$ for both $\mathcal{C}_0 \cup \mathcal{C}_1$ and $\mathcal{C}_1$ in (2.10) because $p$ and $q$ here act like nuisance parameters, and our inferential interest lies in the combinatorial parameter space of the community assignments ($\mathcal{C}_0$ and $\mathcal{C}_1$) rather than the continuous parameter space of $p$ and $q$. Differentiation of $p$ and $q$ between $\mathcal{C}_0$ and $\mathcal{C}_1$ will have a minor contribution to the test statistic. So long as their plug-in estimators are consistent (which is the case for $\sup_{z \in \mathcal{C}_0 \cup \mathcal{C}_1}$), we will have a valid test statistic. Hence we also use $\widehat{p}$ and $\widehat{q}$ obtained from $\sup_{z \in \mathcal{C}_0 \cup \mathcal{C}_1}$ for $\mathcal{C}_1$ to improve computational efficiency.

## A.3    METRIC FOR PACKING NUMBER

In Definition 4.1, we define the packing number based upon the combinatorial metric $|\mathcal{E}_{1,2}(z_0, z_j) \cap \mathcal{E}_{1,2}(z_0, z_k)|$, with $\mathcal{E}_{1,2}(z_0, z_k)$ being the union of misaligned edge sets $\mathcal{E}_1(z_0, z_k)$ and $\mathcal{E}_2(z_0, z_k)$ defined in Definition 2.3. The intuition is that for any two assignments $z_j, z_k \in \mathcal{C}_1$, $|\mathcal{E}_{1,2}(z_0, z_j) \cap \mathcal{E}_{1,2}(z_0, z_k)|$ measures how many common misaligned edges they share. Intuitively, the more misaligned edges there are, the closer $z_j$ and $z_k$ should be, and hence the packing constraint $|\mathcal{E}_{1,2}(z_0, z_j) \cap \mathcal{E}_{1,2}(z_0, z_k)| \leq \epsilon$ is comparable to the constraint $\|\theta_i - \theta_j\| > \epsilon$ for the traditional definition of packing number. When $\epsilon = 0$, the constraint forces that $z_j$ and $z_k$ should be sufficiently different from each other. For example, in Example 1.1, for a null assignment $z_0 \in \mathcal{C}_0$, and $z_i, z_j \in B_{z_0}$, if $z_i$ and $z_j$ are obtained by relabeling nodes $i_1 \in [m]$ and $i_2 \in [m]$ of $z_0$ respectively, then $|\mathcal{E}_{1,2}(z_0, z_j) \cap \mathcal{E}_{1,2}(z_0, z_k)| \geq 1$ since the edge $(i_1, i_2)$ will always be contained in the misaligned set $\mathcal{E}_{1,2}(z_0, z_j) \cap \mathcal{E}_{1,2}(z_0, z_k)$, which results in $N(B_{z_0}, 0) = 1$. On the other hand, if we relax the packing constraint by taking $\epsilon = \sqrt{d(z_0, \mathcal{C}_1)}$, more common misaligned edges are allowed and we will have $N(B_{z_0}, \sqrt{d(z_0, \mathcal{C}_1)}) = m$ for Example 1.1.

# B    GENERAL FRAMEWORK FOR UNEVEN COMMUNITY SIZES

In this section, we generalize our theory to the community property tests when the community sizes in $\mathcal{C}_0$ and $\mathcal{C}_1$ are not necessarily even, e.g., Example 1.3. For any $z \in \mathcal{C}_0 \cup \mathcal{C}_1$, denote the community size $n_k(z) = |\{z(i) = k \mid i \in [n]\}|$ for $k \in [K]$. Let

$$c_K = \max_{z \in \mathcal{C}_0 \cup \mathcal{C}_1} \max_{1 \leq k \leq K} |n_k(z) - n/K|. \tag{B.1}$$

When the community sizes are even, we have $c_K = 0$. In this section, we consider the cases when $c_K$ could be larger than zero. We will show that the shadowing bootstrap method in Section 2.4 can be applied to test the uneven community property as well. The information-theoretic lower bound is also similar to the one in Section 4.

### B.1 GENERAL SYMMETRIC COMMUNITY PROPERTIES

For the uneven community class, we still need some symmetry property for the assignments in $\mathcal{C}_0$ and $\mathcal{C}_1$. When community sizes are even, Definition 2.2 depicts the symmetry via the representative node set $\mathcal{N}$ and the representative assignment $\widetilde{z}$. However, for many community properties of interest, e.g., the community size test in Example 1.3, we cannot find such $\mathcal{N}$ and $\widetilde{z}$. In Example 1.3, we are interested in testing the community size and thus there is no representative nodes. See Figure 6 for illustration.

Therefore, we define the following generalized symmetric community property pair.

**Definition B.1** (Generalized symmetric community property pair). We say two disjoint community properties $\mathcal{C}_0$ and $\mathcal{C}_1$ is a *generalized symmetric property pair* if for any $z, z' \in \mathcal{C}_0$, there exist permutations $\sigma \in S_K$ and $\tau \in S_n$ such that

(1) $\tau \circ \sigma(z) := (\sigma(z(\tau(1))), \ldots, \sigma(z(\tau(n)))) = z'$ and

(2) $\mathcal{C}_1$ is also closed under such transform $\tau \circ \sigma$, i.e., for any $z'' \in \mathcal{C}_1$, $\tau \circ \sigma(z'') \in \mathcal{C}_1$.

Definition B.1 generalizes the concept of symmetric community property in Definition 2.2 via introducing the permutation transform. We can check that Examples 1.1 and 1.2 are still symmetric by Definition B.1. See Figure 6(a) for an example of choosing $\sigma$ and $\tau$. On the other hand, the community sizes properties

$$\mathcal{C}_0 = \{z \in [K]^n : \text{all community sizes } = n/K\} \text{ and } \mathcal{C}_1 = \mathcal{C}_0^c, \tag{B.2}$$

are also symmetric by Definition B.1 but not by Definition 2.2. See Figure 6(b) for illustration. In fact, the following proposition shows that Definition 2.2 is a special case of Definition B.1.

**Proposition B.1.** If $\mathcal{C}_0, \mathcal{C}_1 \subseteq \mathcal{K}^n$ satisfy Assumption 2.1, then $\mathcal{C}_0$ and $\mathcal{C}_1$ is a generalized symmetric property pair. Moreover, the property pairs in (2.1), (2.2) and (B.2) are generalized symmetric property pairs.

We defer the proof of the proposition to Appendix C.2. In Figure 6, we show how to choose concrete permutation transforms $\sigma$ and $\tau$ for Examples 1.1 and 1.3.

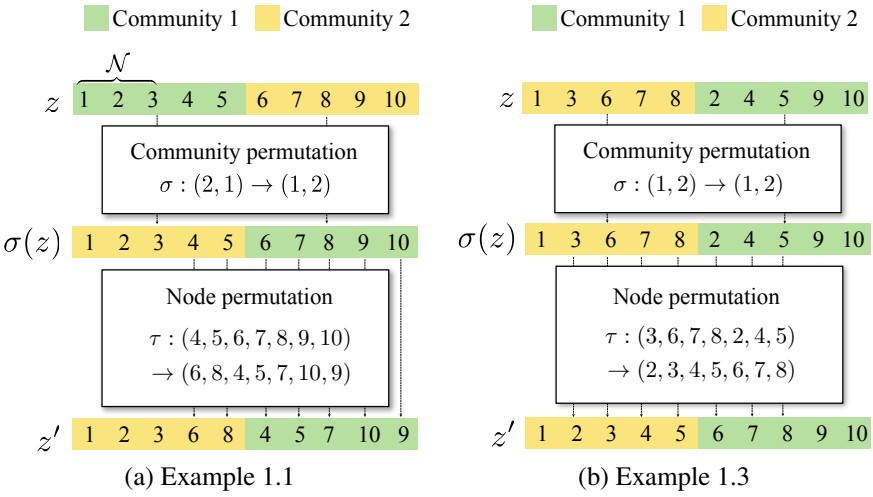

(a) Example 1.1          (b) Example 1.3

Figure 6: Permutation of null assignments in Example 1.1 and Example 1.3

## B.2 SHADOWING BOOTSTRAP FOR GENERAL CASE

We now generalize the testing method proposed in Section 2.4 to the uneven case. A key step is to generalize the boundary $B_{z_0}$ in Definition 2.5. Recall that for the even case, our insight is that the statistic $L$ in (2.6) taking the supremum over $\mathcal{C}_1$ is asymptotically equal to the $L_0$ in (2.7) taking the supremum over $B_{z^*}$, which is much smaller than $\mathcal{C}_1$. Similar insight applies to the uneven case using the following generalized definition of the boundary.

**Definition B.2.** For a given $z_0 \in \mathcal{C}_0$, we define the boundary centered at $z_0$ with radius $r$ as

$$B_{z_0}(r) = \{z \in \mathcal{C}_1 | d(z_0, z) \le r\}.$$

We illustrate the two types of boundary in Figure 7. From Figure 7(a), we can see that $B_{z_0} = B_{z_0}(d(\mathcal{C}_0, \mathcal{C}_1))$. Therefore, Definition B.2 is a generalization of Definition 2.5. For the uneven case, $L_0$ is no longer asymptotically equal to $L$. We need to enlarge $B_{z_0}$ to $B_{z_0}(r)$ for some $r > d(\mathcal{C}_0, \mathcal{C}_1)$ and modify the statistic $L_0$ in (2.7) by taking the supremum over $B_{z^*}(r)$.

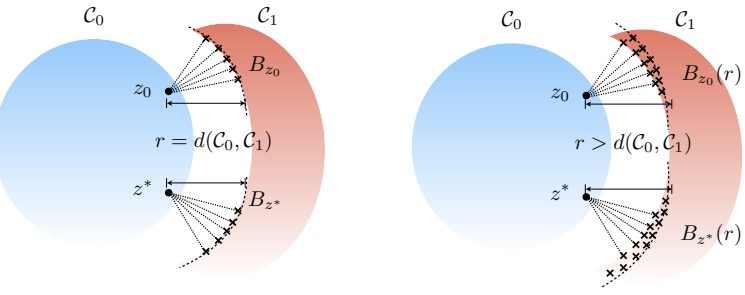

(a) Boundary $B_{z_0}$ in Definition 2.5      (b) Generalized boundary $B_{z_0}(r)$

Figure 7: The boundary $B_{z_0}$ defined previously for even cases is in essence a ball centered at $z_0$ with radius $r = d(\mathcal{C}_0, \mathcal{C}_1)$

In fact, we can still apply the shadowing bootstrap method in Section 2.4 to the uneven case. All procedures are exactly the same as in Section 2.4 except that we need to replace the bootstrap statistic $W_n$ in (2.11) by

$$W_n = \sup_{z \in B_{z_0}(r)} \sum_{1 \le i < j \le n} \left(\widehat{\mathbf{A}}_{ij} - \mathbb{E}_{\widehat{p}, \widehat{q}}(\widehat{\mathbf{A}}_{ij})\right)\left(\mathbb{1}[(i, j) \in \mathcal{E}_2(z_0, z)] - \mathbb{1}[(i, j) \in \mathcal{E}_1(z_0, z)]\right)e_{ij}, \quad \text{(B.3)}$$

where $r$ is a tuning parameter to be specified in the following theorem.

**Theorem B.2.** Suppose $\mathcal{C}_0$ and $\mathcal{C}_1$ are generalized symmetric community property pair and $c_K = O(1)$. Suppose $d(\mathcal{C}_0, \mathcal{C}_1) = o(n^{c_1})$ for some constant $c_1 < 2$, and $1/\rho_n = o(n^{1-c_2})$ for some constant $c_2 > 0$. We choose the radius $r$ in (B.3) as $r \ge r_K := d(\mathcal{C}_0, \mathcal{C}_1) + c_K^2 pK/(2(p-q))$ and $r = d(\mathcal{C}_0, \mathcal{C}_1) + O(1)$. If for any $z_0 \in \mathcal{C}_0$, we have $|B_{z_0}(r)| = O(n^{c_0})$ for some positive constant $c_0$, then

$$\lim_{n \to \infty} \sup_{z^* \in \mathcal{C}_0} \mathbb{P}(p_W \le \alpha) = \alpha \text{ and } \lim_{n \to \infty} \sup_{z^* \in \mathcal{C}_0} \mathbb{P}(\text{reject } H_0) = \alpha.$$

Moreover, if $d(\mathcal{C}_0, \mathcal{C}_1)I(p, q) = \Omega(n^\varepsilon)$ for some arbitrarily small constant $\varepsilon > 0$, we have

$$\lim_{n \to \infty} \inf_{z^* \in \mathcal{C}_1} \mathbb{P}(\text{reject } H_0) = 1.$$

We defer the proof of the theorem to Appendix D.2. The scaling assumptions in Theorem B.2 are similar to Theorem 3.2. The condition $|B_{z_0}(r)| = O(n^{c_0})$ for some $c_0 > 0$ is similar to Assumption 3.1. We need $c_K$ in (B.1) to be bounded to prevent a specific community from being too large. By the theorem, we need to choose $r \ge r_K := d(\mathcal{C}_0, \mathcal{C}_1) + c_K^2 pK/(2(p-q))$, while $p, q, c_K$ are unknown. In practice, we suggest to choose the radius as $r = d(\mathcal{C}_0, \mathcal{C}_1) + C\widehat{p}K/(\widehat{p} - \widehat{q})$ for some sufficiently large $C$. In fact, for many concrete examples, even though $r_K$ is unknown, we can directly construct $B_{z_0}(r_K)$. The following proposition shows how to construct $B_{z_0}(r_K)$ for

Examples 1.1-1.3. Moreover, it shows the conditions on $d(\mathcal{C}_0, \mathcal{C}_1)$ and $|B_{z_0}(r_K)|$ in Theorem B.2 are true for all these examples.

**Proposition B.3.** For any $z_0 \in \mathcal{C}_0$, $B_{z_0}(r_K)$ can be constructed as follows.

(1) Example 1.1: $B_{z_0}(r_K)$ is composed of all the assignments obtained from reassigning one node of any $z_0 \in \mathcal{C}_0$ in $[m]$ to a different community. See Figure 8(a) for an illustration. Moreover, we have $d(\mathcal{C}_0, \mathcal{C}_1) = n/K$ and $|B_{z_0}(r_K)| = m(K-1)$.

(2) Example 1.2: Suppose $m \wedge m' \le c_K$, $B_{z_0}(r_K)$ is composed of all the assignments obtained from reassigning nodes $m+1, \ldots, m+m'$ in any $z_0 \in \mathcal{C}_0$ collectively to a different community. Moreover, we have $d(\mathcal{C}_0, \mathcal{C}_1) = n(m \wedge m')/K$ and $|B_{z_0}(r_K)| = K-1$. Suppose $m \wedge m' > c_K$, $B_{z_0}(r_K)$ is composed of all the assignments obtained from exchanging label of nodes $m+1, \ldots, m+m'$ collectively with another $m'$ nodes from a different community for any $z_0 \in \mathcal{C}_0$. See Figure 8(b) for an illustration. Moreover, we have $d(\mathcal{C}_0, \mathcal{C}_1) = 2m \wedge m'(n/K - m \wedge m')$ and $|B_{z_0}(r_K)| = O(K(n/K)^{m \wedge m'})$.

(3) Example 1.3: For an arbitrary $z_0 \in \mathcal{C}_0$, $B_{z_0}(r_K)$ can be constructed by reassigning any node of $z_0$ to a different community. See Figure 8(c) for an illustration. Moreover, we have $d(\mathcal{C}_0, \mathcal{C}_1) = n/K$ and $|B_{z_0}(r_K)| = n(K-1)$.

We defer the proof to Appendix C.3. The construction of $B_{z_0}(r_K)$ is visualized in Figure 8. We also summarize the results in Table 1.

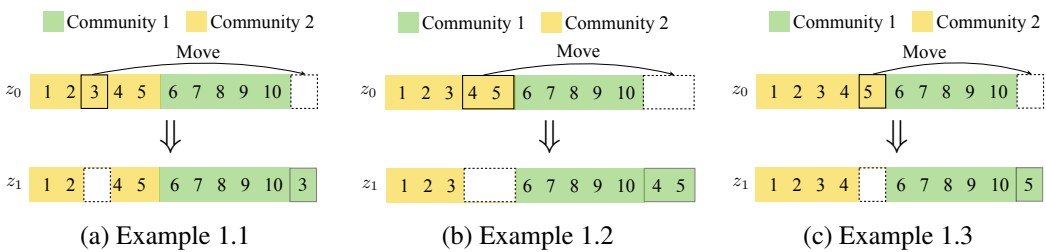

(a) Example 1.1        (b) Example 1.2        (c) Example 1.3

Figure 8: Construction of $B_{z_0}$ in Proposition B.3: (a) $\mathcal{C}_0$ is that nodes $\{1, 2, 3\}$ belong to the same community; (b) $\mathcal{C}_0$ is that the nodes set $\{1, 2, 3\}$ and $\{4, 5\}$ belong to the same community; (c) $\mathcal{C}_0$ is that community 1 and community 2 have an equal size of 5.

We, therefore, have the following corollary of Theorem B.2.

**Corollary B.4** (Examples 1.1 -1.3). Suppose $1/\rho_n = o(n^{1-c_2})$ for some constant $c_2 > 0$ and $c_K = O(1)$. We assume that $m \wedge m' = O(1)$ in Example 1.2. For Examples 1.1 -1.3, with $B_{z_0}(r_K)$ constructed in Proposition B.3 our test for the hypothesis $H_0 : z^* \in \mathcal{C}_0$ versus $H_1 : z^* \in \mathcal{C}_1$ is honest, i.e.,

$$\lim_{n \to \infty} \sup_{z^* \in \mathcal{C}_0} \mathbb{P}(\text{reject } H_0) = \alpha.$$

Moreover, if $I(p, q)n/K = \Omega(n^\varepsilon)$ for some small positive constant $\varepsilon$, we have

$$\lim_{n \to \infty} \sup_{z^* \in \mathcal{C}_1} \mathbb{P}(\text{reject } H_0) = 1.$$

### B.3 GENERAL LOWER BOUND

We can also generalize the information-theoretic lower bound in Theorem 4.1 and Theorem 4.2 to the uneven case. Similar to the even case, we need to define the packing number of $B_{z_0}(r)$, which follows the same definition of $N(B_{z_0}, \varepsilon)$ in Definition 4.1. We then have the lower bound of the general case as follows.

**Theorem B.5.** Suppose $1/\rho_n = o(n^{1-c_2})$ for some constant $c_2 > 0$, $p \le 1 - \delta$ for some constant $\delta > 0$ and $c_K = O(1)$. If there exists a $z_0 \in \mathcal{C}_0$ and some $r = d(z_0, \mathcal{C}_1) + O(1)$ such that

| | $d(z_0, \mathcal{C}_1)$ | $|B_{z_0}(r_K)|$ | $N(B_{z_0}(r_K), \sqrt{d(z_0, \mathcal{C}_1)})$ |
|---|---|---|---|
| Example 1.1 | $n/K$ | $m(K-1)$ | $m$ |
| Example 1.2 $m \wedge m' \leq c_K$ | $n(m \wedge m')/K$ | $K-1$ | $1$ |
| Example 1.2 $m \wedge m' > c_K$ | $2m \wedge m'(n/K - m \wedge m')$ | $O(K(n/K)^{m \wedge m'})$ | $1$ |
| Example 1.3 | $n/K$ | $n(K-1)$ | $n$ |

Table 1: Important values for general cases of Examples 1.1-1.3.

$\log N\big(B_{z_0}(r), \sqrt{d(z_0, \mathcal{C}_1)}\big) = O(\log n)$, and

$$\limsup_{n \to \infty} \frac{d(z_0, \mathcal{C}_1) I(p, q)}{\log N\big(B_{z_0}(r), \sqrt{d(z_0, \mathcal{C}_1)}\big)} < 1, \tag{B.4}$$

then $\liminf_{n \to \infty} r(\mathcal{C}_0, \mathcal{C}_1) \geq 1/2$.

**Remark B.1.** If we choose $r = d(z_0, \mathcal{C}_1)$, as $B_{z_0} = B_{z_0}(d(z_0, \mathcal{C}_1))$, (B.4) reduces to (4.1). The relaxed assumption on $r = d(z_0, \mathcal{C}_1) + O(1)$ can give us a better lower bound.

We can also generalize Theorem 4.2 to the following theorem.

**Theorem B.6.** Suppose $0 < q < p \leq 1 - \delta$ for some constant $\delta > 0$ and $\lim_{n \to \infty} d(\mathcal{C}_0, \mathcal{C}_1)p = \infty$. If one of the following conditions:

(1) $d(\mathcal{C}_0, \mathcal{C}_1)I(p, q) \leq c$ for some sufficiently small constant $c$;

(2) $\lim_{n \to \infty} d(\mathcal{C}_0, \mathcal{C}_1)I(p, q) = \infty$, but there exists a $z_0 \in \mathcal{C}_0$ and some $r = d(z_0, \mathcal{C}_1) + O(1)$ such that $\limsup_{n \to \infty} d(z_0, \mathcal{C}_1)I(p, q)/\log N(B_{z_0}(r), 0) < 1$,

is satisfied, then $\liminf_{n \to \infty} r(\mathcal{C}_0, \mathcal{C}_1) \geq 1/2$.

We defer the proof of the above two theorems to Appendix E.1.

To apply the general lower bound theorem to Examples 1.1-1.3, we need the following proposition on the packing number.

**Proposition B.7.** We have the packing number $N(B_{z_0}(r_K), \sqrt{d(z_0, \mathcal{C}_1)})$ for three examples as follows:

- Example 1.1: $N(B_{z_0}(r_K), \sqrt{d(z_0, \mathcal{C}_1)}) = m$;

- Example 1.2: $N(B_{z_0}(r_K), \sqrt{d(z_0, \mathcal{C}_1)}) = 1$;

- Example 1.3: $N(B_{z_0}(r_K), \sqrt{d(z_0, \mathcal{C}_1)}) = n$.

We defer the proof to Appendix C.4. The results is also summarized in Table 1.

Since $r_K = d(\mathcal{C}_0, \mathcal{C}_1) + c_K^2 pK/(2(p-q))$, where $c_K^2 pK/(2(p-q)) = O(1)$ and $d(\mathcal{C}_0, \mathcal{C}_1) = d(z_0, \mathcal{C}_1)$ by the symmetry of $\mathcal{C}_0, \mathcal{C}_1$, we have that $r_K = d(z_0, \mathcal{C}_1) + O(1)$. Applying Theorem B.5 and Proposition B.7, we have the following lower bound of same community test in Example 1.1.

**Corollary B.8.** For $\mathcal{C}_0$ and $\mathcal{C}_1$ defined in Example 1.1, if $1/\rho_n = o(n^{1-c_2})$ for some constant $c_2 > 0$, $p < 1 - \delta$ for some constant $\delta > 0$, $c_K = O(1)$ and

$$\limsup_{n \to \infty} nI(p, q)/(K \log m) < 1,$$

we have $\liminf_{n \to \infty} r(\mathcal{C}_0, \mathcal{C}_1) \geq 1/2$.

Applying Theorem B.6 and Proposition B.7, we have the following lower bound of same community test for groups in Example 1.2.

**Corollary B.9.** For $\mathcal{C}_0$ and $\mathcal{C}_1$ defined in (2.2), if $np \to \infty$, $0 < q < p < 1 - \delta$ for some $\delta > 0$ and

$$\limsup_{n \to \infty} nI(p, q) < c,$$

for some sufficiently small constant $c > 0$, we have $\liminf_{n \to \infty} r(\mathcal{C}_0, \mathcal{C}_1) \geq 1/2$.

For Example 1.3, applying Theorem B.5 and Proposition B.7, we have the following result.

**Corollary B.10.** For $\mathcal{C}_0$ and $\mathcal{C}_1$ defined in (B.2), if $1/\rho_n = o(n^{1-c_2})$ for some constant $c_2 > 0$, $p < 1 - \delta$ for some constant $\delta > 0$ and

$$\limsup_{n \to \infty} nI(p, q)/(K \log n) < 1,$$

we have $\liminf_{n \to \infty} r(\mathcal{C}_0, \mathcal{C}_1) \geq 1/2$.

## C  PROOFS OF COMMUNITY PROPERTIES

In this section, we mainly focus on the proofs concerning community properties, including the generalization of symmetric community property pairs from even to uneven cluster sizes, the size of the ball $B_{z_0}(r_K)$ in three examples, and the packing number of the ball in each case.

### C.1  PROOF OF PROPOSITION 4.3

**Example 1.1:**  In this case, for a given $z_0 \in \mathcal{C}_0$, we have derived the form of $B_{z_0}$. For any $z_i, z_j \in \mathcal{P}\big(B_{z_0}, \sqrt{d(z_0, \mathcal{C}_1)}\big)$, we know from Section 2.4 that they are transformed from $z_0$ by swapping one of the first $m$ nodes with another node from a different cluster. The node among the first $m$ to be swapped $s \in [m]$ cannot be the same for the two assignments, otherwise $|\mathcal{E}_{1,2}(z_0, z_i) \cap \mathcal{E}_{1,2}(z_0, z_j)| \geq |\mathcal{E}_1(z_0, z_i) \cap \mathcal{E}_1(z_0, z_j)| = n/K - 1 \gg \sqrt{d(z_0, \mathcal{C}_1)}$. Thus each $z \in \mathcal{P}\big(B_{z_0}, \sqrt{d(z_0, \mathcal{C}_1)}\big)$ corresponds to a different swapped node among the first $m$ nodes, and we have $N\big(B_{z_0}, \sqrt{d(z_0, \mathcal{C}_1)}\big) \leq m$. On the other hand, for the given assignment $z_0$, we can construct the following set $\{z_k\}_{k=1}^m$: we take a set of nodes $\mathcal{S} = \{s_1, s_2, ..., s_m\}$ from a cluster different from the cluster to which the first $m$ nodes of $z_0$ belong. Then for each $k$, we swap the cluster assignment of node $k$ with node $s_k$, $k = 1, ..., m$, and obtain the corresponding alternative assignment $z_k$. Then for any two alternative assignments $z_i$ and $z_j$ obtained this way, we have $|\mathcal{E}_{1,2}(z_0, z_i) \cap \mathcal{E}_{1,2}(z_0, z_j)| \leq 4$. Thus $N\big(B_{z_0}, \sqrt{d(z_0, \mathcal{C}_1)}\big) = m$.

**Example 1.2:**  For a given $z_0 \in \mathcal{C}_0$ and the corresponding boundary $B_{z_0}$, it can be perceived that $N(B_{z_0}, \sqrt{d(z_0, \mathcal{C}_1)}) = N(B_{z_0}, 0) = 1$, because any $z \in B_{z_0}$ involves swapping the set $\mathcal{S}_2$ so that $\forall z_i, z_j \in B_{z_0}, |\mathcal{E}_{1,2}(z_0, z_i) \cap \mathcal{E}_{1,2}(z_0, z_j)| \geq m \wedge m'(n/K - m \wedge m')$.

### C.2  PROOF OF PROPOSITION B.1

To prove that Definition 2.2 is a special case of Definition B.1 when the community size is even, it suffices for us to construct a concrete community label permutation $\sigma$ and node label permutation $\tau$ satisfying Definition B.1 based on $\mathcal{N}$ and $\tilde{z}$. Here we use Figure 6 to illustrate the construction. Given any $z, z' \in \mathcal{C}_0$, we first construct $\sigma$. Since $z_{\mathcal{N}} \simeq z'_{\mathcal{N}} \simeq \tilde{z}_{\mathcal{N}}$, by Definition 2.2, there must exist a $\sigma \in S_K$ mapping $z$ to $z'$ on the support $\mathcal{N}$, i.e., $\sigma(z_{\mathcal{N}}) = z'_{\mathcal{N}}$. For example, in Figure 6, we construct a $\sigma$ swapping communities 1 and 2. After matching the community labels, we now construct $\tau$ in order to transform $\sigma(z)$ to $z'$. Since the community size is even and $\sigma(z_{\mathcal{N}}) = z'_{\mathcal{N}}$, $\sigma(z)$ and $z'$ have equal cluster sizes on the support of $\mathcal{N}^c$. Therefore, there exists $\tau \in S_n$ such that $\tau(\sigma(z)_{\mathcal{N}^c}) = z'_{\mathcal{N}^c}$ and $\tau(\sigma(z)_{\mathcal{N}}) = \sigma(z)_{\mathcal{N}} = z'_{\mathcal{N}}$. We can see the example of $\tau$ in Figure 6. Using $\sigma$ and $\tau$ constructed above, we can check that $\tau \circ \sigma(z) = z'$. We now check the last condition in Definition B.1. For any $z'' \in \mathcal{C}_1$, since $\tau$ is invariant on $\mathcal{N}$, we have $\tau \circ \sigma(z''_{\mathcal{N}}) = \sigma(z''_{\mathcal{N}}) \simeq z''_{\mathcal{N}}$. By Definition 2.2, the alternative community $\mathcal{C}_1$ is closed under permutation on the support of $\mathcal{N}$, we have $\tau \circ \sigma(z'') \in \mathcal{C}_1$. Therefore, we check that Definition 2.2 is a special case of Definition B.1.

Since the property pairs in (2.1) and (2.2) are symmetric property pairs, they are also generalized symmetric property pairs following the preceding arguments. As for the property pair in (B.2), we can see from Figure 6(b) that for any two assignments $z, z' \in \mathcal{C}_0$, since they have equal community

sizes, we can take $\sigma$ to be the identity map and there exists $\tau \in S_n$ such that $\tau(z) = z'$. Then for any $z'' \in \mathcal{C}_1$, since $\tau$ does not change the community sizes, we know that $\tau(z'')$ still have uneven community sizes and $\tau(z'') \in \mathcal{C}_1$. Therefore, by Definition B.1, the property pair in (B.2) is a generalized symmetric property pair.

## C.3 PROOF OF PROPOSITION B.3

**Example 1.1:** To construct $B_{z_0}(r_K)$, we need to find all the assignments in $\mathcal{C}_1$ whose distance from $z_0$ is no larger than $d(\mathcal{C}_0, \mathcal{C}_1)$ by an extra constant term. To construct assignments in $\mathcal{C}_1$ closest to $z_0$, we would pick one node in $[m]$ and reassign it to a different community (see Figure 8 (a)). Assignments constructed in such ways will satisfy $d(z_0, z_1) = d(\mathcal{C}_0, \mathcal{C}_1) = n/K$. If we make community changes to any other nodes on the basis of such construction, then $d(z_0, z_1)$ would increase by at least $n/K - 2$, which exceeds the constant level. Thus $B_{z_0}(r_K)$ consists of all assignments constructed by moving one node of $z_0$ in $[m]$ to a different cluster. Since we can pick $m$ nodes in total and reassign them to $K - 1$ different clusters, $|B_{z_0}(r_K)| = (K-1)m = O(m)$.
**Example 1.2:** For an arbitrary $z_0 \in \mathcal{C}_0$, without loss of generality, we assume that $m' \leq m$. Then when $m' \leq c_K$, to construct assignments in $\mathcal{C}_1$ that are closest to $z_0$, we need to reassign nodes $m + 1, \ldots, m + m'$ collectively to a different community (see Figure 8 (b)). Such constructed assignments have distance $d(z_0, z_1) = d(\mathcal{C}_0, \mathcal{C}_1) = m'n/K$. Similar to the previous example, any community changes to other nodes on the basis of such construction would result in an increase of $d(z_0, z_1)$ by at least $n/K - m' - 1$. Therefore, $B_{z_0}(r_K)$ consists of those assignments in $\mathcal{C}_1$ constructed by reassigning nodes $m + 1, \ldots, m + m'$. Since there are $K - 1$ other clusters to reassign in total, we have $|B_{z_0}(r_K)| = K - 1 = O(1)$. On the other hand, when $m' > c_K$, then we cannot reassign nodes $m + 1, \ldots, m + m'$ collectively without exchanging with other nodes, otherwise, the community size bound will be violated. Then $d(\mathcal{C}_0, \mathcal{C}_1)$ and $B_{z_0}(r_K)$ is exactly the same as the even case and the claim follows.
**Example 1.3:** As for the ball $B_{z_0}(r_K)$ for an arbitrary $z_0 \in \mathcal{C}_0$, to transform $z_0$ into an assignment $z_1 \in \mathcal{C}_1$, the simplest way is to reassign an arbitrary node to a different community, and $d(z_0, z_1) = d(\mathcal{C}_0, \mathcal{C}_1) = n/K \asymp n$. Further community changes will result in increasing in $d(z_0, z_1)$ that exceeds the constant level. Since we can obtain such $z_1$ by reassigning any one of the $n$ nodes into the other $K - 1$ clusters, we have $|B_{z_0}(r_K)| = n(K-1) = O(n)$.

## C.4 PROOF OF PROPOSITION B.7

The arguments for Example 1.1 and Example 1.2 are almost the same as in the even cases and are hence omitted.

**Example 1.3:** For a given $z_0 \in \mathcal{C}_0$, from previous discussion we can see that the ball $B_{z_0}(r)$ with $r = d(z_0, \mathcal{C}_1) + O(1)$ is composed of all the assignments that differ from $z_0$ by one mis-aligned node. For any $z_i, z_j \in \mathcal{P}(B_{z_0}(r), \sqrt{d(z_0, \mathcal{C}_1)})$, the misaligned node $s$ cannot be the same, otherwise $|\mathcal{E}_{1,2}(z_0, z_i) \cap \mathcal{E}_{1,2}(z_0, z_j)| \geq n/K \gg \sqrt{d(z_0, \mathcal{C}_1)}$. Thus we have $N(B_{z_0}(r), \sqrt{d(z_0, \mathcal{C}_1)}) \leq n$. Also since the set $\{z_k\}_{k=1}^n$ where each $z_k$ is obtained by reassigning the node $k$ into another cluster obviously satisfies the condition that $|\mathcal{E}_1(z_0, z_i) \cap \mathcal{E}_1(z_0, z_j)| + |\mathcal{E}_2(z_0, z_i) \cap \mathcal{E}_2(z_0, z_j)| \leq 1$, we have that $N(B_{z_0}(r), \sqrt{d(z_0, \mathcal{C}_1)}) = n$.

# D  PROOF OF INFERENCE RESULTS

In this section, we provide the proofs of the theorems on inference results. We will first prove Proposition 2.3 which implies that the p-value based on the maximal leading term $L_0$ can be estimated without knowing the true assignment, then we prove the main Theorem 3.2 using Proposition 2.3 along with other lemmas. The proof of the technical lemmas will be deferred to Section F.

In the following part of our paper, we use $c, C, c_1, c_2, C_1, C_2, \ldots$ to represent generic constants and their values may vary in different places.

## D.1  PROOF OF PROPOSITION 2.3

To prove Proposition 2.3, we need the following generalized version of Lemma 2.2 stated previously

**Lemma D.1** (Shadowing symmetry lemma). For a given $z \in \mathcal{C}_0$ and a given radius $r > 0$, we list the assignments in the ball $B_z(r)$ as $z_1, z_2, \ldots, z_{|B_z(r)|}$. Define a $|B_z(r)|$-dimensional vector $\boldsymbol{L}_z$ as

$$(\boldsymbol{L}_z)_k = g(p,q)\Big( \sum_{(i,j)\in\mathcal{E}_2(z,z_k)} \mathbf{A}_{ij} - \sum_{(i,j)\in\mathcal{E}_1(z,z_k)} \mathbf{A}_{ij} \Big), \quad \text{for } k = 1, 2, \ldots, |B_z(r)|.$$

Suppose $\mathcal{C}_0$ and $\mathcal{C}_1$ satisfy definition B.1, then for any $z_0, z_0' \in \mathcal{C}_0$, we have $|B_{z_0}(r)| = |B_{z_0'}(r)|$ and $\mathrm{Cov}(\boldsymbol{L}_{z_0})$ is equal to $\mathrm{Cov}(\boldsymbol{L}_{z_0'})$ up to permutation, i.e., there existing a permutation $\tau \in S_{|B_{z_0}(r)|}$ such that $\mathrm{Cov}(\boldsymbol{L}_{z_0})_{kl} = \mathrm{Cov}(\boldsymbol{L}_{z_0'})_{\tau(k)\tau(l)}$ for all $k, l = 1, \ldots, |B_{z_0}(r)|$.

We defer the proof of Lemma D.1 to Section F.1. Now we are ready to prove Proposition 2.3. In fact, the boundary in the definition of $L_0$ can be generalized to the ball $B_z(r)$ with $r \geq r_K := d(\mathcal{C}_0, \mathcal{C}_1) + c_K^2 pK/(2(p-q))$ and $r = d(\mathcal{C}_0, \mathcal{C}_1) + O(1)$. For the true assignment $z^* \in \mathcal{C}_0$, we have that

$$L_0 = \sup_{z_k \in B_{z^*}(r)} \left\{ g(p,q)\Big( \sum_{(i,j)\in\mathcal{E}_2(z^*,z_k)} \mathbf{A}_{ij} - \sum_{(i,j)\in\mathcal{E}_1(z^*,z_k)} \mathbf{A}_{ij} \Big) \right\}$$

$$= g(p,q) \sup_{z_k \in B_{z^*}(r)} \left\{ \sum_{i<j} \Big\{ \big(\mathbf{A}_{ij} - \mathbb{E}(\mathbf{A}_{ij})\big)\big(\mathbb{1}[(i,j)\in\mathcal{E}_2(z^*,z_k)] - \mathbb{1}[(i,j)\in\mathcal{E}_1(z^*,z_k)]\big) \Big\} \right\}$$

$$+ g(p,q)\mu_0 + \delta_n = g(p,q)\sigma_0 \sup_{k\in[|B_{z^*}(r)|]} \left\{ \frac{1}{\sigma_0} \sum_{i<j}(\boldsymbol{X}_{ij})_k \right\} + g(p,q)\mu_0 + \delta_n.$$

where the vector $\boldsymbol{X}_{ij} \in \mathbb{R}^{|B_{z^*}(r)|}$ and

$$(\boldsymbol{X}_{ij})_k = \big(\mathbf{A}_{ij} - \mathbb{E}(\mathbf{A}_{ij})\big)\big(\mathbb{1}[(i,j)\in\mathcal{E}_2(z^*,z_k)] - \mathbb{1}[(i,j)\in\mathcal{E}_1(z^*,z_k)]\big), \quad \delta_n = O(\rho_n),$$

$$\text{and} \quad \sigma_0 = \sqrt{d(\mathcal{C}_0, \mathcal{C}_1)\big(p(1-p) + q(1-q)\big)}, \quad \mu_0 = d(\mathcal{C}_0, \mathcal{C}_1)(q-p).$$

We can see that for different $(i,j)$, the vector $\boldsymbol{X}_{ij}$ are independent of each other. For a fixed $k \in [|B_{z^*}(r)|]$, when $(i,j) \notin \mathcal{E}_{1,2}(z^*,z_k)$, $(\boldsymbol{X}_{ij})_k = 0$. When $(i,j) \in \mathcal{E}_{1,2}(z^*,z_k)$, under the regime $1/\rho_n = o(n^{1-c_2})$ for some positive $c_2$, there exists $B_n = 1/\sqrt{\rho_n} = o(n^{(1-c_2)/2})$ such that $|(\boldsymbol{X}_{ij})_k/\sqrt{\rho_n}| \leq B_n$ and $B_n^2(\log 2d(C_0, C_1)|B_{z_0}(r)|)^7/n \leq n^{-c_2/2}$, where $d(\mathcal{C}_0, \mathcal{C}_1) = o(n^2)$. Therefore, following a very similar proof as Theorem 2.2 and Corollary 2.1 in Chernozhukov et al. (2013), we have

$$g(p,q) \sup_{k\in[|B_{z^*}(r)|]} \left\{ \sum_{i<j}(\boldsymbol{X}_{ij})_k/\sigma_0 \right\} \xrightarrow{d} \sup_{k\in[|B_{z^*}(r)|]} \widetilde{Z}_k,$$

where $\widetilde{Z} \sim N(0, \boldsymbol{\Sigma}_{z^*}/\sigma_0^2)$, and $\Sigma_{z^*} = \mathrm{Cov}(\boldsymbol{L}_{z^*})$. Therefore, we have that

$$\sup_{t \in \mathbb{R}} \left| \mathbb{P}(L_0 \le t) - \mathbb{P}(\sigma_0 \sup_{k \in [|B_{z^*}(r)|]} \widetilde{Z}_k + g(p,q)\mu_0 \le t) \right|$$

$$\le \sup_{t \in \mathbb{R}} \left| \mathbb{P}(L_0 \le t) - \mathbb{P}(\sigma_0 \sup_{k \in [|B_{z^*}(r)|]} \widetilde{Z}_k + g(p,q)\mu_0 + \delta_n \le t) \right|$$

$$+ \sup_{t \in \mathbb{R}} \left| \mathbb{P}(\sigma_0 \sup_{k \in [|B_{z^*}(r)|]} \widetilde{Z}_k + g(p,q)\mu_0 + \delta_n \le t) - \mathbb{P}(\sigma_0 \sup_{k \in [|B_{z^*}(r)|]} \widetilde{Z}_k + g(p,q)\mu_0 \le t) \right|$$

$$\le o(1) + \sup_{t \in \mathbb{R}} \left| \mathbb{P}\Big( \big| \sup_{k \in [|B_{z^*}(r)|]} \widetilde{Z}_k - (t - g(p,q)\mu_0)/\sigma_0 \big| \le \delta_n/\sigma_0 \Big) \right|.$$

We know that $\min_{k \in [|B_{z^*}(r)|]} \mathrm{Var}(\widetilde{Z}_k) = \Omega(g(p,q)^2) = \Omega(1)$, $\log|B_{z^*}(r)| = O(\log n)$ and $\delta_n/\sigma_0 = O(n^{-1/2})$. Then by Lemma 2.1 in Chernozhukov et al. (2013), we have

$$\sup_{t \in \mathbb{R}} \left| \mathbb{P}\Big( \big| \sup_{k \in [|B_{z^*}(r)|]} \widetilde{Z}_k - (t - g(p,q)\mu_0)/\sigma_0 \big| \le \delta_n/\sigma_0 \Big) \right|$$

$$\lesssim \frac{\delta_n}{\sigma_0} \left\{ \sqrt{2\log|B_{z^*}(r)|} + \sqrt{\min_{k \in [|B_{z^*}(r)|]} \mathrm{Var}(\widetilde{Z}_k)\sigma_0/\delta_n} \right\} \le n^{-1/4}.$$

And thus we have

$$\sup_{t \in \mathbb{R}} \left| \mathbb{P}(L_0 \le t) - \mathbb{P}(\sigma_0 \sup_{k \in [|B_{z^*}(r)|]} \widetilde{Z}_k + g(p,q)\mu_0 \le t) \right| = o(1).$$

Following the same procedure with $z^*$ replaced by $z_0$, we also have

$$\sup_{t \in \mathbb{R}} \left| \mathbb{P}(L_0(z_0) \le t) - \mathbb{P}\Big( \sigma_0 \sup_{k \in [|B_{z_0}(r)|]} (\widetilde{Z}')_k + g(p,q)\mu_0 \le t \Big) \right| = o(1),$$

where $\widetilde{Z}' \sim N(0, \boldsymbol{\Sigma}_{z_0}/\sigma_0^2)$, and $\boldsymbol{\Sigma}_{z_0} = \mathrm{Cov}(\boldsymbol{L}_{z_0})$. By Lemma D.1 we know that $\boldsymbol{\Sigma}_{z^*}$ and $\boldsymbol{\Sigma}_{z_0}$ are equal up to permutation. Therefore, the claim follows. We may also note that the validity of the proof does not depend on the values of $p, q$ as long as the regime is $1/\rho_n = o(n^{1-c_2})$ for some constant $c_2 > 0$, and thus the statement is also true for $\widehat{L}_0 := \sup_{z_k \in B_{z^*}(r)} \Big\{ g(\widehat{p}, \widehat{q}) \big( \sum_{(i,j) \in \mathcal{E}_2(z^*, z_k)} \mathbf{A}_{ij} - \sum_{(i,j) \in \mathcal{E}_1(z^*, z_k)} \mathbf{A}_{ij} \big) \Big\}$ with plugged-in estimators $\widehat{p}, \widehat{q}$.

### D.2 PROOF OF THEOREM 3.2

In fact, Proposition B.1 shows that the symmetric community property pairs defined in Section 2 are general symmetric property pairs under the general framework, and Theorem B.2 is a generalization of Theorem 3.2 under uneven cluster sizes. Thus we can just prove the more general Theorem B.2 and the proof will also apply to Theorem 3.2.

The proof of the main theorem requires the help of Proposition 2.3 and the following lemma that shows why the maximizer in the alternative assignment space can be restricted to the ball centered at the true assignment $z_0 \in \mathcal{C}_0$.

**Lemma D.2.** We denote $z^*$ as the true assignment, and $B_{z^*}(r_K)$ is the ball centered at $z^*$ with radius $r_K = d(z^*, \mathcal{C}_1) + \frac{pK}{2(p-q)}c_K^2$, $c_K = O(1)$. Under the same conditions of Theorem B.2, when $z^* \in \mathcal{C}_0$

$$\sup_{z \in \mathcal{C}_1} \log f(\mathbf{A}; z, \widehat{p}, \widehat{q}) = \sup_{z \in B_{z^*}(r_K)} \log f(\mathbf{A}; z, \widehat{p}, \widehat{q}) + O_P(\rho_n); \tag{D.1}$$

Moreover, for any true assignment $z^*$, we have

$$\sup_{z \in \mathcal{C}_0 \cup \mathcal{C}_1} \log f(\mathbf{A}; z, \widehat{p}, \widehat{q}) = \log f(\mathbf{A}; z^*, \widehat{p}, \widehat{q}) + O_P(\rho_n). \tag{D.2}$$

With help of this lemma, instead of taking the supremum over the entire assignment space $\mathcal{C}_1$, we are able to restrict the maximizer to a much smaller set $B_{z_0}(r_K)$ so that the Central Limit Theorem can be applied. Recall that the boundary $B_{z^*}$ defined in Section 2.3 is in essence a ball with radius $d(z^*, \mathcal{C}_1)$. We defer the proof of Lemma D.2 to Appendix F.2.

Now we are ready to present the proof of Theorem B.2:

We first define the $\alpha$ quantile of the $\widehat{\mathrm{LRT}}$ statistic. Let $C_W(\alpha)$ be the $1 - \alpha$ quantile of $W_n$ conditioning on $\widehat{\mathbf{A}}$ and $\mathbf{A}$, i.e., $\mathbb{P}(W_n \leq C_W(\alpha)|\widehat{\mathbf{A}}, \mathbf{A}) = 1 - \alpha$. We then estimate the quantile of LRT by

$$q_\alpha = g(\widehat{p}, \widehat{q})C_W(\alpha) + g(\widehat{p}, \widehat{q})\widehat{\mu}_0, \tag{D.3}$$

then it can be seen that the two events $\{\widehat{\mathrm{LRT}} \geq q_\alpha\}$ and $\{p_W \leq \alpha\}$ are equivalent. Therefore, it suffices to show that $\lim_{n \to \infty} \sup_{z^* \in \mathcal{C}_0} \mathbb{P}(\widehat{\mathrm{LRT}} \leq q_\alpha) = \alpha$. The proof is mainly composed of three parts. The first part is to briefly illustrate the derivation of $L_0$ as the leading term of the log-likelihood ratio, the second part is to control the error caused by plugging in the estimators of connection probabilities $\widehat{p}, \widehat{q}$, and the third part is to illustrate the multiplier bootstrap as a valid approximation of the LRT quantile.

### D.2.1 DERIVATION OF THE LEADING TERM FOR LRT

For a given true assignment $z^* \in \mathcal{C}_0$, by Lemma D.2 we have:

$$\begin{aligned}
\widehat{\mathrm{LRT}} &= \log \frac{\sup_{z \in \mathcal{C}_1} f(\mathbf{A}; z, \widehat{p}, \widehat{q})}{\sup_{z \in \mathcal{C}_0 \cup \mathcal{C}_1} f(\mathbf{A}; z, \widehat{p}, \widehat{q})} \\
&= \sup_{z \in \mathcal{C}_1} \log f(\mathbf{A}; z, \widehat{p}, \widehat{q}) - \log f(\mathbf{A}; z^*, \widehat{p}, \widehat{q}) + O_P(\rho_n) \\
&= \sup_{z_k \in B_{z^*}(r)} \Big( \log f(\mathbf{A}; z_k, \widehat{p}, \widehat{q}) - \log f(\mathbf{A}; z^*, \widehat{p}, \widehat{q}) \Big) + O_P(\rho_n),
\end{aligned}$$

where $r \geq r_K := d(\mathcal{C}_0, \mathcal{C}_1) + c_K^2 pK/(2(p-q))$ and $r = d(\mathcal{C}_0, \mathcal{C}_1) + O(1)$. In practice, due to the consistency of $\widehat{p}, \widehat{q}$, when we choose the radius $r = d(\mathcal{C}_0, \mathcal{C}_1) + C\widehat{p}K/(\widehat{p} - \widehat{q})$ for some sufficiently large $C$, we can make sure that the conditions on the radius is satisfied with probability $1 - o(1)$.

Thus we can see that the $\widehat{\mathrm{LRT}}$ is essentially the supremum of the log-likelihood difference between the true assignment $z^*$ and the alternative assignments in the ball $B_{z^*}(r)$. We further expand the log-likelihood terms and can write

$$\begin{aligned}
\widehat{\mathrm{LRT}} &= \sup_{z_k \in B_{z^*}(r)} \left\{ g(\widehat{p}, \widehat{q})\Big( \sum_{(i,j) \in \mathcal{E}_2(z^*, z_k)} \mathbf{A}_{ij} - \sum_{(i,j) \in \mathcal{E}_1(z^*, z_k)} \mathbf{A}_{ij}\Big) + \log\Big(\frac{1 - \widehat{q}}{1 - \widehat{p}}\Big)\Big(n_1(z^*, z_k) - n_2(z^*, z_k)\Big) \right\} \\
&\quad + O_P(\rho_n) \\
&= \widehat{L}_0 + \delta_n.
\end{aligned}$$

where $\delta_n = \sup_{z_k \in B_{z^*}(r)} \left\{ \log\big((1 - \widehat{q})/(1 - \widehat{p})\big)\big(n_1(z^*, z_k) - n_2(z^*, z_k)\big) \right\} + O_P(\rho_n) = O_P(\rho_n)$, and $\widehat{L}_0 = g(\widehat{p}, \widehat{q}) \sup_{z_k \in B_{z^*}(r)} \big(\sum_{(i,j) \in \mathcal{E}_2(z^*, z_k)} \mathbf{A}_{ij} - \sum_{(i,j) \in \mathcal{E}_1(z^*, z_k)} \mathbf{A}_{ij}\big)$. From Proposition 2.3 we have that $\lim_{n \to \infty} \sup_{t \in \mathbb{R}} |\mathbb{P}(\widehat{L}_0 < t) - \mathbb{P}(\widehat{L}_0(z_0) < t)| = 0$ for any $z_0 \in \mathcal{C}_0$. Therefore, it suffices for us to prove that $\mathbb{P}(\widehat{\mathrm{LRT}} \geq q_\alpha) = \alpha + o(1)$ for one given true assignment $z_0 \in \mathcal{C}_0$. Now we are ready to prove the validity of the multiplier bootstrap for estimating the quantile based on the leading term.

### D.2.2 Bounding of error caused by plugging in $\widehat{p}, \widehat{q}$

From previous sections, we know that

$$\widehat{L}_0(z_0) = g(\widehat{p}, \widehat{q})\sigma_0 \sup_{k \in [|B_{z^*}(r)|]} \left\{ \frac{1}{\sigma_0} \sum_{i<j} (\boldsymbol{X}_{ij})_k \right\} + g(\widehat{p}, \widehat{q})\mu_0 + O_P(\rho_n),$$

where $(\boldsymbol{X}_{ij})_k = \big(\mathbf{A}_{ij} - \mathbb{E}(\mathbf{A}_{ij})\big)\big(\mathbb{1}[(i,j) \in \mathcal{E}_2(z_0, z_k)] - \mathbb{1}[(i,j) \in \mathcal{E}_1(z_0, z_k)]\big)$. For any $z_0 \in \mathcal{C}_0$, we give the following notations:

$$T_0 = \sup_{k \in [|B_{z_0}(r)|]} \left\{ \frac{1}{\sigma_0} \sum_{i<j} (\boldsymbol{X}_{ij})_k \right\}, \quad \Xi_0 = \sup_{k \in [|B_{z_0}(r)|]} \left\{ \frac{1}{\sigma_0} \sum_{i<j} \{\xi_{ij}\}_k \right\}, \quad \Xi_0' = \sup_{k \in [|B_{z_0}(r)|]} \left\{ \frac{1}{\widehat{\sigma}_0} \sum_{i<j} \{\widehat{\xi}_{ij}\}_k \right\},$$

and denote

$$\widetilde{W}_n = W_n / \widehat{\sigma}_0 = \sup_{k \in [|B_{z_0}(r)|]} \left\{ \frac{1}{\widehat{\sigma}_0} \sum_{i<j} (\widehat{\boldsymbol{X}}_{ij})_k e_{ij} \right\},$$

where $(\widehat{\boldsymbol{X}}_{ij})_k = \big(\widehat{\mathbf{A}}_{ij} - \mathbb{E}_{\widehat{p}, \widehat{q}}(\widehat{\mathbf{A}}_{ij})\big)\big(\mathbb{1}[(i,j) \in \mathcal{E}_2(z_0, z_k)] - \mathbb{1}[(i,j) \in \mathcal{E}_1(z_0, z_k)]\big)$ and the adjacency matrix $\widehat{\boldsymbol{A}}$ is generated by $\widehat{p}, \widehat{q}$, and $\widehat{\sigma}_0 = \sqrt{d(\mathcal{C}_0, \mathcal{C}_1)\big(\widehat{p}(1-\widehat{p}) + \widehat{q}(1-\widehat{q})\big)}$. $\xi_{ij}$ and $\widehat{\xi}_{ij}$ are the independent mean zero Gaussian vectors with covariance matrix equal to that of $\boldsymbol{X}_{ij}$ and $\widehat{\boldsymbol{X}}_{ij}$ respectively ($\{\xi_{ij}\}_k = 0$ if $(i,j) \notin \mathcal{E}_{1,2}(z_0, z_k)$, and the same for $\{\widehat{\xi}_{ij}\}_k$). $\{e_{ij}\}_{i<j}$ are i.i.d standard Gaussian. By Corollary 2.1 in Chernozhukov et al. (2013), we have

$$\sup_{t \in \mathbb{R}} |\mathbb{P}(T_0 \leqslant t) - \mathbb{P}(\Xi_0 \leqslant t)| = o(1);$$

Also, by Lemma 3.2 and Corollary 3.1 of Chernozhukov et al. (2013) we have

$$\sup_{t \in \mathbb{R}} \left| \mathbb{P}\left(\widetilde{W}_n \leqslant t | \widehat{\boldsymbol{X}}_{ij}\right) - \mathbb{P}(\Xi_0' \leqslant t) \right| = o_P(1).$$

We let $\boldsymbol{\Sigma}^{\Xi_0}$ and $\boldsymbol{\Sigma}^{\Xi_0'}$ be the covariance matrix of the vectors $\left\{ \sum_{i<j} \{\xi_{ij}\}_k / \sigma_0 \right\}_k$ and $\left\{ \sum_{i<j} \{\widehat{\xi}_{ij}\}_k / \widehat{\sigma}_0 \right\}_k$ respectively. Thus for $k, l \in [|B_{z_0}(r)|]$ we have:

$$\begin{aligned}
\boldsymbol{\Sigma}_{k,l}^{\Xi_0} &= \frac{1}{\sigma_0^2} \operatorname{Cov}(\sum_{ij} \{\xi_{ij}\}_k, \sum_{ij} \{\xi_{ij}\}_l) \\
&= \frac{1}{\sigma_0^2} \operatorname{Cov}(\sum_{i<j} (\boldsymbol{X}_{ij})_k, \sum_{i<j} (\boldsymbol{X}_{ij})_l) \\
&= \frac{|\mathcal{E}_2(z_0, z_k) \cap \mathcal{E}_2(z_0, z_l)|q(1-q) + |\mathcal{E}_1(z_0, z_k) \cap \mathcal{E}_1(z_0, z_l)|p(1-p)}{d(\mathcal{C}_0, \mathcal{C}_1)\big(p(1-p) + q(1-q)\big)}.
\end{aligned}$$

Accordingly,

$$\begin{aligned}
\boldsymbol{\Sigma}_{k,l}^{\Xi_0'} &= \frac{1}{\widehat{\sigma}_0^2} \operatorname{Cov}(\sum_{i<j} (\widehat{\boldsymbol{X}}_{ij})_k, \sum_{i<j} (\widehat{\boldsymbol{X}}_{ij})_l) \\
&= \frac{|\mathcal{E}_2(z_0, z_k) \cap \mathcal{E}_2(z_0, z_l)|\widehat{q}(1-\widehat{q}) + |\mathcal{E}_1(z_0, z_k) \cap \mathcal{E}_1(z_0, z_l)|\widehat{p}(1-\widehat{p})}{d(\mathcal{C}_0, \mathcal{C}_1)\big(\widehat{p}(1-\widehat{p}) + \widehat{q}(1-\widehat{q})\big)} \\
&= \frac{|\mathcal{E}_2(z_0, z_k) \cap \mathcal{E}_2(z_0, z_l)|q(1-q) + |\mathcal{E}_1(z_0, z_k) \cap \mathcal{E}_1(z_0, z_l)|p(1-p) + O_P(d(\mathcal{C}_0, \mathcal{C}_1)\sqrt{\rho_n}/n)}{d(\mathcal{C}_0, \mathcal{C}_1)\big(p(1-p) + q(1-q)\big) + O_P(d(\mathcal{C}_0, \mathcal{C}_1)\sqrt{\rho_n}/n)}.
\end{aligned}$$

Then we have

$$\Delta_0 = \max_{k,l} |\boldsymbol{\Sigma}_{k,l}^{\Xi_0} - \boldsymbol{\Sigma}_{k,l}^{\Xi_0'}| \leq \left| \frac{O_P(d(\mathcal{C}_0, \mathcal{C}_1)\sqrt{\rho_n}/n)}{\widehat{\sigma}_0^2} \right| + \left| \boldsymbol{\Sigma}_{k,l}^{\Xi_0} \frac{O_P(d(\mathcal{C}_0, \mathcal{C}_1)\sqrt{\rho_n}/n)}{\widehat{\sigma}_0^2} \right| = O_P(\frac{1}{\sqrt{n^2 \rho_n}}).$$

Thus by Lemma 3.1 in Chernozhukov et al. (2013), there exists a constant $C$ such that

$$\sup_{t \in \mathbb{R}} |\mathbb{P}(\Xi_0 \leqslant t) - \mathbb{P}(\Xi_0' \leqslant t)| \leq C\Delta_0^{1/3} (1 \vee \log(|B_{z_0}(r)|/\Delta_0))^{2/3} = o_P(n^{-1/6 - c_2/12}).$$

and thus

$$\sup_{t \in \mathbb{R}} |\mathbb{P}(\Xi_0 \leqslant t) - \mathbb{P}(\Xi_0' \leqslant t)| = o_P(1),$$

and in turn we have

$$\sup_{t \in \mathbb{R}} \left| \mathbb{P}(T_0 \leqslant t) - \mathbb{P}\left(\widetilde{W}_n \leqslant t | \widehat{\mathbf{X}}_{ij}\right) \right| = o_P(1).$$

### D.2.3 VALIDITY OF MULTIPLIER BOOTSTRAP IN ESTIMATING LRT QUANTILE

Now recall that $C_{\widetilde{W}_n}(\alpha)$ is the $\alpha$ quantile of $\widetilde{W}_n$ conditional on $\widehat{\mathbf{X}}_{ij}$, and we would like to control the order of $C_{\widetilde{W}_n}(\alpha)$ in order to bound the error in estimating the quantile of $\widehat{\mathrm{LRT}}$. Give a constant $t > \sqrt{2c_0}$, we have

$$\mathbb{P}(\widetilde{W}_n \geqslant t\sqrt{\log n} | \widehat{\mathbf{X}}_{ij}) = \mathbb{P}(\Xi_0' \geqslant t\sqrt{\log n}) + o_P(1)$$

$$\leq \sum_{k \in [|B_{z_0}(r)|]} \mathbb{P}\left( \left\{ \frac{1}{\widehat{\sigma}_0} \sum_{i<j} \{\widehat{\xi}_{ij}\}_k \right\} \geqslant t\sqrt{\log n} \right) + o_P(1)$$

$$\lesssim |B_{z_0}(r)| e^{-\frac{t^2}{2} \log n} + o_P(1) = O_P\left( n^{c_0 - t^2/2} \right) + o_P(1) = o_P(1).$$

Thus we know that $C_{\widetilde{W}_n}(\alpha) = O_P(\sqrt{\log n})$. We know that $q_\alpha = g(\widehat{p}, \widehat{q})\widehat{\sigma}_0 C_{\widetilde{W}_n}(\alpha) + g(\widehat{p}, \widehat{q})\widehat{\mu}_0$, $\widehat{\mathrm{LRT}} = \widehat{L}_0 + \delta_n$ and also $\lim_{n \to \infty} \sup_{t \in \mathbb{R}} |\mathbb{P}(\widehat{L}_0 < t) - \mathbb{P}(\widehat{L}_0(z_0) < t)| = 0$. Therefore,

$$\mathbb{P}(\widehat{\mathrm{LRT}} \geq q_\alpha) = \mathbb{P}(\widehat{L}_0 + \delta_n \geq q_\alpha) = \mathbb{P}(\widehat{L}_0(z_0) + \delta_n \geq q_\alpha) + o(1)$$

$$= \mathbb{P}(g(\widehat{p}, \widehat{q})\sigma_0 T_0 + g(\widehat{p}, \widehat{q})\mu_0 + \delta_n \geq g(\widehat{p}, \widehat{q})\widehat{\sigma}_0 C_{\widetilde{W}_n}(\alpha) + g(\widehat{p}, \widehat{q})\widehat{\mu}_0) + o(1)$$

$$= \mathbb{P}\left( T_0 \geq \frac{\widehat{\sigma}_0}{\sigma_0} C_{\widetilde{W}_n}(\alpha) + \frac{\widehat{\mu}_0 - \mu_0}{\sigma_0} - \frac{\delta_n}{g(\widehat{p}, \widehat{q})\sigma_0} \right) + o(1).$$

We have that $|\widehat{\sigma}_0 - \sigma_0| = O_P(\sqrt{d(\mathcal{C}_0, \mathcal{C}_1)}/n)$ and $|\widehat{\mu}_0 - \mu_0| = O_P(d(\mathcal{C}_0, \mathcal{C}_1)\sqrt{\rho_n}/n)$. Therefore,

$$\mathbb{P}(\widehat{\mathrm{LRT}} \geq q_\alpha) = \mathbb{P}\left( T_0 \geq C_{\widetilde{W}_n}(\alpha) + \frac{C_1\sqrt{d(\mathcal{C}_0, \mathcal{C}_1)}}{\sigma_0 n} C_{\widetilde{W}_n}(\alpha) + \frac{C_2 d(\mathcal{C}_0, \mathcal{C}_1)\sqrt{\rho_n}}{\sigma_0 n} - \frac{\delta_n}{g(\widehat{p}, \widehat{q})\sigma_0} \right) + o(1)$$

$$= \mathbb{P}\left( T_0 \geq C_{\widetilde{W}_n}(\alpha) + C_1\sqrt{\log n}/(n^2\rho_n) + C_2\sqrt{d(\mathcal{C}_0, \mathcal{C}_1)}/n + C_3\sqrt{\frac{\rho_n}{d(\mathcal{C}_0, \mathcal{C}_1)}} \right) + o(1)$$

$$= \mathbb{P}(T_0 \geq C_{\widetilde{W}_n}(\alpha) + \Delta_n) + o(1),$$

where $\Delta_n = o_P(n^{-c})$ for some positive constant $c > 0$. Now from previous results we have

$$|\mathbb{P}(\widehat{\mathrm{LRT}} \geq q_\alpha) - \alpha| \leq |\mathbb{P}(T_0 \geq C_{\widetilde{W}_n}(\alpha) + \Delta_n) - \mathbb{P}(\widetilde{W}_n \geq C_{\widetilde{W}_n}(\alpha) + \Delta_n)|$$

$$+ |\mathbb{P}(\widetilde{W}_n \geq C_{\widetilde{W}_n}(\alpha) + \Delta_n) - \mathbb{P}(\widetilde{W}_n \geq C_{\widetilde{W}_n}(\alpha))| + o(1)$$

$$\leq \mathbb{P}(|\widetilde{W}_n - C_{\widetilde{W}_n}(\alpha)| \leq \Delta_n) + o_P(1).$$

Now we study the distribution of $\widetilde{W}_n$: if we denote $Y_k = \frac{1}{\widehat{\sigma}_0} \sum_{i<j} (\widehat{\mathbf{X}}_{ij})_k e_{ij}$, then $Y_k | \widehat{\mathbf{X}} \sim N(0, \sigma_k^2)$, where $\sigma_k^2 = \sum_{i<j} (\widehat{\mathbf{X}}_{ij})_k^2 / \widehat{\sigma}_0^2$, and $\sup_k |\mathbb{E}(\sigma_k^2) - 1| \leq |\widehat{\sigma}_0^2 / \sigma_0^2 - 1| + o_P(1) = o_P(1)$. Also, $|(\widehat{\mathbf{X}}_{ij})_k^2| < 1$. Under the event $\mathcal{A} = \{\widehat{p} = o(1)\} \cap \{\widehat{q} = o(1)\}$ with $\mathbb{P}(\mathcal{A}) = 1 - o(1)$, by Bernstein's inequality, we have

$$\mathbb{P}_{\widehat{\mathbf{X}}}(|\sigma_k^2 - \mathbb{E}_{\widehat{\mathbf{X}}}(\sigma_k^2)| > 1/2) \leq 2\exp\left( -\frac{\frac{1+1}{8}\widehat{\sigma}_0^4}{(\frac{1}{6} + 1)\widehat{\sigma}_0^2} \right) = 2\exp\left( -\frac{3\widehat{\sigma}_0^2}{14} \right),$$

where $\mathbb{P}_{\widehat{\mathbf{X}}}$ and $\mathbb{E}_{\widehat{\mathbf{X}}}$ denotes probability and expectation with $\widehat{p}$ and $\widehat{q}$ fixed and consider only the randomness of $\widehat{\mathbf{X}}$. Also

$$\mathbb{P}_{\widehat{\mathbf{X}}}(\min_k \sigma_k^2 < 1/2) \leq \sum_k \mathbb{P}_{\widehat{\mathbf{X}}}(|\sigma_k^2 - \mathbb{E}_{\widehat{\mathbf{X}}}(\sigma_k^2)| > 1/2)$$

$$= 2|B_{z_0}(r)| \exp\left(-\frac{3\widehat{\sigma}_0^2}{14}\right) = o_P(1),$$

where the last $o_P(1)$ term is due to the fact that $\widehat{\sigma}_0^2 = \Omega_P(n\rho_n) = \Omega_P(n^{c_2})$ and $|B_{z_0}(r)| = O(n^{c_0})$. Then by Lemma 2.1 in Chernozhukov et al. (2013), we have

$$\mathbb{P}(|\widetilde{W}_n - C_{\widetilde{W}_n}(\alpha)| \leq \Delta_n) = \mathbb{P}(|\max_k Y_k - C_{\widetilde{W}_n}(\alpha)| \leq \Delta_n) \leq \sup_{z \in \mathbb{R}} \mathbb{P}(|\max_k Y_k - z| \leq \Delta_n)$$

$$= O_P\left(\Delta_n\left\{\sqrt{2\log|B_{z_0}(r)|} + \sqrt{\log(\min_k \sigma_k^2/\Delta_n)}\right\}\right) = o_P(1),$$

and thus $\lim_{n\to\infty} \sup_{z^* \in \mathcal{C}_0} \mathbb{P}(\widehat{\mathrm{LRT}} \geq q_\alpha) = \alpha$.

As for the Type I error, from the preceding proof we see that $\mathbb{P}(\mathrm{LRT} \geq q_\alpha) = \alpha + o_P(1)$, and the convergence of the $o_P(1)$ term is independent of $z^* \in \mathcal{C}_0$ due to the symmetry of $\mathcal{C}_0$. Therefore, we have

$$\sup_{z^* \in \mathcal{C}_0} \mathbb{P}(\text{reject } H_0) = \sup_{z^* \in \mathcal{C}_0} \mathbb{P}(\mathrm{LRT} \geq q_\alpha) = \alpha + o_P(1),$$

and hence the claim follows. As for the Type II error, when the true assignment is $z^* \in \mathcal{C}_1$, by (D.2) in Lemma D.2, we have

$$\mathrm{LRT} = \log \frac{\sup_{z \in \mathcal{C}_1} f(\mathbf{A}; z, \widehat{p}, \widehat{q})}{\sup_{z \in \mathcal{C}_0 \cup \mathcal{C}_1} f(\mathbf{A}; z, \widehat{p}, \widehat{q})}$$

$$= \log \frac{\sup_{z \in \mathcal{C}_1} f(\mathbf{A}; z, \widehat{p}, \widehat{q})}{f(\mathbf{A}; z^*, \widehat{p}, \widehat{q})} + \log \frac{f(\mathbf{A}; z^*, \widehat{p}, \widehat{q})}{\sup_{z \in \mathcal{C}_0 \cup \mathcal{C}_1} f(\mathbf{A}; z, \widehat{p}, \widehat{q})} = O_P(\rho_n).$$

And since $\widehat{\sigma}_0 \asymp \sqrt{d(\mathcal{C}_0, \mathcal{C}_1)\widehat{p}}, \widehat{\mu}_0 \asymp -d(\mathcal{C}_0, \mathcal{C}_1)\widehat{p} = -\Omega_P(n^{c_2})$ and $C_{\widetilde{W}_n}(\alpha) = O_P(\sqrt{\log n})$, we have $q_\alpha = g(\widehat{p}, \widehat{q})\widehat{\sigma}_0 C_{\widetilde{W}_n}(\alpha) + g(\widehat{p}, \widehat{q})\widehat{\mu}_0 \to -\infty$. Since the convergence is independent of $z^*$, we have for any true assignment $z_1 \in \mathcal{C}_1$,

$$\inf_{z^* \in \mathcal{C}_1} \mathbb{P}(\text{reject } H_0) = 1 - \sup_{z^* \in \mathcal{C}_1} \mathbb{P}(\mathrm{LRT} \leq q_\alpha) = 1 - o_P(1).$$

## E  PROOF OF THEOREMS FOR THE LOWER BOUND

In this section, we will prove the theorems for the lower bound. Similar to the upper bound, since Theorem B.5 and Theorem B.6 are general versions of Theorem 4.1 and Theorem 4.2, we will only prove the general versions and the proof can be applied to Theorem 4.1 and Theorem 4.2, too. Also, the proof of Theorem B.5 is actually based on the proof of Theorem B.6 under a stronger regime. Therefore, we will prove the two theorems together: we will first prove Theorem B.6 under more general conditions, and then we will apply the proof of Theorem B.6 to the proof of Theorem B.5 under stronger conditions.

### E.1  PROOF OF THEOREM B.5 AND THEOREM B.6

The proof proceeds in the following order: we first prove the results under the two conditions of Theorem B.6, namely the proof of Theorem B.6 (1) and the proof of Theorem B.6 (2), then we provide the proof of Theorem B.5.

### E.1.1 PROOF OF THEOREM B.6 (1)

As for the minimax rate, we have:

$$r(\mathcal{C}_0, \mathcal{C}_1) = \min_{\psi} \left\{ \sup_{z \in \mathcal{C}_0} \mathbb{P}_z(\psi = 1) + \sup_{z \in \mathcal{C}_1} \mathbb{P}_z(\psi = 0) \right\}$$

$$\geq \min_{\psi} \left\{ \mathbb{P}_{z_0}(\psi = 1) + \mathbb{P}_{z_1}(\psi = 0) \right\}.$$

where $z_0$ and $z_1$ are fixed cluster assignments in $\mathcal{C}_0$ and $\mathcal{C}_1$ respectively. For a given adjacency matrix $\mathbf{A}$, we know that $\psi$ is a function of $\mathbf{A}$, and the only information of $\mathbf{A}$ relevant to classification of the true assignment is $\{\mathbf{A}_{ij}, (i,j) \in \mathcal{E}_1(z_0, z_1) \bigcup \mathcal{E}_2(z_0, z_1)\}$. Larger sizes of $\mathcal{E}_1(z_0, z_1)$ and $\mathcal{E}_2(z_0, z_1)$ will provide more information and lead to smaller type I and type II errors. Thus, the worst case is when the size of $\mathcal{E}_1(z_0, z_1)$ and $\mathcal{E}_2(z_0, z_1)$ obtains the infimum, i.e., $d(z_0, z_1) = n_1(z_0, z_1) \vee n_2(z_0, z_1) = d(\mathcal{C}_0, \mathcal{C}_1)$.

To obtain $\inf_{\psi} \left\{ \sup_{z \in \mathcal{C}_0} \mathbb{P}_z(\psi = 1) + \sup_{z \in \mathcal{C}_1} \mathbb{P}_z(\psi = 0) \right\}$, the optimal method $\widetilde{\psi}$ must be the mode of the posterior distribution. For the convenience of notations, we denote $L(z, \mathbf{A})$ as $f(\mathbf{A}; z, p, q)$, and $n_i$ as $n_i(z_0, z_1)$, $i = 1, 2$ for short:

$$L(z_0, \mathbf{A}) \propto p^{\sum_{(i,j) \in \mathcal{E}_1(z_0, z_1)} \mathbf{A}_{ij}} (1-p)^{n_1 - \sum_{(i,j) \in \mathcal{E}_1(z_0, z_1)} \mathbf{A}_{ij}} q^{\sum_{(i,j) \in \mathcal{E}_2(z_0, z_1)} \mathbf{A}_{ij}} (1-q)^{n_2 - \sum_{(i,j) \in \mathcal{E}_2(z_0, z_1)} \mathbf{A}_{ij}}$$

$$L(z_1, \mathbf{A}) \propto p^{\sum_{(i,j) \in \mathcal{E}_2(z_0, z_1)} \mathbf{A}_{ij}} (1-p)^{n_2 - \sum_{(i,j) \in \mathcal{E}_2(z_0, z_1)} \mathbf{A}_{ij}} q^{\sum_{(i,j) \in \mathcal{E}_1(z_0, z_1)} \mathbf{A}_{ij}} (1-q)^{n_1 - \sum_{(i,j) \in \mathcal{E}_1(z_0, z_1)} \mathbf{A}_{ij}}$$

and correspondingly,

$$\widetilde{\psi}(\mathbf{A}) = \begin{cases} 0, & \text{if } L(z_0, \mathbf{A}) > L(z_1, \mathbf{A}); \\ 1, & \text{if } L(z_0, \mathbf{A}) \leq L(z_1, \mathbf{A}). \end{cases}$$

Then $\mathbb{P}_{z_0}(\widetilde{\psi} = 1) = \mathbb{P}_{z_0}(L(z_0, \mathbf{A}) \leq L(z_1, \mathbf{A}))$ and $\mathbb{P}_{z_1}(\widetilde{\psi} = 0) = \mathbb{P}_{z_1}(L(z_0, \mathbf{A}) > L(z_1, \mathbf{A}))$. Without loss of generality, we assume that $n_1(z_0, z_1) \geq n_2(z_0, z_1)$. Then, if we expend the size of $\mathcal{E}_2(z_0, z_1)$ to be the same as $\mathcal{E}_1(z_0, z_1)$, adding i.i.d entries $\{\mathbf{A}_{ij}, (i,j) \in \mathcal{E}_2^L(z_0, z_1) \backslash \mathcal{E}_2(z_0, z_1)\}$ conforming to the same distribution as $\{\mathbf{A}_{ij}, (i,j) \in \mathcal{E}_2(z_0, z_1)\}$, more information will be provided and the error rate will decrease, where $\mathcal{E}_2^L(z_0, z_1)$ denotes the set expended on $\mathcal{E}_2(z_0, z_1)$, and we have:

$$\widetilde{L}(z_0, \mathbf{A}) \propto p^{\sum_{(i,j) \in \mathcal{E}_1(z_0, z_1)} \mathbf{A}_{ij}} (1-p)^{n_1 - \sum_{(i,j) \in \mathcal{E}_1(z_0, z_1)} \mathbf{A}_{ij}} q^{\sum_{(i,j) \in \mathcal{E}_2^L(z_0, z_1)} \mathbf{A}_{ij}} (1-q)^{n_1 - \sum_{(i,j) \in \mathcal{E}_2^L(z_0, z_1)} \mathbf{A}_{ij}}$$

$$\widetilde{L}(z_1, \mathbf{A}) \propto p^{\sum_{(i,j) \in \mathcal{E}_2^L(z_0, z_1)} \mathbf{A}_{ij}} (1-p)^{n_1 - \sum_{(i,j) \in \mathcal{E}_2^L(z_0, z_1)} \mathbf{A}_{ij}} q^{\sum_{(i,j) \in \mathcal{E}_1(z_0, z_1)} \mathbf{A}_{ij}} (1-q)^{n_1 - \sum_{(i,j) \in \mathcal{E}_1(z_0, z_1)} \mathbf{A}_{ij}}$$

Thus we can obtain a lower bound on the minimax rate:

$$r(\widetilde{\psi}) = \mathbb{P}_{z_0}(L(z_0, \mathbf{A}) \leq L(z_1, \mathbf{A})) + \mathbb{P}_{z_1}(L(z_0, \mathbf{A}) > L(z_1, \mathbf{A}))$$

$$\geq \mathbb{P}_{z_0}(\widetilde{L}(z_0, \mathbf{A}) \leq \widetilde{L}(z_1, \mathbf{A})) + \mathbb{P}_{z_1}(\widetilde{L}(z_0, \mathbf{A}) > \widetilde{L}(z_1, \mathbf{A}))$$

$$= \mathbb{P}_{z_0}\left( \sum_{(i,j) \in \mathcal{E}_1(z_0, z_1)} \mathbf{A}_{ij} \leq \sum_{(i,j) \in \mathcal{E}_2^L(z_0, z_1)} \mathbf{A}_{ij} \right) + \mathbb{P}_{z_1}\left( \sum_{(i,j) \in \mathcal{E}_1(z_0, z_1)} \mathbf{A}_{ij} > \sum_{(i,j) \in \mathcal{E}_2^L(z_0, z_1)} \mathbf{A}_{ij} \right)$$

$$\geq 2\mathbb{P}\left( \sum_{u=1}^{n_1} X^u \geq \sum_{u=1}^{n_1} Y^u \right).$$

where $\{X_u\} \overset{\text{i.i.d}}{\sim} \text{Ber}(q)$, $\{Y^u\} \overset{\text{i.i.d}}{\sim} \text{Ber}(p)$, and $\{X^u\}$ are independent to $\{Y^u\}$.

Now $n_1 = d(\mathcal{C}_0, \mathcal{C}_1)$, and both $p$ and $q$ change with $n_1$. We have $\mathbb{E}(|X^u - Y^u - \mathbb{E}[X^u - Y^u]|^3) \asymp p(1-q) + q(1-p)$. Since $0 < q < p < 1 - \delta$, we have $\delta p < p(1-q) + q(1-p) < 2p$. Thus $\mathbb{E}(|X^u - Y^u - \mathbb{E}[X^u - Y^u]|^3) \asymp p$. Similarly $\text{Var}(X^u - Y^u) \asymp p$. Thus,

$$\frac{\sum_{u=1}^{n_1} \mathbb{E}(|X^u - Y^u - \mathbb{E}[X^u - Y^u]|^3)}{\text{Var}\left( \sum_{u=1}^{n_1} \{X^u - Y^u\} \right)^{3/2}} \asymp \frac{n_1[p(1-q) + q(1-p)]}{n_1^{3/2}(q(1-q) + p(1-p))^{3/2}} \asymp \frac{1}{\sqrt{n_1 p}} \to 0$$

Therefore, by the Lyapunov's Central Limit Theorem and the independence of $\{X^u\}$ and $\{Y^u\}$, as $n \to \infty$, we have $\sum_{u=1}^{n_1} X^u - \sum_{u=1}^{n_1} Y^u \xrightarrow{d} N(n_1(q-p), n_1 q(1-q) + n_1 p(1-p))$. Therefore,

$$\mathbb{P}(\sum_{u=1}^{n_1} X^u \geq \sum_{u=1}^{n_1} Y^u) = \mathbb{P}(\sum_{u=1}^{n_1} X^u - \sum_{u=1}^{n_1} Y^u \geq 0) = 1 - \Phi(\frac{\sqrt{n_1}(p-q)}{\sqrt{p(1-p) + q(1-q)}}) + o(1).$$

When $\limsup\limits_{n \to \infty} n_1 I(p,q) = O(1)$, we can see that $p - q = o(1)$. We have

$$I(p,q) = -2\log\left(1 - \frac{(\sqrt{p} - \sqrt{q})^2 + (\sqrt{1-p} - \sqrt{1-q})^2}{2}\right)$$

$$= \left(\frac{(p-q)^2}{(\sqrt{p} + \sqrt{q})^2} + \frac{(p-q)^2}{(\sqrt{1-p} + \sqrt{1-q})^2}\right)(1 + o(1))$$

$$\geq \frac{\delta}{2} \frac{(p-q)^2}{p(1-p) + q(1-q)}(1 + o(1)).$$

Thus, if $\limsup\limits_{n \to \infty} n_1 I(p,q) \leq \delta\Phi^{-1}(3/4)^2/2$, namely, $\limsup\limits_{n \to \infty} \sqrt{n_1}(p - q)/\sqrt{p(1-p) + q(1-q)} \leq \limsup\limits_{n \to \infty} \sqrt{2n_1 I(p,q)/\delta} \leq \Phi^{-1}(3/4)$, we have

$$\mathbb{P}(\sum_{u=1}^{n_1} X^u \geq \sum_{u=1}^{n_1} Y^u) \geq 1 - \Phi(\sqrt{n_1}(p-q)/\sqrt{p(1-p) + q(1-q)}) \geq 1/4.$$

and

$$r(\mathcal{C}_0, \mathcal{C}_1) \geq 2(1 - \Phi(\sqrt{n_1}(p-q)/\sqrt{p(1-p) + q(1-q)})) \geq 1/2.$$

### E.1.2 PROOF OF THEOREM B.6 (2)

When $d(z_0, \mathcal{C}_1)I(p,q) \to \infty$, if there exists a $z_0 \in \mathcal{C}_0$ and some $r = d(z_0, \mathcal{C}_1) + O(1)$ such that $\limsup_{n \to \infty} d(\mathcal{C}_0, \mathcal{C}_1)I(p,q)/\log N(B_{z_0}(r), 0) < 1$, then we take a 0-packing $\mathcal{P}(B_{z_0}(r), 0)$ (denoted $\mathcal{P}(0)$ for short) of the ball $B_{z_0}(r)$, and we have:

$$r(\mathcal{C}_0, \mathcal{C}_1) = \min_\psi \left\{ \sup_{z \in \mathcal{C}_0} \mathbb{P}_z(\psi = 1) + \sup_{z \in \mathcal{C}_1} \mathbb{P}_z(\psi = 0) \right\} \geq \min_\psi \left\{ \mathbb{P}_{z_0}(\psi = 1) + \sup_{z \in \mathcal{P}(0)} \mathbb{P}_z(\psi = 0) \right\}$$

$$= \min_\psi \left\{ \sum_A \left( \mathbb{P}_{z_0}(\psi = 1 | \mathbf{A} = A)\mathbb{P}_{z_0}(\mathbf{A} = A) + \sup_{z \in \mathcal{P}(0)} \mathbb{P}_z(\psi = 0 | \mathbf{A} = A)\mathbb{P}_z(\mathbf{A} = A) \right) \right\}$$

$$= \min_\psi \left\{ \sum_A \left( \mathbb{1}(\psi(A) = 1)\mathbb{P}_{z_0}(\mathbf{A} = A) + \mathbb{1}(\psi(A) = 0) \sup_{z \in \mathcal{P}(0)} \mathbb{P}_z(\mathbf{A} = A) \right) \right\},$$

where the sum over A is the summation over all possible realizations of the adjacency matrix $\mathbf{A}$. Thus the optimal method $\widetilde{\psi}$ in this scenario should be:

$$\widetilde{\psi}(\mathbf{A}) = \begin{cases} 0, & \text{if } L(z_0, \mathbf{A} = A) \geq \sup_{z \in \mathcal{P}(0)} L(z, \mathbf{A} = A); \\ 1, & \text{if } L(z_0, \mathbf{A} = A) < \sup_{z \in \mathcal{P}(0)} L(z, \mathbf{A} = A). \end{cases}$$

and we have $L(z_0, \mathbf{A} = A) < \sup_{z \in \mathcal{P}(0)} L(z, \mathbf{A} = A)$

$$r(\mathcal{C}_0, \mathcal{C}_1) \geq \mathbb{P}_{z_0}(\widetilde{\psi} = 1) + \sup_{z \in \mathcal{P}(0)} \mathbb{P}_z(\widetilde{\psi} = 0) = \mathbb{P}_{z_0}\left(L(z_0, \mathbf{A} = A) < \sup_{z \in \mathcal{P}(0)} L(z, \mathbf{A} = A)\right)$$

$$+ \sup_{z \in \mathcal{P}(0)} \mathbb{P}_z\left(L(z_0, \mathbf{A} = A) \geq \sup_{z \in \mathcal{P}(0)} L(z, \mathbf{A} = A)\right)$$

$$= \mathbb{P}_{z_0}\left(\sup_{z \in \mathcal{P}(0)} \log L(z, \mathbf{A} = A) - \log L(z_0, \mathbf{A} = A) > 0\right)$$

$$+ \sup_{z \in \mathcal{P}(0)} \mathbb{P}_z\left(\sup_{z \in \mathcal{P}(0)} \log L(z, \mathbf{A} = A) - \log L(z_0, \mathbf{A} = A) \leq 0\right).$$

Similar with the case when $d(z_0, \mathcal{C}_1)I(p, q) = O(1)$, we can expand each $\mathcal{E}_2(z_0, z)$ to $\mathcal{E}_2^L(z_0, z)$ (or $\mathcal{E}_1(z_0, z)$ to $\mathcal{E}_1^L(z_0, z)$, we use the former notation for convenience) so that $\mathcal{E}_1(z_0, z)$ and $\mathcal{E}_2(z_0, z)$ are of equal sizes, and then we have

$$
\begin{aligned}
r(\mathcal{C}_0, \mathcal{C}_1) &\geq \mathbb{P}_{z_0}\bigg(\sup_{z \in \mathcal{P}(0)} \Big(\sum_{(i,j) \in \mathcal{E}_2^L(z_0, z)} \mathbf{A}_{ij} - \sum_{(i,j) \in \mathcal{E}_1(z_0, z)} \mathbf{A}_{ij}\Big) > 0\bigg) \\
&\quad + \sup_{z \in \mathcal{P}(0)} \mathbb{P}_z\bigg(\sup_{z \in \mathcal{P}(0)} \Big(\sum_{(i,j) \in \mathcal{E}_2^L(z_0, z)} \mathbf{A}_{ij} - \sum_{(i,j) \in \mathcal{E}_1(z_0, z)} \mathbf{A}_{ij}\Big) \leq 0\bigg) \\
&\geq \mathbb{P}_{z_0}\bigg(\sup_{z \in \mathcal{P}(0)} \Big(\sum_{(i,j) \in \mathcal{E}_2^L(z_0, z)} \mathbf{A}_{ij} - \sum_{(i,j) \in \mathcal{E}_1(z_0, z)} \mathbf{A}_{ij}\Big) > 0\bigg) \\
&= \mathbb{P}_{z_0}\bigg(\sup_{z \in \mathcal{P}(0)} \Big(\sum_{u=1}^{n_1(z_0, z)} X_z^u - \sum_{u=1}^{n_1(z_0, z)} Y_z^u\Big) > 0\bigg),
\end{aligned}
$$

where $\{X_z^u\} \overset{i.i.d}{\sim} \text{Ber}(q)$, $\{Y_z^u\} \overset{i.i.d}{\sim} \text{Ber}(p)$, $\{X_{z_i}^u\} \perp \{X_{z_j}^u\}, i \neq j$, $\{Y_{z_i}^u\} \perp \{Y_{z_j}^u\}, i \neq j$ and $\{X_{z_i}^u\} \perp \{Y_{z_j}^u\}, \forall i, j$. By Lemma 5.2 in Zhang & Zhou (2016), we know that there exists $\eta \to 0$ such that

$$
\mathbb{P}_{z_0}\Big(\sum_{u=1}^{n_1(z_0, z)} X_z^u - \sum_{u=1}^{n_1(z_0, z)} Y_z^u > 0\Big) \geq \exp\big(-(1+\eta)d(z_0, \mathcal{C}_1)I(p, q)\big).
$$

When $\limsup_{n \to \infty} d(z_0, \mathcal{C}_1)I(p, q)/\log|\mathcal{P}(0)| < 1$, for sufficiently large $n$ we have $(1 + \eta)d(z_0, \mathcal{C}_1)I(p, q) \leq \log|\mathcal{P}(0)|$, and since $x > 1 - (1/2)^x$ for $x > 0$, we have that for $n$ large enough

$$
\begin{aligned}
\mathbb{P}_{z_0}\Big(\sum_{u=1}^{n_1(z_0, z)} X_z^u - \sum_{u=1}^{n_1(z_0, z)} Y_z^u > 0\Big) &\geq \exp\big(-(1+\eta)d(z_0, \mathcal{C}_1)I(p, q)\big) \\
&\geq \exp\big(-\log|\mathcal{P}(0)|\big) = 1/|\mathcal{P}(0)| \geq 1 - (1/2)^{1/|\mathcal{P}(0)|},
\end{aligned}
$$

and thus

$$
\begin{aligned}
r(\mathcal{C}_0, \mathcal{C}_1) &\geq \mathbb{P}_{z_0}\bigg(\sup_{z \in \mathcal{P}(0)} \Big(\sum_{u=1}^{n_1(z_0, z)} X_z^u - \sum_{u=1}^{n_1(z_0, z)} Y_z^u\Big) > 0\bigg) = 1 - \mathbb{P}_{z_0}\bigg(\sup_{z \in \mathcal{P}(0)} \Big(\sum_{u=1}^{n_1(z_0, z)} X_z^u - \sum_{u=1}^{n_1(z_0, z)} Y_z^u\Big) \leq 0\bigg) \\
&= 1 - \Pi_{z \in \mathcal{P}(0)} \mathbb{P}_{z_0}\bigg(\sum_{u=1}^{n_1(z_0, z)} X_z^u - \sum_{u=1}^{n_1(z_0, z)} Y_z^u \leq 0\bigg) \geq 1 - \Big\{(1/2)^{1/|\mathcal{P}(0)|}\Big\}^{|\mathcal{P}(0)|} = 1/2.
\end{aligned}
$$

The statement is true for any 0-packing of the ball $B_{z_0}(r)$, and thus the statement follows.

### E.1.3 PROOF OF THEOREM B.5

Under the regime $1/\rho_n = o(n^{1-c_2})$, we take one $\sqrt{d(z_0, \mathcal{C}_1)}$-packing $\mathcal{P}(B_{z_0}(r), \sqrt{d(z_0, \mathcal{C}_1)})$ (denoted $\widetilde{\mathcal{P}}$ for short) of the ball $B_{z_0}(r)$, similar with the proof of Theorem 3.2, by Corollary 2.1 in Chernozhukov et al. (2013), we have:

$$
\begin{aligned}
r(\mathcal{C}_0, \mathcal{C}_1) &\geq \mathbb{P}_{z_0}\bigg(\sup_{z \in \widetilde{\mathcal{P}}} \Big(\sum_{u=1}^{n_1(z_0, z)} X_z^u - \sum_{u=1}^{n_1(z_0, z)} Y_z^u\Big) > 0\bigg) = \mathbb{P}_{z_0}\bigg(\sup_{z \in \widetilde{\mathcal{P}}} \Big(\sum_{u=1}^{d(z_0, \mathcal{C}_1)} X_z^u - \sum_{u=1}^{d(z_0, \mathcal{C}_1)} Y_z^u\Big) > \delta_n\bigg) \\
&= \mathbb{P}_{z_0}\bigg(\sup_{z \in \widetilde{\mathcal{P}}} \Big(\sum_{u=1}^{d(z_0, \mathcal{C}_1)} X_z^u - \sum_{u=1}^{d(z_0, \mathcal{C}_1)} Y_z^u + d(z_0, \mathcal{C}_1)(p - q)\Big) > d(z_0, \mathcal{C}_1)(p - q) + \delta_n\bigg) \\
&= \mathbb{P}_{z_0}\bigg(\sup_{z \in \widetilde{\mathcal{P}}} \xi_z > \frac{d(z_0, \mathcal{C}_1)(p - q)}{\sigma_d} + \delta_n/\sigma_d\bigg) + o(1).
\end{aligned}
$$

where $\delta_n = \sup_{z \in \widehat{\mathcal{P}}} \left( \sum_{u=1}^{d(z_0, \mathcal{C}_1)} X_z^u - \sum_{u=1}^{d(z_0, \mathcal{C}_1)} Y_z^u \right) - \sup_{z \in \widetilde{\mathcal{P}}} \left( \sum_{u=1}^{n_1(z_0, z)} X_z^u - \sum_{u=1}^{n_1(z_0, z)} Y_z^u \right) = O(1)$ and $\sigma_d = \sqrt{d(z_0, \mathcal{C}_1)(p(1-p) + q(1-q))}$, and $\{\xi_z\}_{z \in \widetilde{\mathcal{P}}}$ are standard Gaussian variables with the same covariance matrix as $\{(\sum_{u=1}^{d(z_0, \mathcal{C}_1)} X_z^u - \sum_{u=1}^{d(z_0, \mathcal{C}_1)} Y_z^u + d(z_0, \mathcal{C}_1)(p-q))/\sigma_d\}_{z \in \widetilde{\mathcal{P}}}$.

By Lemma 2.1 in Chernozhukov et al. (2013), combined with the fact that $d(z_0, \mathcal{C}_1) = \Omega_P(n)$ and $1/\rho_n = o(n^{1-c_2})$, we have that

$$\left| \mathbb{P}_{z_0} \left( \sup_{z \in \widetilde{\mathcal{P}}} \xi_z > \frac{d(z_0, \mathcal{C}_1)(p-q)}{\sigma_d} + \delta_n/\sigma_d \right) - \mathbb{P}_{z_0} \left( \sup_{z \in \widetilde{\mathcal{P}}} \xi_z > \frac{d(z_0, \mathcal{C}_1)(p-q)}{\sigma_d} \right) \right| \lesssim \frac{\delta_n}{\sigma_d} \sqrt{\log n} = o(1).$$

We let $\{\widetilde{X}_z^u\}_{u,z}$ be i.i.d Ber($q$) random variables and $\{\widetilde{Y}_z^u\}_{u,z}$ be i.i.d Ber($p$) random variables, and $\{\widetilde{X}_z^u\}_{u,z}$ and $\{\widetilde{Y}_z^u\}_{u,z}$ are independent of each other. Then for each $z \in \widetilde{\mathcal{P}}$, $\sum_{u=1}^{d(z_0, \mathcal{C}_1)} \widetilde{X}_z^u - \sum_{u=1}^{d(z_0, \mathcal{C}_1)} \widetilde{Y}_z^u + d(z_0, \mathcal{C}_1)(p-q)$ shares the same distribution with $\sum_{u=1}^{d(z_0, \mathcal{C}_1)} X_z^u - \sum_{u=1}^{d(z_0, \mathcal{C}_1)} Y_z^u + d(z_0, \mathcal{C}_1)(p-q)$. We let $\{\widetilde{\xi}_z\}_{z \in \widetilde{\mathcal{P}}}$ be the corresponding Gaussian analog of $\{(\sum_{u=1}^{d(z_0, \mathcal{C}_1)} \widetilde{X}_z^u - \sum_{u=1}^{d(z_0, \mathcal{C}_1)} \widetilde{Y}_z^u + d(z_0, \mathcal{C}_1)(p-q))/\sigma_d\}_{z \in \widetilde{\mathcal{P}}}$. Then we have:

$$| \text{Cov}(\xi_{z_i}, \xi_{z_j}) - \text{Cov}(\widetilde{\xi}_{z_i}, \widetilde{\xi}_{z_j}) | = \begin{cases} 0 & \text{if } i = j, \\ O(\frac{1}{\sqrt{d(z_0, \mathcal{C}_1)}}) & \text{if } i \neq j. \end{cases}$$

Thus by Lemma 3.1 in Chernozhukov et al. (2013), we have $\Delta_0 = O\left(1/\sqrt{d(z_0, \mathcal{C}_1)}\right)$, and

$$\sup_{t \in \mathbb{R}} \left| \mathbb{P}_{z_0} \left( \sup_{z \in \widetilde{\mathcal{P}}} \xi_z > t \right) - \mathbb{P}_{z_0} \left( \sup_{z \in \widetilde{\mathcal{P}}} \widetilde{\xi}_z > t \right) \right| \leq C \Delta_0^{1/3} (\log |\widetilde{\mathcal{P}}|/\Delta_0)^{2/3} = o(1),$$

and thus

$$\mathbb{P}_{z_0} \left( \sup_{z \in \widetilde{\mathcal{P}}} \left( \sum_{u=1}^{d(z_0, \mathcal{C}_1)} X_z^u - \sum_{u=1}^{d(z_0, \mathcal{C}_1)} Y_z^u + d(z_0, \mathcal{C}_1)(p-q) \right) > d(z_0, \mathcal{C}_1)(p-q) \right)$$

$$= \mathbb{P}_{z_0} \left( \sup_{z \in \widetilde{\mathcal{P}}} \left( \sum_{u=1}^{d(z_0, \mathcal{C}_1)} \widetilde{X}_z^u - \sum_{u=1}^{d(z_0, \mathcal{C}_1)} \widetilde{Y}_z^u + d(z_0, \mathcal{C}_1)(p-q) \right) > d(z_0, \mathcal{C}_1)(p-q) \right) + o(1).$$

Then similar with previous proof, we have when $\limsup \lim_{n \to \infty} d(z_0, \mathcal{C}_1) I(p,q)/\log |\widetilde{\mathcal{P}}| < 1$,

$$r(\mathcal{C}_0, \mathcal{C}_1) \geq \mathbb{P}_{z_0} \left( \sup_{z \in B(r_K)} \left( \sum_{u=1}^{n_1(z_0, z)} \widetilde{X}_z^u - \sum_{u=1}^{n_1(z_0, z)} \widetilde{Y}_z^u \right) > 0 \right) + o(1) \geq 1/2 + o(1).$$

Also the results hold for any $\sqrt{d(z_0, \mathcal{C}_1)}$-packing of the ball $B_{z_0}(r)$. Therefore, we proved the claim.

# F    PROOF OF TECHNICAL LEMMAS

Now we will provide proofs for the technical lemmas used for the proof of Theorem B.2.

## F.1    PROOF OF LEMMA 2.2

It suffices for us to prove Lemma D.1, the more general version of Lemma 2.2. Due to the structure of $\boldsymbol{L}_z$, it suffices for us to prove that the edge-wise distance between assignments is permutation-invariant.

For any given $z_0 \in \mathcal{C}_0$ and $z_1, z_1' \in \mathcal{C}_1$, we have:

$$
\begin{aligned}
n_1(z_0, z_1) &= \sum_{i<j, i,j \in [n]} \mathbb{1}\big(z_0(i) = z_0(j), z_1(i) \neq z_1(j)\big) \\
&= \sum_{i<j, i,j \in [n]} \mathbb{1}\big(\sigma(z_0(i)) = \sigma(z_0(j)), \sigma(z_1(i)) \neq \sigma(z_1(j))\big) \\
&= \sum_{\tau(i)<\tau(j), i,j \in [n]} \mathbb{1}\big(\tau \circ \sigma(z_0)(\tau(i)) = \tau \circ \sigma(z_0)(\tau(j)), \tau \circ \sigma(z_1)(\tau(i)) \neq \tau \circ \sigma(z_1)(\tau(j))\big) \\
&= n_1(\tau \circ \sigma(z_0), \tau \circ \sigma(z_1)).
\end{aligned}
$$

Then very similarly we have $n_2(z_0, z_1) = n_2\big(\tau \circ \sigma(z_0), \tau \circ \sigma(z_1)\big)$ and thus $d(z_0, z_1) = n_1(z_0, z_1) \vee n_2(z_0, z_1) = n_1\big(\tau \circ \sigma(z_0), \tau \circ \sigma(z_1)\big) \vee n_2\big(\tau \circ \sigma(z_0), \tau \circ \sigma(z_1)\big) = d\big(\tau \circ \sigma(z_0), \tau \circ \sigma(z_1)\big)$. This suggests that the permutation $\tau \circ \sigma$ does not change the distance between assignments. Also,

$$
|\mathcal{E}_1(z_0, z_1) \cap \mathcal{E}_1(z_0, z_1')| = \sum_{i<j, i,j \in [n]} \mathbb{1}\big(z_0(i) = z_0(j), z_1(i) \neq z_1(j), z_1'(i) \neq z_1'(j)\big)
$$

$$
= \sum_{i<j, i,j \in [n]} \mathbb{1}\big(\tau \circ \sigma(z_0)(i) = \tau \circ \sigma(z_0)(j), \tau \circ \sigma(z_1)(i) \neq \tau \circ \sigma(z_1')(j), \tau \circ \sigma(z_1')(i) \neq \tau \circ \sigma(z_1)(j)\big)
$$

$$
= \big|\mathcal{E}_1\big(\tau \circ \sigma(z_0), \tau \circ \sigma(z_1)\big) \cap \mathcal{E}_1\big(\tau \circ \sigma(z_0), \tau \circ \sigma(z_1')\big)\big|.
$$

And similarly,

$$
|\mathcal{E}_2(z_0, z_1) \cap \mathcal{E}_2(z_0, z_1')| = |\mathcal{E}_2(\tau \circ \sigma(z_0), \tau \circ \sigma(z_1)) \cap \mathcal{E}_2(\tau \circ \sigma(z_0), \tau \circ \sigma(z_1'))|.
$$

Thus the cardinality of the intersection of the sets $\mathcal{E}_i, i = 1, 2$ is also invariant under the permutation $\tau \circ \sigma$.

Now for any $z_0, z_0' \in \mathcal{C}_0$, if $\tau \circ \sigma(z_0) = z_0'$, and $d(z_0, z_1) = d(z_0, \mathcal{C}_1)$, from previous results we have $d(z_0', \tau \circ \sigma(z_1)) = d(z_0, \mathcal{C}_1)$. If there exists an assignment $z_1' \in \mathcal{C}_1$ such that $d(z_0', z_1') < d(z_0', \tau \circ \sigma(z_1))$, then $d(z_0, (\tau \circ \sigma)^{-1}(z_1')) = d(z_0', z_1') < d(z_0, \mathcal{C}_1)$ due to the fact that $\tau \circ \sigma$ is a one to one mapping. Since $z_0 = (\tau \circ \sigma)^{-1}(z_0')$, we know that $\mathcal{C}_1$ is closed under $(\tau \circ \sigma)^{-1}$ and $(\tau \circ \sigma)^{-1}(z_1') \in \mathcal{C}_1$. This is contradictory to the fact that $z_1 = \operatorname{argmin}_{z \in \mathcal{C}_1} d(z_0, z)$. Therefore, $d(z_0', \mathcal{C}_1) = d(z_0', \tau \circ \sigma(z_1)) = d(z_0, \mathcal{C}_1)$.

Similarly, if $z_1 \in B_{z_0}(r)$, then $\tau \circ \sigma(z_1) \in B_{z_0'}(r)$. If $z_1' \in B_{z_0'}(r)$, then $(\tau \circ \sigma)^{-1}(z_1') \in B_{z_0}(r)$. Therefore, $\tau \circ \sigma$ is a one to one mapping from $B_{z_0}(r)$ to $B_{z_0'}(r)$, and $|B_{z_0}(r)| = |B_{z_0'}(r)|$.

Now for a given radius $r$, we find the permutation $\tau \in S_{|B_{z_0'}(r)|}$ such that $\tau(z_i) = \tau \circ \sigma(z_i) = z_i'$ for $z_i \in B_{z_0}(r)$ and $z_i' \in B_{z_0'}(r)$.

When the true assignment is $z_0$, the $(k, l)$-th entry of the covariance matrix for the vector $\boldsymbol{L}_{z_0}$ can be expressed as

$$
\begin{aligned}
\operatorname{Cov}(\boldsymbol{L}_{z_0})_{kl} &= g(p,q)^2 \Big(|\mathcal{E}_2(z_0, z_k) \cap \mathcal{E}_2(z_0, z_l)|q(1-q) + |\mathcal{E}_1(z_0, z_k) \cap \mathcal{E}_1(z_0, z_l)|p(1-p)\Big) \\
&= g(p,q)^2 \Big(|\mathcal{E}_2(\tau \circ \sigma(z_0), \tau \circ \sigma(z_k)) \cap \mathcal{E}_2(\tau \circ \sigma(z_0), \tau \circ \sigma(z_l))|q(1-q) \\
&\qquad + |\mathcal{E}_1(\tau \circ \sigma(z_0), \tau \circ \sigma(z_k)) \cap \mathcal{E}_1(\tau \circ \sigma(z_0), \tau \circ \sigma(z_l))|p(1-p)\Big) \\
&= \operatorname{Cov}\Big(g(p,q)\big(\sum_{\mathcal{E}_2(\tau(z_0'), \tau(z_k'))} \mathbf{A}_{ij} - \sum_{\mathcal{E}_1(\tau(z_0'), \tau(z_k'))} \mathbf{A}_{ij}\big), g(p,q)\big(\sum_{\mathcal{E}_2(\tau(z_0'), \tau(z_l'))} \mathbf{A}_{ij} - \sum_{\mathcal{E}_1(\tau(z_0'), \tau(z_l'))} \mathbf{A}_{ij}\big)\Big) \\
&= \operatorname{Cov}(\boldsymbol{L}_{z_0'})_{\tau(k)\tau(l)}.
\end{aligned}
$$

Hence we finish the proof.

## F.2 PROOF OF LEMMA D.2

The proof mainly follows from Lemma 2.3 and Lemma 2.6 in Wang & Bickel (2017) with modifications for the function $F(\cdot)$ and $G(\cdot)$. We provide the sketch of proof as follows:
We first define the count statistics as proposed in Wang & Bickel (2017):

$$\mathbf{A}_{i,j} \,|\, (z(i) = a, z(j) = b) \sim \mathrm{Ber}\,(\mathbf{H}_{a,b})\,, i \neq j, a, b \in [K].$$

where $\mathbf{H}_{a,b} = p = \lambda_1 \rho_n$ if $a = b$, and $\mathbf{H}_{a,b} = q = \lambda_2 \rho_n$ if $a \neq b$. $\mathbf{H} = \rho_n \mathbf{S}$.

$$\boldsymbol{O}_{a,b}(z) = \sum_{i=1}^{n} \sum_{j \neq i} \mathbb{1}\,(z(i) = a, z(j) = b)\,\mathbf{A}_{ij},$$

$$L = \sum_{i=1}^{n} \sum_{j=i+1}^{n} \mathbf{A}_{ij}, \mu_n = n^2 \rho_n.$$

For two assignments $z, z'$, The confusion matrix is:

$$\boldsymbol{R}_{k,a}(z, z') = n^{-1} \sum_{i=1}^{n} \mathbb{1}\,(z(i) = k, z'(i) = a)\,.$$

By definition, we have $|n_k(z) - n/K| \leq c_K, \forall z \in \mathcal{C}_0 \cup \mathcal{C}_1, \forall k = 1, 2, ..., K$. We let $\widetilde{n}(z)$ denote the number of within-cluster edges, and assume

$$n_k(z) = n/K + a_k, |a_k| \leq c_K, k = 1, 2, ..., K,$$

$$\sum_{k=1}^{K} a_k = 0.$$

Then

$$\widetilde{n}(z) = \frac{\sum_{k=1}^{K}(n/K + a_k)^2 - n}{2} = \frac{n^2/K - n}{2} + \frac{\sum_{k=1}^{K} a_k^2}{2}$$

$$\leq \frac{n^2/K - n}{2} + Kc_K^2/2.$$

Therefore, $\forall z, z' \in \mathcal{C}_1$ we have $\widetilde{n}(z) + n_2(z, z') - n_1(z, z') = \widetilde{n}(z'), |n_2(z, z') - n_1(z, z')| = |\widetilde{n}(z) - \widetilde{n}(z')| \leq Kc_K^2/2$. Thus, we denote $z^*$ as the true assignment, and $\forall z \in \mathcal{C}_0 \cup \mathcal{C}_1$ we have

$$\log f(\mathbf{A}; z, \widehat{p}, \widehat{q}) = \frac{1}{2}\Big(\sum_{a,b=1}^{K} \boldsymbol{O}_{a,b}(z) \log \frac{\widehat{\mathbf{H}}_{a,b}}{1 - \widehat{\mathbf{H}}_{a,b}}\Big) + \widetilde{n}(z) \log(1 - \widehat{p}) + \big(n(n-1)/2 - \widetilde{n}(z)\big) \log(1 - \widehat{q})$$

$$= \frac{1}{2}\Big(\sum_{a,b=1}^{K} \boldsymbol{O}_{a,b}(z)\big(\log \widehat{\mathbf{S}}_{a,b} + \log \rho_n - \log(1 - \widehat{\mathbf{H}}_{a,b})\big)\Big) + C(z^*) + O_P(\rho_n)$$

$$= \frac{\mu_n}{2}\Big(\sum_{a,b=1}^{K} \frac{\boldsymbol{O}_{a,b}(z)}{\mu_n}\{\log \widehat{\mathbf{S}}_{a,b} + O_P(\rho_n)\}\Big) + \log \rho_n L + C(z^*) + O_P(\rho_n).$$

where $C(z^*) = \widetilde{n}(z^*) \log(1 - \widehat{p}) + \big(n(n-1)/2 - \widetilde{n}(z^*)\big) \log(1 - \widehat{q})$. We let $F(\boldsymbol{O}(z)/\mu_n) = \sum_{a,b=1}^{K} \frac{\boldsymbol{O}_{a,b}(z)}{\mu_n} \log \frac{\widehat{\mathbf{S}}_{a,b}}{1 - \widehat{\mathbf{H}}_{a,b}}$ and $F(\boldsymbol{R}\mathbf{S}\boldsymbol{R}^\top(z)) = \sum_{a,b=1}^{K}(\boldsymbol{R}\mathbf{S}\boldsymbol{R}^\top(z))_{a,b} \log \frac{\widehat{\mathbf{S}}_{a,b}}{1 - \widehat{\mathbf{H}}_{a,b}}$, where $\boldsymbol{R}(z) = \boldsymbol{R}(z, z^*)$ and $\boldsymbol{R}\mathbf{S}\boldsymbol{R}^\top(z) = \boldsymbol{R}(z, z^*)\mathbf{S}\boldsymbol{R}(z, z^*)^\top$. We denote $\widetilde{\mathcal{C}} \subseteq \mathcal{C}_0 \cup \mathcal{C}_1$ as some subset of assignments, and we let $\mathcal{V}_G$ denote the set of $z \in \widetilde{\mathcal{C}}$ that maximizes $F(\boldsymbol{R}\mathbf{S}\boldsymbol{R}^\top(z))$. Obviously $F(\cdot)$ is Lipschitz, for $\epsilon_n \to 0$ slowly,

$$\left| F(\boldsymbol{O}(z)/\mu_n) - F(\boldsymbol{R}\mathbf{S}\boldsymbol{R}^\top(z)) \right| \leq C \cdot \sum_{k,l} \left| \boldsymbol{O}_{k,l}(z)/\mu_n - \big(\boldsymbol{R}\mathbf{S}\boldsymbol{R}^\top(z)\big)_{k,l} \right| = O_P(\epsilon_n).$$

We choose some positive $\delta_n \to 0$ slowly enough such that $\delta_n/\epsilon_n \to \infty$. We take any $Z' \in \mathcal{V}_G$, then we define

$$J_{\delta_n} = \left\{ z \in [K]^n : F(\boldsymbol{RSR}^\top(z)) - F(\boldsymbol{RSR}^\top(Z')) < -\delta_n \right\}.$$

Then we have

$$\sum_{z \in J_{\delta_n}} e^{\log f(\mathbf{A}; z, \widehat{p}, \widehat{q})} \leq f(\mathbf{A}; Z', \widehat{p}, \widehat{q}) K^n e^{O_P(\mu_n \epsilon_n) - \mu_n \delta_n/2 + O_P(\rho_n)}$$

$$= f(\mathbf{A}; Z', \widehat{p}, \widehat{q}) o_P(1).$$

For $z \in \widetilde{\mathcal{C}} \backslash \{J_{\delta_n} \cup \mathcal{V}_G\}$, $|F(\boldsymbol{RSR}^\top(z)) - F(\boldsymbol{RSR}^\top(Z'))| \to 0$ and $\|\boldsymbol{RSR}^\top(z) - \boldsymbol{RSR}^\top(Z')\|_\infty \to 0$. Treating $\boldsymbol{R}(z)$ as a vector, choosing $z_\perp$ be such that $\boldsymbol{R}(z_\perp) := \min_{\boldsymbol{R}(z_0): z_0 \in \mathcal{V}_G} \|\boldsymbol{R}(z) - \boldsymbol{R}(z_0)\|_2$ for a given $z \in \widetilde{\mathcal{C}} \backslash \{J_{\delta_n} \cup \mathcal{V}_G\}$. Due to the consistency of $\widehat{p}, \widehat{q}$, the function $F(\cdot)$ is a linear function with constant coefficients. We know that with probability $1 - o(1)$:

$$\left. \frac{\partial F\left( (1-\epsilon)\boldsymbol{RSR}^\top(z_\perp) + \epsilon \boldsymbol{RSR}^\top(z) \right)}{\partial \epsilon} \right|_{\epsilon=0^+} < 0.$$

Given a matrix $A$, we denote the matrix maximum norm $\|A\|_\infty = \max_{jk} |A_{jk}|$. Letting $\bar{z} = \min_{\sigma(z)} |\sigma(z) - z_\perp|$, and $\boldsymbol{X}(z) = \boldsymbol{O}(z)/\mu_n - \boldsymbol{RSR}^\top(z)$, we have

$$\mathbb{P}\left( \max_{z \notin \mathcal{S}(z_\perp)} \|\boldsymbol{X}(\bar{z}) - \boldsymbol{X}(z_\perp)\|_\infty > \epsilon |\bar{z} - z_\perp|/n \right)$$

$$\leq \sum_{m=1}^n \mathbb{P}\left( \max_{z: z = \bar{z}, |\bar{z} - z_\perp| = m} \|\boldsymbol{X}(z) - \boldsymbol{X}(z_\perp)\|_\infty > \epsilon \frac{m}{n} \right)$$

$$\leq \sum_{m=1}^n 2K^K n^m K^{m+2} \exp\left( -C \frac{m\mu_n}{n} \right) \to 0.$$

where $\mathcal{S}(z) = \{\sigma(z) | \sigma \in S_K\}$. Since $\boldsymbol{RSR}^\top(\bar{z}) - \boldsymbol{RSR}^\top(z_\perp) = \Omega(|\bar{z} - z_\perp|)$, we have that

$$\frac{\boldsymbol{O}(\bar{z})}{\mu_n} - \frac{\boldsymbol{O}(z_\perp)}{\mu_n} = (1 + o_P(1))\left( \boldsymbol{RSR}^\top(\bar{z}) - \boldsymbol{RSR}^\top(z_\perp) \right).$$

And thus we probability $1 - o(1)$ uniform on all $z$, we have

$$F(\boldsymbol{O}(\bar{z})/\mu_n) < F(\boldsymbol{O}(z_\perp)/\mu_n).$$

In turn, we have

$$\log f(\mathbf{A}; z, \widehat{p}, \widehat{q}) \leq \log f(\mathbf{A}; z_\perp, \widehat{p}, \widehat{q}) + O_P(\rho_n).$$

Since from Lemma A.1 in Wang & Bickel (2017) the high probability is uniform on all assignments, we have that, with probability $1 - o(1)$, for any $z \in \widetilde{\mathcal{C}} \backslash \mathcal{V}_G$ we can find $z' \in \mathcal{V}_G$ such that $\log f(\mathbf{A}; z, p, q) = \log f(\mathbf{A}; z', p, q) + O_P(\rho_n)$, and therefore,

$$\sup_{z \in \widetilde{\mathcal{C}}} \log f(\mathbf{A}; z, \widehat{p}, \widehat{q}) = \sup_{z \in \mathcal{V}_G} \log f(\mathbf{A}; z, \widehat{p}, \widehat{q}) + O_P(\rho_n).$$

Now we consider $F(\boldsymbol{RSR}^\top(z))$:

$$F(\boldsymbol{RSR}^\top(z)) = \mathbb{E}(F(\boldsymbol{O}(z)/\mu_n))$$

$$= \frac{1}{\mu_n}\left( C_1(z^*) + \log \frac{\widehat{\lambda}_1(1-\widehat{p})}{\widehat{\lambda}_2(1-\widehat{q})} \left( n_2(z^*, z)q - n_1(z^*, z)p \right) \right)$$

$$= \frac{1}{\mu_n}\left( C_1(z^*) + \log \frac{\widehat{\lambda}_1(1-\widehat{p})}{\widehat{\lambda}_2(1-\widehat{q})} \left( n_2(z^*, z) \vee n_1(z^*, z)(\lambda_2 - \lambda_1) + c_0(z) \right)\rho_n \right),$$

where $\widehat{\lambda}_1 = \widehat{p}/\rho_n, \widehat{\lambda}_2 = \widehat{q}/\rho_n$ and $C_1(z^*) = \log \frac{\widehat{\lambda}_1}{1-\widehat{p}}\widetilde{n}(z^*)p + \log \frac{\widehat{\lambda}_2}{1-\widehat{q}}(n(n-1)/2 - \widetilde{n}(z^*))q$, and $c_0(z) \le \lambda_1 K c_K^2/2, \forall z \in \mathcal{C}_0 \cup \mathcal{C}_1$.

Thus when $z^* \in \mathcal{C}_0$ and $\widetilde{\mathcal{C}} = \mathcal{C}_1$, it can be easily perceived that $\mathcal{V}_G \subseteq B_{z^*}(r_K)$ with high probability, and hence

$$\sup_{z \in \mathcal{C}_1} \log f(\mathbf{A}; z, \widehat{p}, \widehat{q}) = \sup_{z \in B_{z^*}(r_K)} \log f(\mathbf{A}; z, \widehat{p}, \widehat{q}) + O_P(\rho_n).$$

Moreover, when $z^* \in \widetilde{\mathcal{C}}$, $\mathcal{V}_G \subseteq B_{z^*}(r^*)$ with high probability, where $r^* = \lambda_1 K c_K^2 / \{2(\lambda_1 - \lambda_2)\} = O(1)$. By Lemma 5.3 in Zhang & Zhou (2016), for any $z \in B_{z^*}(r^*)$, if $z \ne z^*$, then $d(z^*, z) = \Omega(n)$. Therefore, $B_{z^*}(r^*) = \{z^*\}$. In other words, we have

$$\sup_{z \in \widetilde{\mathcal{C}}} \log f(\mathbf{A}; z, \widehat{p}, \widehat{q}) = \log f(\mathbf{A}; z^*, \widehat{p}, \widehat{q}) + O_P(\rho_n). \tag{F.1}$$

More concretely, if we take $\widetilde{\mathcal{C}} = \mathcal{C}_0 \cup \mathcal{C}_1$, we have

$$\sup_{z \in \mathcal{C}_0 \cup \mathcal{C}_1} \log f(\mathbf{A}; z, \widehat{p}, \widehat{q}) = \log f(\mathbf{A}; z^*, \widehat{p}, \widehat{q}) + O_P(\rho_n),$$

and if $z^* \in \mathcal{C}_1$, we have

$$\sup_{z \in \mathcal{C}_1} \log f(\mathbf{A}; z, \widehat{p}, \widehat{q}) = \log f(\mathbf{A}; z^*, \widehat{p}, \widehat{q}) + O_P(\rho_n).$$

### F.3 CONSISTENCY OF PROBABILITY ESTIMATION

Recall the estimators $\widehat{p}$ and $\widehat{q}$ are defined in (2.9), and that $\widehat{\lambda}_1 = \widehat{p}/\rho_n$ and $\widehat{\lambda}_2 = \widehat{q}/\rho_n$. The following lemma shows that $\widehat{\lambda}_1$ and $\widehat{\lambda}_2$ are consistent.

**Lemma F.1.** Under the same condition of Theorem B.2, we have

$$|\widehat{\lambda}_i - \lambda_i| = O(1/\sqrt{n^2\rho_n}), \quad i = 1, 2$$

*Proof.* From Lemma 1 and Theorem 2 in Bickel et al. (2013), we know that $|\log(\widehat{p}/(1-\widehat{p})) - \log(p/(1-p))| = O(1/\sqrt{n^2\rho_n})$ and $|\log(\widehat{q}/(1-\widehat{q})) - \log(q/(1-q))| = O(1/\sqrt{n^2\rho_n})$. We let $\nu_1$ and $\nu_2$ denote the logit of $p$ and $q$. Then since $(\nu_1, \nu_2)$ is a one-to-one function of $(p, q)$, we know the relationship between $(\widehat{\nu}_1, \widehat{\nu}_2)$ and $(\widehat{p}, \widehat{q})$ should be $\widehat{\nu}_1 = \log \widehat{p}/(1-\widehat{p})$ and $\widehat{\nu}_2 = \log \widehat{q}/(1-\widehat{q})$. Then we have

$$\begin{aligned}
\widehat{\nu}_i - \nu_i &= \log \frac{\widehat{\lambda}_i \rho_n}{1 - \widehat{\lambda}_i \rho_n} - \log \frac{\lambda_i \rho_n}{1 - \lambda_i \rho_n} \\
&= \log \widehat{\lambda}_i \rho_n - \log \lambda_i \rho_n + \log(1 - \lambda_i \rho_n) - \log(1 - \widehat{\lambda}_i \rho_n) \\
&= \log\left(1 + \frac{\widehat{\lambda}_i - \lambda_i}{\lambda_i}\right) - \log\left(1 + \frac{(\lambda_i - \widehat{\lambda}_i)\rho_n}{1 - \lambda_i \rho_n}\right) \\
&= (1 + o(1))\frac{\widehat{\lambda}_i - \lambda_i}{\lambda_i} + (1 + o(1))\frac{(\widehat{\lambda}_i - \lambda_i)\rho_n}{1 - \lambda_i \rho_n} \\
&\asymp \widehat{\lambda}_i - \lambda_i
\end{aligned}$$

and thus by previous results we have

$$|\widehat{\lambda}_i - \lambda_i| = O(1/\sqrt{n^2\rho_n}), \quad i = 1, 2$$

$\square$

## G ADDITIONAL NUMERICAL RESULTS

We present in this section some additional numerical results to further evaluate the performance of our method. In Table 2 and Table 3, we provide the numerical results of the type-I and type-II errors

corresponding to Figure 4 (a) and Figure 5 at different settings of $\Delta$. We list the values of the metric product $d(\mathcal{C}_0, \mathcal{C}_1)I(p, q)$ corresponding to each setting for reference.

| | $n = 200$ | | | $n = 600$ | | | $n = 1000$ | | |
|---|---|---|---|---|---|---|---|---|---|
| $\Delta$ | $d(\mathcal{C}_0, \mathcal{C}_1)I(p, q)$ | Type-I error | Type-II error | $d(\mathcal{C}_0, \mathcal{C}_1)I(p, q)$ | Type-I error | Type-II error | $d(\mathcal{C}_0, \mathcal{C}_1)I(p, q)$ | Type-I error | Type-II error |
| 0.005 | 0.002 | 0.928 | 0.050 | 0.003 | 0.924 | 0.058 | 0.005 | 0.934 | 0.104 |
| 0.010 | 0.007 | 0.922 | 0.056 | 0.013 | 0.922 | 0.074 | 0.018 | 0.930 | 0.110 |
| 0.020 | 0.027 | 0.912 | 0.042 | 0.053 | 0.924 | 0.062 | 0.073 | 0.930 | 0.118 |
| 0.050 | 0.166 | 0.926 | 0.050 | 0.330 | 0.918 | 0.068 | 0.458 | 0.920 | 0.082 |
| 0.100 | 0.666 | 0.906 | 0.074 | 1.322 | 0.728 | 0.244 | 1.835 | 0.198 | 0.766 |
| 0.200 | 2.687 | 0.136 | 0.882 | 5.331 | 0.004 | 1.000 | 7.399 | 0.010 | 1.000 |
| 0.300 | 6.134 | 0.008 | 1.000 | 12.160 | 0.062 | 0.002 | 16.876 | 0.050 | 0.000 |
| 0.400 | 11.141 | 0.044 | 0.002 | 22.060 | 0.042 | 0.000 | 30.606 | 0.046 | 0.000 |
| 0.500 | 17.927 | 0.044 | 0.000 | 35.441 | 0.040 | 0.000 | 49.155 | 0.054 | 0.000 |
| 0.600 | 26.853 | 0.060 | 0.000 | 52.984 | 0.038 | 0.000 | 73.453 | 0.048 | 0.000 |
| 0.700 | 38.542 | 0.060 | 0.000 | 75.864 | 0.036 | 0.000 | 105.114 | 0.040 | 0.000 |

Table 2: Type-I and type-II errors of Example 1.1.

| | $n = 200$ | | | $n = 600$ | | | $n = 1000$ | | |
|---|---|---|---|---|---|---|---|---|---|
| $\Delta$ | $d(\mathcal{C}_0, \mathcal{C}_1)I(p, q)$ | Type-I error | Type-II error | $d(\mathcal{C}_0, \mathcal{C}_1)I(p, q)$ | Type-I error | Type-II error | $d(\mathcal{C}_0, \mathcal{C}_1)I(p, q)$ | Type-I error | Type-II error |
| 0.005 | 0.001 | 0.942 | 0.054 | 0.002 | 0.922 | 0.088 | 0.002 | 0.936 | 0.068 |
| 0.010 | 0.003 | 0.944 | 0.058 | 0.007 | 0.910 | 0.086 | 0.009 | 0.926 | 0.078 |
| 0.020 | 0.013 | 0.938 | 0.062 | 0.026 | 0.916 | 0.078 | 0.037 | 0.914 | 0.082 |
| 0.050 | 0.084 | 0.948 | 0.056 | 0.165 | 0.924 | 0.084 | 0.229 | 0.916 | 0.088 |
| 0.100 | 0.336 | 0.902 | 0.082 | 0.663 | 0.788 | 0.232 | 0.919 | 0.236 | 0.786 |
| 0.200 | 1.357 | 0.122 | 0.844 | 2.674 | 0.000 | 1.000 | 3.707 | 0.002 | 1.000 |
| 0.300 | 3.098 | 0.000 | 0.998 | 6.100 | 0.002 | 0.994 | 8.455 | 0.004 | 0.994 |
| 0.400 | 5.627 | 0.008 | 0.982 | 11.067 | 0.042 | 0.000 | 15.334 | 0.044 | 0.000 |
| 0.500 | 9.054 | 0.050 | 0.086 | 17.780 | 0.050 | 0.000 | 24.627 | 0.030 | 0.000 |
| 0.600 | 13.562 | 0.044 | 0.000 | 26.581 | 0.042 | 0.000 | 36.800 | 0.038 | 0.000 |
| 0.700 | 19.465 | 0.040 | 0.000 | 38.059 | 0.034 | 0.000 | 52.662 | 0.036 | 0.000 |

Table 3: Type-I and type-II errors of Example 1.3.

We also compare our method with an alternative method SIMPLE proposed in Fan et al. (2019). SIMPLE was designed to conduct a two-sample test on whether two nodes belong to the same community. This is the case of Example 1.1 in our paper with $m = 2$. We compare SIMPLE with our method under Example 1.1 with $m \geq 2$ by combining SIMPLE with Bonferroni correction. We consider $n = 600$, $K = 2$ and $m = 2, 3, 28$, where $m = 28$ is chosen by setting $\delta = 0.7$ in the formula $\lceil (n/K)^\delta / 2 \rceil$. The probabilities $p$ and $q$ are chosen in the same way as in Section 5. The results of the comparison are shown in Table 4. We applied the Bonferroni Correction to the SIMPLE method when doing the multiple comparisons. From the results, our method outperforms SIMPLE in both the single-pair testing and the multiple testing. For the multiple testing results, Bonferroni Correction would result in a more conservative type-I error, yet the reported size from SIMPLE is still larger than the desired size of 0.05, indicating a lack of accuracy of the method under our setting of parameters. In comparison, the performance of our method is good and stable under all settings.

To evaluate the performance of our method as the cluster size and the number of clusters increase, we provide in Table 5 additional simulation results of Example 1.1 on the type-I error, type-II error and runtime under different settings. Specifically, we set $\rho_n = n^{-\delta_{\rho_n}}$, where $\delta_{\rho_n}$ is the signal strength parameter taking values at 0.15, 0.3 and 0.5 respectively, and we set $p = 1.5\rho_n$, $q = 0.5\rho_n$ and $m = \lceil (n/K)^{0.5}/2 \rceil$. Consistent with the results in Table 2 and Table 3, we have high type-I and type-II errors when $d(\mathcal{C}_0, \mathcal{C}_1)I(p, q)$ is approximately smaller than 10, suggesting that we need large enough $d(\mathcal{C}_0, \mathcal{C}_1)I(p, q)$ to perform valid hypothesis testing.

| $m = 2$ | $\Delta$ | 0.1 | 0.2 | 0.3 | 0.4 | 0.5 | 0.6 | 0.7 |
|---|---|---|---|---|---|---|---|---|
| Our Method | Size | 0.770 | 0.008 | 0.056 | 0.040 | 0.060 | 0.048 | 0.056 |
| | Power | 0.768 | 0.018 | 0.998 | 1.000 | 1.000 | 1.000 | 1.000 |
| SIMPLE | Size | 0.270 | 0.116 | 0.102 | 0.086 | 0.082 | 0.076 | 0.064 |
| | Power | 0.312 | 0.756 | 0.990 | 1.000 | 1.000 | 1.000 | 1.000 |
| $m = 3$ | $\Delta$ | 0.1 | 0.2 | 0.3 | 0.4 | 0.5 | 0.6 | 0.7 |
| Our Method | Size | 0.752 | 0.014 | 0.048 | 0.048 | 0.044 | 0.046 | 0.046 |
| | Power | 0.728 | 0.018 | 0.998 | 1.000 | 1.000 | 1.000 | 1.000 |
| SIMPLE | Size | 0.344 | 0.118 | 0.070 | 0.074 | 0.070 | 0.084 | 0.070 |
| | Power | 0.426 | 0.816 | 0.992 | 1.000 | 1.000 | 1.000 | 1.000 |
| $m = 28$ | $\Delta$ | 0.1 | 0.2 | 0.3 | 0.4 | 0.5 | 0.6 | 0.7 |
| Our Method | Size | 0.792 | 0.000 | 0.054 | 0.040 | 0.036 | 0.030 | 0.038 |
| | Power | 0.784 | 0.000 | 0.752 | 1.000 | 1.000 | 1.000 | 1.000 |
| SIMPLE | Size | 0.938 | 0.232 | 0.126 | 0.088 | 0.072 | 0.06 | 0.044 |
| | Power | 0.950 | 0.998 | 1.000 | 1.000 | 1.000 | 1.000 | 1.000 |

Table 4: Comparison of our method with SIMPLE for Example 1.1 with $m = 2, 3, 28$.

| $n/K$ | $K$ | $\delta_{\rho_n}$ | $d(\mathcal{C}_0,\mathcal{C}_1)I(p,q)$ | Type-I error | Type-II error | Runtime (in seconds) |
|---|---|---|---|---|---|---|
| 100 | 2 | 0.150 | 45.787 | 0.077 | 0.000 | 2.066 |
| 100 | 2 | 0.300 | 13.657 | 0.067 | 0.000 | 2.065 |
| 100 | 2 | 0.500 | 4.038 | 0.007 | 1.000 | 2.041 |
| 100 | 3 | 0.150 | 40.696 | 0.060 | 0.000 | 7.498 |
| 100 | 3 | 0.300 | 11.735 | 0.043 | 0.000 | 7.471 |
| 100 | 3 | 0.500 | 3.252 | 0.007 | 1.000 | 7.659 |
| 100 | 4 | 0.150 | 37.597 | 0.083 | 0.000 | 16.029 |
| 100 | 4 | 0.300 | 10.565 | 0.040 | 0.000 | 15.894 |
| 100 | 4 | 0.500 | 2.793 | 0.037 | 0.987 | 15.326 |
| 100 | 5 | 0.150 | 35.432 | 0.060 | 0.000 | 27.801 |
| 100 | 5 | 0.300 | 9.752 | 0.100 | 0.000 | 27.722 |
| 100 | 5 | 0.500 | 2.484 | 0.043 | 0.963 | 26.485 |
| 150 | 2 | 0.150 | 61.250 | 0.050 | 0.000 | 8.989 |
| 150 | 2 | 0.300 | 17.661 | 0.047 | 0.000 | 8.951 |
| 150 | 2 | 0.500 | 4.894 | 0.003 | 1.000 | 8.977 |
| 150 | 3 | 0.150 | 54.829 | 0.057 | 0.000 | 37.333 |
| 150 | 3 | 0.300 | 15.241 | 0.053 | 0.000 | 37.347 |
| 150 | 3 | 0.500 | 3.951 | 0.007 | 1.000 | 36.861 |
| 150 | 4 | 0.150 | 50.867 | 0.077 | 0.000 | 82.281 |
| 150 | 4 | 0.300 | 13.758 | 0.027 | 0.000 | 82.371 |
| 150 | 4 | 0.500 | 3.399 | 0.003 | 0.993 | 79.962 |
| 150 | 5 | 0.150 | 48.077 | 0.067 | 0.000 | 135.419 |
| 150 | 5 | 0.300 | 12.722 | 0.057 | 0.003 | 135.634 |
| 150 | 5 | 0.500 | 3.026 | 0.007 | 1.000 | 130.579 |
| 200 | 2 | 0.150 | 75.573 | 0.047 | 0.000 | 30.893 |
| 200 | 2 | 0.300 | 21.238 | 0.040 | 0.000 | 31.069 |
| 200 | 2 | 0.500 | 5.614 | 0.007 | 1.000 | 30.606 |
| 200 | 3 | 0.150 | 67.937 | 0.063 | 0.000 | 115.165 |
| 200 | 3 | 0.300 | 18.375 | 0.053 | 0.000 | 115.183 |
| 200 | 3 | 0.500 | 4.540 | 0.010 | 1.000 | 114.633 |
| 200 | 4 | 0.150 | 63.188 | 0.070 | 0.000 | 232.970 |
| 200 | 4 | 0.300 | 16.614 | 0.027 | 0.000 | 233.203 |
| 200 | 4 | 0.500 | 3.909 | 0.007 | 1.000 | 227.977 |
| 200 | 5 | 0.150 | 59.828 | 0.077 | 0.000 | 372.845 |
| 200 | 5 | 0.300 | 15.380 | 0.030 | 0.003 | 371.971 |
| 200 | 5 | 0.500 | 3.483 | 0.003 | 1.000 | 363.691 |
| 300 | 2 | 0.150 | 102.076 | 0.057 | 0.000 | 133.766 |
| 300 | 2 | 0.300 | 27.609 | 0.040 | 0.000 | 131.782 |
| 300 | 2 | 0.500 | 6.821 | 0.007 | 1.000 | 125.900 |
| 300 | 3 | 0.150 | 92.227 | 0.053 | 0.000 | 406.462 |
| 300 | 3 | 0.300 | 23.964 | 0.060 | 0.000 | 412.823 |
| 300 | 3 | 0.500 | 5.526 | 0.007 | 1.000 | 409.896 |
| 300 | 4 | 0.150 | 86.047 | 0.067 | 0.000 | 867.116 |
| 300 | 4 | 0.300 | 21.708 | 0.040 | 0.000 | 865.617 |
| 300 | 4 | 0.500 | 4.763 | 0.007 | 1.000 | 867.681 |
| 300 | 5 | 0.150 | 81.647 | 0.047 | 0.000 | 1490.006 |
| 300 | 5 | 0.300 | 20.123 | 0.030 | 0.000 | 1514.533 |
| 300 | 5 | 0.500 | 4.247 | 0.000 | 1.000 | 1450.086 |
| 400 | 2 | 0.150 | 126.694 | 0.060 | 0.000 | 377.973 |
| 400 | 2 | 0.300 | 33.311 | 0.037 | 0.000 | 374.540 |
| 400 | 2 | 0.500 | 7.838 | 0.000 | 1.000 | 374.889 |
| 400 | 3 | 0.150 | 114.825 | 0.057 | 0.000 | 1349.159 |
| 400 | 3 | 0.300 | 28.969 | 0.037 | 0.000 | 1353.448 |
| 400 | 3 | 0.500 | 6.356 | 0.007 | 1.000 | 1336.669 |
| 400 | 4 | 0.150 | 107.334 | 0.037 | 0.000 | 2715.127 |
| 400 | 4 | 0.300 | 26.274 | 0.053 | 0.000 | 2810.849 |
| 400 | 4 | 0.500 | 5.483 | 0.000 | 1.000 | 2796.724 |
| 400 | 5 | 0.150 | 101.982 | 0.063 | 0.000 | 4626.111 |
| 400 | 5 | 0.300 | 24.376 | 0.033 | 0.000 | 4583.932 |
| 400 | 5 | 0.500 | 4.891 | 0.003 | 1.000 | 4525.389 |

Table 5: Type-I error, type-II error and runtime (in seconds) for Example 1.1 under different cluster size $n/K$, number of clusters $K$ and signal strength parameter $\delta_{\rho_n}$.

