# OpenReview forum: "Combinatorial-Probabilistic Trade-Off: P-Values of Community Properties Test in the Stochastic Block Models"
_ICLR.cc/2023/Conference — ICLR 2023 notable top 25%_

### Official Review · Reviewer_XzCa · 2022-10-24

**Confidence:** 4
**Correctness:** 3
**Technical Novelty And Significance:** 3
**Empirical Novelty And Significance:** Not applicable
**Recommendation:** 8

**Clarity, Quality, Novelty And Reproducibility:**

Minor remarks:
- $\rho_n$ should be defined earlier, since it is used as early as section 2.3
- you make a lot of reference to diverse assumptions/examples, the paper would be slightly easier to follow if they were hyperlinked
- In the definition of LRT (2.3), why is the sup in the denominator over $\mathcal C_0 \cup \mathcal C_1$ ? This is especially weird since the remainder of the paper seems to assume that this sup always belongs to $\mathcal C_0$.
- I had some trouble understanding the notion of packing that you introduce in Definition 4.1, especially when $\epsilon = 0$; could you expand on it a bit more ?
- in the numerical results section, could you specify the value of the threshold $I(p, q)d(\mathcal C_0, \mathcal C_1)$ so that we can see how it matches with the predictions ?

**Strength And Weaknesses:**

This is overall a very enjoyable to read paper; it departs from the classical point of view on the SBM, and introduces several new tools to handle hypothesis testing in this setting. It is overall quite clear: all technical definitions are accompanied with figures, and the authors included several extended examples throughout the paper to illustrate the main results. The more technical setting is relegated to the appendix,

My main reservation about the paper is about its practical usefulness. The fact that $\rho_n \gg n^{\epsilon - 1}$ is glossed upon a bit too fast: it is a scaling where the actual assignment $z^*$ can be recovered with absurdly overwhelming probability, and hence the p-values for any (rejected) test should be extremely small. It also does not seem to be a technical assumption, since a non-negligible level of signal is certainly needed for a Gaussian approximation to hold. Can you explain why your framework is more powerful/useful than simply computing $z^*$ and then testing whichever hypotheses we want ?


**Summary Of The Paper:**

This paper studies the problem of hypothesis testing in community detection: instead of asking to recover the exact community assignments, the goal is to perform hypothesis testing on them (e.g. asking whether two given nodes belong to the same community or not).

The authors introduce a framework for measuring distances between community assignments, that depends on the number of mismatched edges between the two assignments. Using this notion, they propose a test based on Gaussian approximation to discriminate between hypothesis classes $\mathcal C_0$ and $\mathcal C_1$. They show that this test has the prescribed asymptotic power, i.e. that the p-values are asymptotically the same for the original hypothesis test and its Gaussian approximation. Finally, they show an information-theoretic lower bound involving the SBM parameters and the classes themselves, under which no test can discriminate between the two.



**Summary Of The Review:**

See above

---

> ### Author Response · Authors · 2022-11-16
> **Response to Reviewer XzCa**
>
> > My main reservation about the paper is about its practical usefulness. The fact that $\rho_n \gg n^{\epsilon -1}$ is glossed upon a bit too fast: it is a scaling where the actual assignment $z^*$ can be recovered with absurdly overwhelming probability, and hence the p-values for any (rejected) test should be extremely small. It also does not seem to be a technical assumption, since a non-negligible level of signal is certainly needed for a Gaussian approximation to hold. Can you explain why your framework is more powerful/useful than simply computing $z^*$ and then testing whichever hypotheses we want?
>
> Thank you very much for proposing this insightful comment. A similar concern was raised by Reviewer uY7c, and we restate our response here for the convenience of reference.  For the scaling condition of the signal strength $\rho_n \gg n^{\epsilon -1}$, indeed exact recovery is possible as $n \rightarrow \infty$. However, the exact recovery is an asymptotic result, and there are no existing methods characterizing the uncertainty of community testing for finite $n$, and we do not know how much gap there is between the asymptotic and the finite scenarios. Without uncertainty quantification or the p-value, we do not know when we shall reject the null for a finite sample, and our paper aims to provide a concrete threshold for general community property testing.
>
> > $\rho_n$ should be defined earlier, since it is used as early as section 2.3
>
> Thank you for pointing this out. We moved the definition of $\rho_n$ to Section 1 in the revised version.
>
> > you make a lot of reference to diverse assumptions/examples, the paper would be slightly easier to follow if they were hyperlinked
>
> Thank you very much for the suggestion. We fixed the hyperlink in the revised version for easier reference.
>
> > In the definition of LRT (2.3), why is the sup in the denominator over $\mathcal{C}_0 \cup \mathcal{C}_1$? This is especially weird since the remainder of the paper seems to assume that this sup always belongs to $\mathcal{C}_0$.
>
> Thank you for pointing out the ambiguity. The equivalence between the supremum over ${\mathcal{C}_0 \cup \mathcal{C}_1} $ and the supremum over ${\mathcal{C}_0 } $ is only under the null, and we add a clarification in Section 2 to address the potential confusion. We take the supremum over ${\mathcal{C}_0 \cup \mathcal{C}_1} $  to guarantee that the approximation in (2.4) is valid  under both $\mathcal{C}_0$ and $\mathcal{C}_1$ according to (C.2) in Lemma C.2, such that we can efficiently estimate its quantile via shadowing bootstrap.
>
> > I had some trouble understanding the notion of packing that you introduce in Definition 4.1, especially when $\epsilon = 0$; could you expand on it a bit more?
>
> Thank you for the question. In the revised version, we added a short comment before Definition 4.1 to explain the intuition. In our paper, we define the packing number based upon the combinatorial metric $ | \mathcal{E}^{1,2} (z_0, z_j) \cap \mathcal{E}^{1,2} (z_0, z_k) | $, with $\mathcal{E}^{1,2}(z_0,z_k)$ being the union of misaligned edge sets $\mathcal{E}^1 (z_0,z_k)$ and $\mathcal{E}^2 (z_0,z_k)$ defined in Definition 2.3. The intuition is that for any two assignments $z_j, z_k \in {\mathcal{C}}_1$, the metric $|\mathcal{E}^{1,2}(z_0,z_j) \cap \mathcal{E}^{1,2}(z_0,z_k)|$ measures how many common misaligned edges they share. Intuitively, the more misaligned edges there are, the closer $z_j$ and $z_k$ should be, and hence $|\mathcal{E}^{1,2}(z_0,z_j) \cap \mathcal{E}^{1,2}(z_0,z_k)| \le \epsilon$ is comparable to the constraint $\|\theta_i - \theta_j\| > \epsilon$ for the traditional definition of packing number. When $\epsilon = 0$, the constraint forces that $z_j$ and $z_k$ should be sufficiently different from each other. For example, in Example 1.1, for a null assignment $z_0$, and $z_i, z_j $ in the boundary corresponding to $z_0$, if $z_i$ and $z_j$ are obtained by relabeling nodes $i_1 \in [m]$ and $i_2 \in [m]$ of $z_0 $ respectively, then  $|\mathcal{E}^{1,2} (z_0,z_j) \cap \mathcal{E}^{1,2} (z_0,z_k)| \ge 1$ since the edge $(i_1, i_2)$ will always be contained in the misaligned set $\mathcal{E}^{1,2}(z_0,z_j) \cap \mathcal{E}^{1,2}(z_0,z_k)$, which results in the packing number of the boundary being equal to 1.  On the other hand, if we relax a little by taking $\epsilon = \sqrt{d(z_0, \mathcal{C}_1)}$, more common misaligned edges are allowed and we will have the packing number of the boundary equal to $m$ for Example 1.1.

---

> ### Author Response · Authors · 2022-11-16
> **Response to Reviewer XzCa**
>
>
> > in the numerical results section, could you specify the value of the threshold $I(p,q)d(\mathcal{C}_0, \mathcal{C}_1)$  so that we can see how it matches with the predictions?
>
> Thank you very much for this suggestion. We added the threshold of the metric product $I(p,q)d(\mathcal{C}_0, \mathcal{C}_1)$ in the numerical section to indicate how large we need the product to be for the algorithm to work.  We also added the values of $I(p,q)d(\mathcal{C}_0, \mathcal{C}_1)$ corresponding to each setting in Table 1 and Table 2 in Appendix A for reference.

---

### Official Review · Reviewer_f6aS · 2022-10-24

**Confidence:** 5
**Correctness:** 2
**Technical Novelty And Significance:** 3
**Empirical Novelty And Significance:** 2
**Recommendation:** 8

**Clarity, Quality, Novelty And Reproducibility:**

The paper is relatively clear in terms of the model and intuition, although there are some areas that need polishing and better understanding of the performance (most notably, conditions that make the framework desirable to be used over other alternatives) require further evaluation. The quality of the ideas are interesting and the model is intriguing. The novelty is more with respect to a possible implementation of the LRT for SBM models though a more thorough evaluation could be beneficial.

**Strength And Weaknesses:**

With respect to its strengths, the paper is well organized, the problem is relevant to the ML community, particularly to hypothesis testing SBM modeling and could be applicable to clustering and other domains. The paper states a clear goals although it is at times obscure in its notation. Another strength is the thorough analysis of permutation impacting computation of likelihood in the SBM which highlights the practical difficulties of LRT for probabilistic models due to isomorphisms. The Shadowing Bootstrap to estimate the p-value of the LRT is also interesting but it seems that it may present scaling challenges.
There are some areas where the paper could be improved. For instance, the evaluations are interesting on the impact of $\Delta$ (distinguishing an SBM from an Erdös-Rényi model) but not fully developed. The computational costs and runtimes (e.g. as K and n grows) could be interesting to identifyfy applicability of the technique and possible future work to ensure scalability. Another weakness is that no evaluation is provided with respect to the choice of other parameters. In general the contribution is more theoretical.

**Summary Of The Paper:**

This paper presents a framework to test whether a certain [symmetric] community property of the stochastic block model (SBM) is satisfied and calculate p-values the quantify the uncertainty. The paper introduces a shadowing bootstrap method to deal with the combinatorial challenges of the test. The framework relies a combinatorial distance (a.k.a community property distance in Def. 2.4) between communities and uses it together with a probabilistic distance (difference of community parameters/probabilities) to ensure the test remains “honest”. The model works applying a likelihood ratio test for the SBM, where the limiting distribution of the likelihood ratio is driven by the two edge sets of misaligned edges (assigned to the same community by one assignment in a community property and to different communities by another assignment in the other property). The paper also presents an analysis of the limiting distribution of the leading term of the LRT statistic and a Shadowing Bootstrap for the test. The theoretical component of the paper concludes with both: a validity analysis of the proposed test for general symmetric community properties, and an alternative (information theoretical) lower bound result relaxing the scaling conditions of the test. The experiments show numerical evaluations (Type I and II errors) of the shadowing bootstrap to test the community hypothesis with $\alpha=0.5$ for various sizes of networks $n$ and varying difference between $p$ and $q$ (i.e., $\Delta$).

**Summary Of The Review:**

The paper has its pros, such as the relevance, the clear stated goals and the analysis of the effect of $\Delta$ over Type I and II errors. However, there are several reservations with respect to the applicability of the framework because a hypothesis testing technique is as important as its application. The advantage seems to be more due to the some of the insights but more evaluations are highly recommended. I detailed my comments on what could be improved in the Strength and Weaknesses section.

%%%Post Rebuttal Comment%%%

Thank you for the answer to my questions. This clarifies most of my doubts and I raise my score.

---

> ### Author Response · Authors · 2022-11-16
> **Response to Reviewer f6aS**
>
> >  There are some areas where the paper could be improved. For instance, the evaluations are interesting on the impact of $\Delta$ (distinguishing an SBM from an Erdös-Rényi model) but not fully developed. The computational costs and runtimes (e.g. as $K$ and $n$ grows) could be interesting to identifyfy applicability of the technique and possible future work to ensure scalability. Another weakness is that no evaluation is provided with respect to the choice of other parameters. In general the contribution is more theoretical.
>
> Thank you very much for your helpful comments. We appreciate your suggestion on improving the numerical evaluation and added extra simulation results in the revised version to provide a more thorough assessment of the scalability and performance of our method. More specifically, in Appendix A of the revised version, we provided  in Table 1 and Table 2 the values of the metric product $I(p,q)d(\mathcal{C}_0, \mathcal{C}_1)$ corresponding to different $\Delta$ to display how the impact of $\Delta$ is mediated by $I(p,q)d(\mathcal{C}_0, \mathcal{C}_1)$, and to provide a more straightforward insight on approximately how large we need $I(p,q)d(\mathcal{C}_0, \mathcal{C}_1)$ to be in order to conduct a valid test.
>
> In Table 3 of Appendix A, we run some additional simulations on Example 1.1 to evaluate how the type-I error, type-II error and the runtime will change as $n$ and $K$ increase. We also evaluate the performance of the method as the signal strength varies. We list the value of $I(p,q)d(\mathcal{C}_0, \mathcal{C}_1)$ corresponding to each setting to validate the results in Table 1 and Table 2.
>
> We also added a comparison of our method with a well-celebrated algorithm SIMPLE for Example 1.1. We show in Table 4 of Appendix A that our method has an advantage over SIMPLE when it comes to multiple testing.
>
> We hope the added simulations may help to address some concerns and thank you again for providing the constructive suggestions that help us to improve the paper.

---

### Official Review · Reviewer_uY7c · 2022-10-26

**Confidence:** 4
**Clarity, Quality, Novelty And Reproducibility:** 1. The signal strength $\rho_n$ is no…
**Correctness:** 4
**Technical Novelty And Significance:** 3
**Empirical Novelty And Significance:** 3
**Recommendation:** 8

**Strength And Weaknesses:**

Strength

1. I believe this paper studies a very important and very general problem about stochastic block model.
2. The authors provide examples for definitions, which help the reader understand the paper better.
3. Overall, this paper is well-written. The statements in definitions, assumptions and theorems are clear.
4. The theoretical results are novel and interesting.

Weakness

There is a critical limitation in the theoretical result, which has been stated after Theorem 3.2. This main theorem assumes $1/\rho_n=o(n^c)$ for $c\in(0,1)$. In the existing result, if $n\rho_n/\log n\to\infty$, then then the assignments can be exactly recover with high probability. The assumption in Theorem 3.2 is even stronger than $n\rho_n/\log n\to\infty$. The case when exact recovery is not possible is still very interesting.

I also have some comments and questions about the clarity in the next part.

**Summary Of The Paper:**

This paper studies a very general hypothesis testing problem about the community assignement on stochastic block model. Given two disjoint community properties, the difficulty of distinguishing between the properties depends on distance between properties and probability distance between communities in the stochastic block model. The authors propose a method to define p-value, which can be applied to define the rejection rule of the test. The author also derive the information-theoretic lower bound.

**Summary Of The Review:**

I believe this is a good paper if the author can clarify my concerns.

---

> ### Author Response · Authors · 2022-11-16
> **Response to Reviewer uY7c**
>
> Thank you very much for the insightful comments. On the concern over the scaling condition of the signal strength $1/\rho_n = o(n^c)$ for $c \in (0,1)$, indeed exact recovery is possible as $n \rightarrow \infty$. However, the exact recovery is an asymptotic result, and there are no existing methods characterizing the uncertainty of community testing for finite $n$, and we do not know how much gap there is between the asymptotic and the finite scenarios. Without uncertainty quantification or the p-value, we do not know when we shall reject the null for a finite sample, and our paper aims to provide a concrete threshold for general community property testing.
>
> >The signal strength $\rho_n$ is not defined when it appears in Section 2.3. I understand the definition of $\rho_n$ when it appears again in Section 3 before Theorem 3.2. I think the authors should move the definition of $\rho_n$ to Section 2.
>
> Thank you very much for pointing this out. In the revised version, we moved the definition of $\rho_n$ to Section 1 (highlighted in blue) when introducing the SBM.
>
> > I am very confused about the definition of likelihood ratio statistic in equation (2.3). In classical hypothesis testing, the likelihood ratio statistic should be defined as
> $$\frac{\sup_{\theta \in \Theta_0}f(X;\theta)}{\sup_{\theta \in \Theta_0 \cup \Theta_1}f(X;\theta)}$$
> and we reject the $H_0$ if this statistic is small. In the definition in the paper, the roles of $\Theta_0$  and $\Theta_1$ are exchanged. I think the author should justify why they use a different definition for likelhood ratio statistic. Let us consider a special case of hypothesis test: $\mathcal{C}_0$ contains a single assignment $z_0$ and  $\mathcal{C}_1$ is the set of all other assignments, and suppose $z_0$ is not the MLE, then LRT=0. In this case, LRT only tells us if $z_0$ is the MLE. Is the LRT still useful in this test?
>
> Thank you for proposing this insightful question and for providing an interesting example. As for the form of the LRT, the traditional LRT in classical hypothesis testing is for the continuous parameter space, whereas when the null and the alternative are both combinatorial, the LRT will have a fundamentally different behavior due to the discrete optimization over an exponential number of candidates. In the numerator, we replace the traditional supremum over $\mathcal{C}_0$ by the supremum over $\mathcal{C}_1$ such that under the null, the exponential number of potential candidates on the numerator can be significantly reduced to the supremum over the boundary, and we can take advantage of the symmetric community property to characterize the null distribution of the LRT, which is not feasible when the numerator is over $\mathcal{C}_0$.  As for the example of $\mathcal{C}_0 = z_0$, indeed testing under this scenario will actually reduce to a recovery problem of $z_0$ by MLE, and our proposed LRT will serve to capture the uncertainty of MLE in recovering the truth $z^*$. On the other hand, under the more general setting, the complexity of the alternative $\mathcal{C}_1$ will impact the significance level of the test statistics, and our LRT will characterize the confidence level and quantify the rejection threshold in response to different alternative spaces under different community tests.
>
> >In equation (2.9), we obtained the MLE of $p$ and $q$ for $z \in \mathcal{C}_0 \cup \mathcal{C}_1$. Why can we also use these MLE for
> $z \in \mathcal{C}_0$ in equation (2.10).
>
> Thank you for this very good question. We apply $\hat{p}$ and $\hat{q}$ obtained from $\mathcal{C}_0 \cup \mathcal{C}_1$ for both $\mathcal{C}_0 \cup \mathcal{C}_1$ and $\mathcal{C}_1$ because $p$ and $q$ here act like nuisance parameters, and our inferential interest lies in the combinatorial parameter space of the community assignments ($\mathcal{C}_0$ and $\mathcal{C}_1$) rather than the continuous parameter space of $p$ and $q$. Differentiation of $p$ and $q$ between $\mathcal{C}_0$ and $\mathcal{C}_1$ will have minor contribution to the test statistic. So long as their plug-in estimators are consistent (which is the case for supremum over ${z \in \mathcal{C}_0 \cup \mathcal{C}_1}$), we will have a valid test statistic. Hence we also use $\hat{p}$ and $\hat{q}$ obtained from ${z \in \mathcal{C}_0 \cup \mathcal{C}_1}$ for $\mathcal{C}_1$ to improve computational efficiency.

---

> > ### Comment · Reviewer_uY7c · 2022-11-18
> > **Response to authors**
> >
> > Thanks for your response.
> >
> > I understand exact recovery (strong consistency) is an asymptotic result. Can we still discuss the p-value when only weak consistency (proportion of wrong labels tends to 0) can be achieved?
> >
> > I can partially understand and agree with the explanations to the other two questions. I think they should appear in the revision (perhaps in the appendix).

---

> > > ### Author Response · Authors · 2022-11-18
> > > **Response to Reviewer uY7c**
> > >
> > > > I understand exact recovery (strong consistency) is an asymptotic result. Can we still discuss the p-value when only weak consistency (proportion of wrong labels tends to 0) can be achieved?
> > >
> > > Thank you very much for your response and for the insightful question. Since weak consistency requires very weak signal strength, even when valid p-value is achievable, it is hard to guarantee both valid p-values and minimax power under a general framework. On the other hand, it is shown in [1] that MLE can achieve weak consistency, and we believe that the MLE will still provide valid p-values for specific categories of $\mathcal{C}_0$ and $\mathcal{C}_1$ under this scenario, but may require establishing a very different theoretical framework. But still this remains a very interesting question that we will try to explore in future work.
> > >
> > > > I can partially understand and agree with the explanations to the other two questions. I think they should appear in the revision (perhaps in the appendix).
> > >
> > > Thank you very much for your suggestion. We will add a section in the appendix for the discussion and will upload the revised version soon.
> > >
> > >
> > >
> > > [1] Zhang, Anderson Y., and Harrison H. Zhou. "Minimax rates of community detection in stochastic block models." The Annals of Statistics 44.5 (2016): 2252-2280.

---

### Official Review · Reviewer_on1z · 2022-10-27

**Confidence:** 3
**Correctness:** 4
**Technical Novelty And Significance:** 3
**Empirical Novelty And Significance:** 2
**Recommendation:** 8

**Clarity, Quality, Novelty And Reproducibility:**

The writing of the paper is clear and the ideas a novel.



**Strength And Weaknesses:**

Strengths:
- The paper addresses a very important problem and proposes the use of a novel distance metric to quantify the quality of the propose statistical test.
- The results are complemented with lower bounds showing that the new distance metric is indeed inherent to the problem.

Weaknesses:
- It seems that the examples on which the test is applied in the paper, do not capture the full generality of the framework. So, it would be nice to see a characterization (or at least a discussion) of what type of problems could be solved using this test.
- To that end, it might also be useful to discuss about how restrictive Assumption 3.1 is in terms of the type of problems that satisfy it.

Minor comments:

- Last line of section 2.1: "$[K]^m$" should be "$[K]^{m+m^\prime}$"

- Section 2.4, lines 9-10: "supreme"->"supremum"

- Page 5, 2nd line from the bottom: "there existing"->"there exists"


**Summary Of The Paper:**

This paper is about testing properties of stochastic block models. In particular, the framework presented in the paper is applicable to symmetric properties. A property is defined to be a set of assignments of the $n$ nodes into $K$ communities and it can be described by a subset of $[K]^n$. A symmetric property is a property that contains all of the assignments that partition a specific subset $\mathcal{N}\subseteq [n]$ of the nodes in a specific way (the labels can be permuted though). The alternative hypothesis for the tests considered in the paper, need to be a union of symmetric properties for the same subset $\mathcal{N}$.

The authors define a notion of distance that depends both on combinatorial properties of the partitions and the probabilities of the models in different instances. In particular, the proposed distance is a product of the 2 distances, which implies a trade-off: the null hypothesis can be rejected if either the edge probabilities are close and the instances are very different combinatorially, or vice versa. Using a likelihood ratio statistic, it is shown that the test accepts the null hypothesis with constant probability when the minimum distance does not exceed a certain threshold and rejects it with probability that approaches 1 when the distance does go above a different threshold. The authors complement these results with information theoretic lower bounds, which show that the aforementioned trade-off is indeed necessary.

**Summary Of The Review:**

The main contribution of this paper is showing that the right metric (justified by upper and lower bounds) for a testing symmetric properties of stochastic block models is a product of a combinatorial and a probabilistic distance metric. I consider this to be a solid contribution and recommend acceptance.

---

> ### Author Response · Authors · 2022-11-16
> **Response to Reviewer on1z**
>
> > Last line of section 2.1: "$[K]^m$" should be "$[K]^{m + m'}$"; Section 2.4, lines 9-10: "supreme"->"supremum"; Page 5, 2nd line from the bottom: "there existing"->"there exists"
>
> Thank you very much for pointing out the typos. We have made corrections in the revised version.
>
> > It seems that the examples on which the test is applied in the paper, do not capture the full generality of the framework. So, it would be nice to see a characterization (or at least a discussion) of what type of problems could be solved using this test; To that end, it might also be useful to discuss about how restrictive Assumption 3.1 is in terms of the type of problems that satisfy it.
>
> Thank you very much for the suggestion. We add a remark after Assumption 3.1 in the revised version to provide a more intuitive understanding on when Assumption 3.1 is satisfied. In general, the proposed framework can perform test on symmetric problems, with the number of worst cases ($B_{z_0}$) in the alternative upper bounded by some polynomial of $n$. A testing problem is symmetric basically when the community properties $\mathcal{C}_0$ and $\mathcal{C}_1$ are robust to permutations of labels. As for the number of worst cases imposed by Assumption 3.1, if changing the community labels for a constant number of nodes can change an assignment from $\mathcal{C}_0$ to $\mathcal{C}_1$, then the number of possible choices will be upper bounded by ${n \choose c_0} = O(n^{c_0}) $ for some integer $c_0 > 0$ and Assumption 3.1 will be satisfied.

---

### Author Response · Authors · 2022-11-16
**Thanks for all the reviewers and revised version uploaded**

We first would like to express our sincere gratitude to the reviewers for putting their valuable time into the reviewing process and for providing constructive suggestions that help us make improvements.  We uploaded a revised version of our paper with major changes colored in blue. We provide in Appendix A some additional numerical results for the evaluation of our method. We will provide detailed responses to each reviewer in the comments below.

---

### Decision · Program_Chairs · 2023-01-20

**Decision:**

Accept: notable-top-25%

**Justification For Why Not Higher Score:**

I believe the paper is suitable for a spotlight presentation but not an oral one because there might be others papers that are more deserving of an oral presentation. I don't have the full information, and hence, I can't tell, so I'll be conservative.

**Justification For Why Not Lower Score:**

Clearly, this paper should not be rejected.

**Metareview: Summary, Strengths And Weaknesses:**

This paper considers the important problem of testing properties of stochastic block models, e.g., whether two nodes belong to the same community. A novel distance metric is introduced and the achievability results are complemented by information-theoretic lower bounds. All reviewers agreed that the contributions are significant.

**Note From Pc:**

if the above contains the word "oral" or "spotlight" please see: "oral" presentation means -> notable-top-5% and "spotlight" means -> notable-top-25%. As stated in our emails, we are disassociating presentation type from AC recommendations

**Summary Of Ac-Reviewer Meeting:**

NIL